# Climate change favours large seasonal loss of Arctic ozone

Peter von der Gathen [1✉], Rigel Kivi [2], Ingo Wohltmann [1], Ross J. Salawitch [3] & Markus Rex [1,4]

Chemical loss of Arctic ozone due to anthropogenic halogens is driven by temperature, with more loss occurring during cold winters favourable for formation of polar stratospheric clouds (PSCs). We show that a positive, statistically significant rise in the local maxima of PSC formation potential (PFP[LM]) for cold winters is apparent in meteorological data collected over the past half century. Output from numerous General Circulation Models (GCMs) also exhibits positive trends in PFP[LM] over 1950 to 2100, with highest values occurring at end of century, for simulations driven by a large rise in the radiative forcing of climate from greenhouse gases (GHGs). We combine projections of stratospheric halogen loading and humidity with GCM-based forecasts of temperature to suggest that conditions favourable for large, seasonal loss of Arctic column $O_3$ could persist or even worsen until the end of this century, if future abundances of GHGs continue to steeply rise.

[1] Alfred Wegener Institute, Helmholtz Centre for Polar and Marine Research, Potsdam, Germany. [2] Finnish Meteorological Institute, Space and Earth Observation Centre, Sodankylä, Finland. [3] Department of Atmospheric and Oceanic Science, Department of Chemistry and Biochemistry, and Earth System Science Interdisciplinary Center, University of Maryland, College Park, MD, USA. [4] Universität Potsdam, Institut für Physik und Astronomie, Potsdam, Germany. ✉email: Peter.von.der.Gathen@awi.de

Variations in ozone within the Arctic polar vortex during winter and spring (hereafter: winter) are driven by anthropogenic chemical loss and dynamical resupply[1,2]. Chemical loss and dynamical resupply of stratospheric ozone show large inter-annual variability, driven by meteorology. Colder, more isolated vortices are associated with smaller values of total column ozone[3,4], less resupply and larger chemical loss of ozone (due to low temperatures). Colder vortices are caused by a weaker Brewer-Dobson Circulation, reduced planetary-scale wave activity and lower eddy heat flux in the extratropical lower stratosphere[5]. The coldest Arctic winters experience the smallest values of total column ozone, due in part to a larger amount of chemical loss[3,4].

Chemical loss of $O_3$ in the Arctic stratosphere occurs following the activation of chlorine on or within cold sulphate aerosols[6,7] and supercooled ternary ($H_2SO_4$-$HNO_3$-$H_2O$) solution droplets[8] (STS), and on the surfaces of nitric acid trihydrate (NAT) particles[9] or water ice when air is exceptionally cold. When temperatures fall during Arctic winter, STS and NAT particles[10–12] are the first types of PSCs to form. The timescale for chemical processing of chlorine reservoir gases on STS droplets transitions from weeks to days near the temperature at which NAT becomes thermodynamically stable ($T_{NAT}$)[7], which is governed by the vapour pressure of nitric acid ($HNO_3$) and water ($H_2O$)[9].

The volume of air cold enough to allow for the existence of polar stratospheric clouds (PSCs) in the Arctic polar vortex, averaged over an ozone loss season ($V_{PSC}$), exhibits a compact, near-linear relation with chemical loss of column ozone[13–17] during recent winters. Rex et al.[13] postulated that the maximum value of $V_{PSC}$ during Arctic winters had risen in a statistically significant manner between 1966 and 2003, and suggested this increase was caused by radiative and dynamical effects of rising levels of greenhouse gases (GHGs). New record values of $V_{PSC}$ were set in the winters of 2005 (ref. [14]), 2011 (ref. [3]), 2016 (refs. [18,19]), and 2020 (ref. [20]). An early evaluation using a general circulation model (GCM) with coupled active chemistry (a chemistry climate model, or CCM) suggested decreases in planetary wave activity reaching the mid-latitude stratosphere due to increased westerly winds in the subtropics, driven by rising levels of GHGs, would lead to stronger, colder Arctic vortices[21]. More recently, a simulation using another CCM suggested that future cooling of the Arctic lower stratosphere during early winter would result from direct radiative cooling driven by GHGs and indirect effects related to declining Arctic sea ice and rising sea surface temperatures[22]. Simulations conducted using a third CCM showed modest cooling (~0.15 K decade$^{-1}$) of the future Arctic stratosphere at 50 hPa also driven by GHGs, with high interannual variability that complicates the assessment of statistical significance[23].

Here we examine trends in the PSC formation potential (PFP), which represents the number of days a volume of air equal to the volume of the polar vortex was exposed to PSC conditions for each Arctic ozone loss season based on $T_{NAT}$ (similar to ref. [24]). We show that positive, statistically significant trends in the local maxima (LM) of the PFP timeseries (PFP$^{LM}$, the upper quartile of PFP relative to a trend line) over the past four decades are apparent in data from four meteorological centres. A central component of our analysis is the examination of output from GCMs that provide estimates of stratospheric conditions until the end of this century, with a focus on models that submitted output for the Shared Socioeconomic Pathways SSP5-8.5, SSP3-7.0, SSP2-4.5, and SSP1-2.6 runs of Climate Model Intercomparison Project Phase 6 (CMIP6)[25]. We combine GCM forecasts of PFP with projections of stratospheric halogen loading and stratospheric humidity to evaluate how the chemical loss of Arctic ozone may evolve, as a function of future levels of atmospheric GHGs and stratospheric $H_2O$. We find that if the future abundance of GHGs continues to rise steeply as in either the SSP3-7.0 or SSP5-8.5 scenario, then continued growth in the atmospheric conditions favourable for large, seasonal loss of column ozone could persist or even worsen until the end of this century, despite the decline in the abundance of anthropogenic halogens that is expected to occur due to compliance with the Montreal Protocol.

## Results

**Chemical loss of ozone.** Figure 1a shows values of column ozone loss between 380 and 550 K potential temperature ($\Delta O_3$) at the end of winter, based on ozonesonde measurements in the Arctic vortex, plotted as a function of PFP (see "Methods" for the detailed definition of PFP). Data values are shown for all of the cold winters that have occurred since the inception of regular ozonesonde launches. The estimates of $\Delta O_3$ are based either on Match events (situations where individual air masses are usually probed twice above different measurement stations)[13,14,17,26] or on the difference between a passive ozone tracer and the vortex mean, observed profile of ozone[20]. Figure 1a also shows computations of $\Delta O_3$ found using the ATLAS Chemistry and Transport Model[27] for meteorological conditions of Arctic winters 2005, 2010, 2011, and 2020. This model includes a comprehensive treatment of stratospheric chemistry, constrained by the abundance of stratospheric chlorine and bromine from long-lived source gases (Fig. 2a) for these four winters[28] plus a constant 5 parts per trillion (pptv) from very short-lived (VSL) bromocarbons[29] (see "Methods").

Measured and modelled values of $\Delta O_3$ display a compact, near-linear relation with PFP for 1993–2020 (data) and 2005–2020 (ATLAS) (Fig. 1a). This behaviour occurs because over this time period, the abundance of stratospheric halogens, commonly represented by equivalent effective stratospheric chlorine (EESC)[30] (Fig. 2a), varies by only ~11% between the value in early 1993 and the maximum in mid-2001. Modelled values of $\Delta O_3$ lie either close to measured $\Delta O_3$ (2011 and 2020) or just below the 1σ uncertainty (2005 and 2010), demonstrating that the primary control on interannual variations in $\Delta O_3$ over the past 15 years has been the exposure of air to PSC temperatures. The near-linear relation between $\Delta O_3$ and $V_{PSC}$ is a robust relation for the contemporary Arctic stratosphere[16,17], despite the fact that in early winter, a small volume of the Arctic vortex can exist below the temperature threshold for chlorine activation and affect a large portion of the vortex[31]. Figure 1a also contains values of $\Delta O_3$ for years 2060 and 2100 computed using the ATLAS model, for projected stratospheric chlorine and bromine for both years, and meteorological conditions for 2020. Modelled $\Delta O_3$ for 2060 and 2100 falls below the compact relation observed and simulated for the contemporary atmosphere due to the projected future decline in EESC (Fig. 2a).

Figure 1b shows measured and modelled values of $\Delta O_3$ as a function of a term we shall refer to as ozone loss potential (OLP), defined as:

$$OLP(yr) = \frac{EESC(yr)^{1.2}}{EESC_{MAX}^{1.2}} \times PFP(yr) \qquad (1)$$

where $EESC_{MAX}$ (4.45 ppbv) is the maximum yearly value of EESC in the polar stratosphere. The variance, $r^2$, in $\Delta O_3$ explained by OLP is quite large, exhibiting values of $r^2$ of 0.89 and 0.96 for measured and modelled $\Delta O_3$, respectively (Fig. 1b). Our OLP is defined in a manner nearly identical to the potential for activation of chlorine term of Tilmes et al.[32], except for the use of 1.2 rather than 1 as the exponent of EESC in Eq. (1). Hassler et al.[33] conducted an analysis of ozone depletion and recovery at

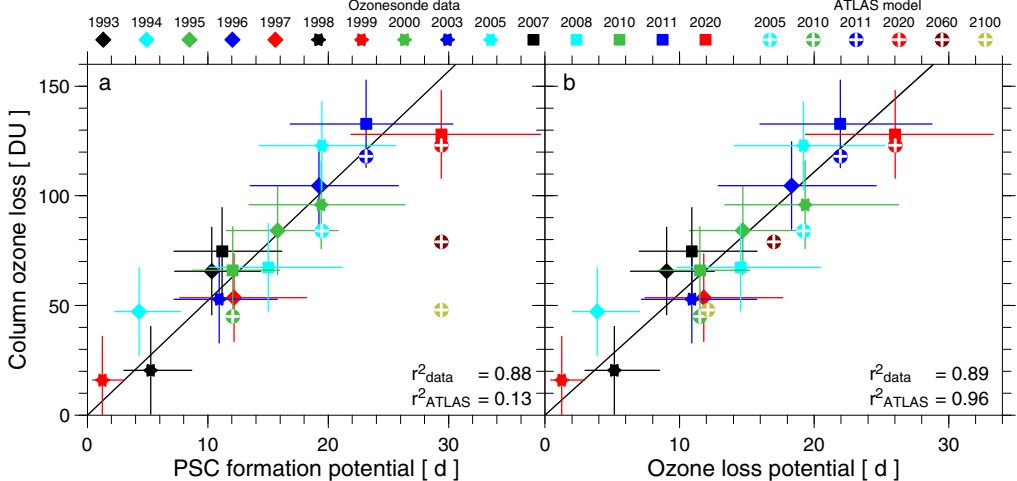

**Fig. 1 Chemical loss of Arctic Ozone. a** Chemical loss of column ozone ($\Delta O_3$) in Dobson Units (DU; 1 DU = 2.687 × 10$^{16}$ molecules cm$^{-2}$) inside the Arctic polar vortex determined by ozonesonde campaigns for various winters since 1993 versus PSC formation potential (PFP) computed from ERA5/ERA5.1 (closed symbols), calculated as the vertical integral of loss profiles between 380 and 550 K potential temperature, which is ~14 and ~24 km altitude. The error bars representing 1σ uncertainty for ozone loss are based upon considerations such as uncertainties in the calculated cooling rates and the potential impact of mixing across the edge of the vortex edge as described in Harris et al.[17]; the 1σ uncertainty for PFP is derived by assuming an error of ±1 K in the ERA5/ERA5.1 temperature field (see "Methods"). Computations of $\Delta O_3$ are found using the global ATLAS Chemistry and Transport Model that includes a comprehensive treatment of stratospheric chemistry, for the halogen loading and meteorological conditions of winter 2005, 2010, 2011, and 2020 as well as halogen loading for 2060 and 2100 with meteorological conditions for 2020 (symbols with crosses). The ATLAS values of $\Delta O_3$ are also based on integrals between the 380 and 550 K potential temperature[20]. **b** Same as panel **a** except ozone loss potential (OLP) is used for the abscissa. The variance in observed (data) and modelled (ATLAS) $\Delta O_3$ explained by PFP and by OLP is reported as the square of the correlation coefficient in both panels. The solid line on both panels shows a linear, least-squares fit to the 15 ozonesonde data points, forced through the origin.

the South Pole assuming a linear relation between ozone loss rate and EESC, even though they state the actual relation may be more complicated. Harris et al.[17] examined model estimates of accumulated ozone losses at the 500 K potential temperature level in the Arctic stratosphere as a function of the abundance of activated chlorine, and reported a small positive non-linearity in this relationship. Here we use an exponent of 1.2 for EESC because this choice leads to the largest value of $r^2$ for the six ATLAS runs shown in Fig. 1b (see "Methods"). The linear, least-squares regression of the ozonesonde-based estimates of $\Delta O_3$ versus OLP in Fig. 1b will be used below to relate estimates of the future evolution of OLP inferred from GCMs to the seasonal loss of Arctic ozone, which we denote $\Delta O_3^{REG}$. We assess the uncertainty in $\Delta O_3^{REG}$ using lower and upper limits of 1 and 1.4 for the exponent in the expression for OLP (see "Methods").

**Observed PSC formation potential.** Figure 3 shows time series of PFP found using data from four meteorological centres (see "Methods"). Our primary source of meteorological data is ERA5/ERA5.1/ERA5 BE (preliminary version) provided by the European Centre for Medium-Range Weather Forecasts (ECMWF)[34]. We also use meteorological fields from Climate Forecast System Reanalysis (CFSR/CFSv2) provided by the National Centers for Environmental Prediction of the U.S. National Oceanic and Atmospheric Administration[35,36], the Modern-Era Retrospective analysis for Research and Applications (MERRA-2) product provided by the U.S. National Aeronautics and Space Administration Goddard Earth Observing System Model[37,38], as well as the Japanese 55-year Reanalysis (JRA-55) provided by the Japanese Meteorological Agency (JMA)[39]. We calculate $V_{PSC}$ based on temperature and wind fields from these meteorological reanalyses to evaluate the consistency of our estimates of $V_{PSC}$ and to assess the robustness of inferred trends in PFP. Diagnostics for the existence of PSCs can vary sub-stantially between reanalyses, such that conclusions based on the

often marginal conditions for PSC condensation in the NH could be affected by small differences among the reanalyses[40].

Meteorological fields from ERA5 have recently been extended back to 1950 and data from JRA-55 are available from 1958 to 2020, whereas the other data sets are available from 1979 (or 1980) to 2020. Stratospheric data in the Arctic mainly rely on radiosonde soundings before 1979 and on satellite data thereafter, which could introduce potential bias (see "Methods"). We use ERA5 and JRA-55 only back to 1965 since this year marks the start of regular radiosonde coverage of the Arctic stratosphere. Finally, reanalyses transitioned from the use of space-borne data from SSU and TOVS to AMSU and ATOVS systems in the 1998 to 1999 timeframe[40]. We obtain similar results for trends in PFP$^{LM}$ (differences within respective uncertainties) when considering data obtained prior and after this transition (see "Methods").

As noted in the Introduction, we had previously suggested a tendency for the highest values of $V_{PSC}$ to have risen over time. These analyses[13,14] were based upon the selection of maximum values of $V_{PSC}$ over successive 5 year time intervals, a trend detection procedure we term here the Maximum in the Interval Method (MIM). Since the publication of these papers, we have developed a more accurate and robust trend detection procedure as documented by a series of Monte-Carlo (MC) simulations (see "Methods"), termed the Iterative Selection Approach (ISA).

The slope of the LM of PFP ($S_{PFP-LM}$) selected by ISA is strongly positive over 1980 to 2020 based upon analysis of data from all four meteorological centres, ranging from a high of 4.77 ± 0.48 d decade$^{-1}$ (CSFR) to a low of 3.85 ± 0.40 d decade$^{-1}$ (MERRA-2) (Fig. 3). The mean and 1σ standard deviation of $S_{PFP-LM}$ over 1980 to 2020 from these four centres is 4.26 ± 0.45 d decade$^{-1}$. The values of $S_{PFP-LM}$ over the longer time period of 1965 to 2020 are 3.84 ± 0.34 d decade$^{-1}$ and 3.50 ± 0.29 d decade$^{-1}$ based on ERA5 and JRA-55, respectively, the only data sets that extend further back than 1979, the start of the modern satellite era. In other words, during particularly cold winters over the past half

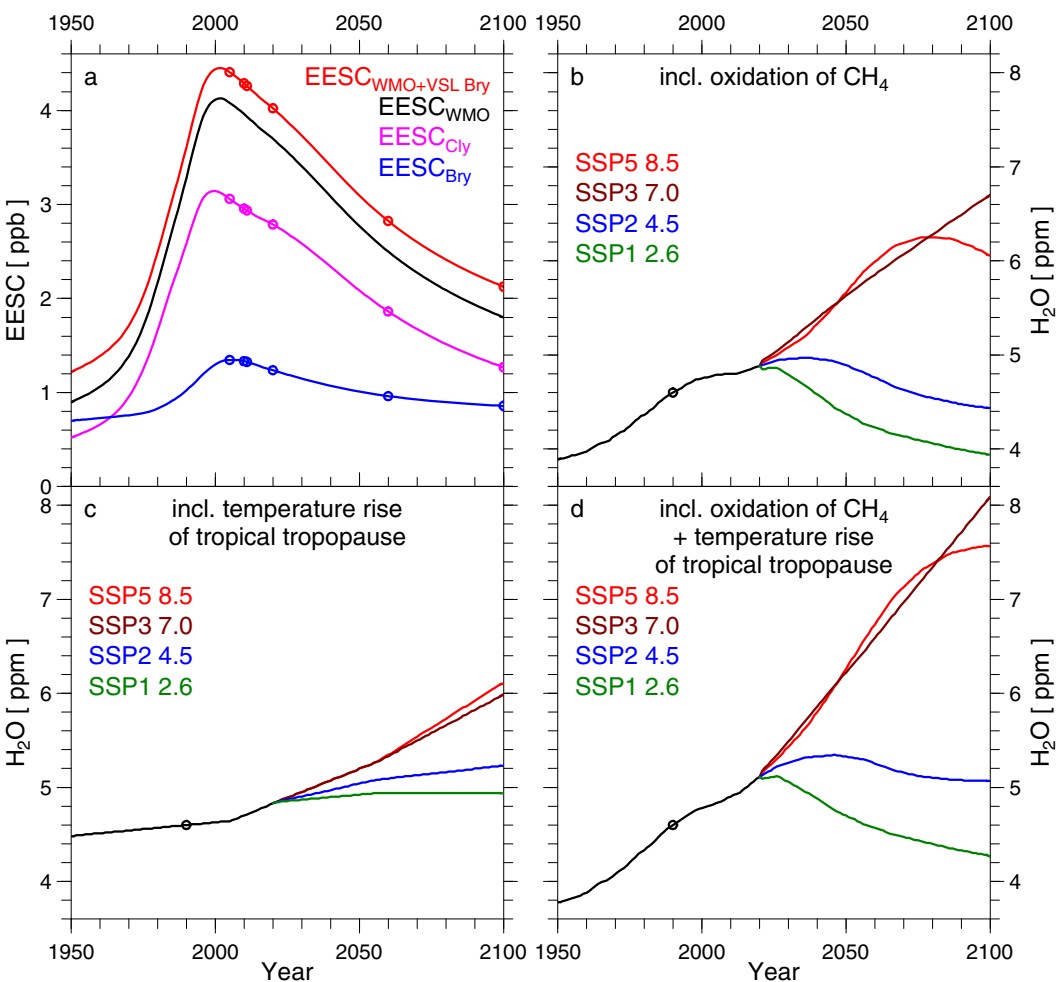

**Fig. 2 Polar Stratospheric EESC and H₂O. a** EESC (equivalent effective stratospheric chlorine) for the polar stratosphere computed using fractional release factors from Newman et al.[30] and values of the abundances of long-lived halogen source gases from Table 6-4 of most recent WMO Ozone Assessment Report[28] (black line). Throughout, we use a slightly modified version of polar EESC, found by accounting for a 5 ppt contribution from very short-lived (VSL) bromocarbons[29] (red line; circles denote years of the ATLAS simulations shown in Fig. 1). The contribution to this modified polar EESC from stratospheric chlorine and bromine are shown by the violet and blue lines, respectively. **b–d** polar stratospheric H₂O (in several SSP scenarios) found accounting for: variations in atmospheric CH₄ (**b**); the temperature rise of the tropical tropopause layer (TTL) (**c**); both CH₄ and warming of the TTL (**d**) (see "Methods"). The circle denotes H₂O = 4.6 ppm, used to compute PFP whenever time-invariant H₂O is specified. (Historical part: black lines; SSP1-2.6: green lines; SSP2-4.5: blue lines; SSP3-7.0: brown lines; SSP5-8.5: red lines).

century, the Arctic polar vortex has tended to experience between 3.5 and 4.8 more days per decade of exposure to conditions cold enough to sustain PSCs and activate chlorine, an increase of about 40% compared to the values that occurred a half century ago. We have conducted MC simulations to assess the statistical significance of $S_{PFP-LM}$ and the $1\sigma$ uncertainty in $S_{PFP-LM}$ ($\Delta S_{PFP-LM}$) found using the ISA selection procedure (see "Methods"). These simulations indicate statistical significance at better than the $2\sigma$ confidence level for this important metric of the trend in $PFP^{LM}$, based upon $p$-values for $S_{PFP-LM}/\Delta S_{PFP-LM}$ from all four meteorological data centres that are <0.001 (see "Methods", Table 1).

**PSC formation potential from GCMs.** In this section, we calculate PFP from the output of all 26 GCMs in CMIP6 that archived results for the SSP5-8.5 scenario[25]. The numerical value after the dash in the SSP designation represents the rise in radiative forcing of climate (RF; units W m⁻²) at end of the century relative to pre-industrial, due to GHGs including ozone-depleting substances as well as tropospheric aerosols[41]. Temperature fields within these GCMs often exhibit biases with respect to observed temperature

that can approach 5 K, with most models being biased warm[42]. Stratospheric H₂O tends to be biased low in many models[43], which together with a high-temperature bias will lead to an underestimation of the accumulated exposure to PSCs in the Arctic. To compensate for the temperature biases, the temperature threshold for the existence of PSCs has been offset by a constant value specific to each model such that the overall magnitude of $PFP^{LM}$ in the GCM matches the observed magnitude of $PFP^{LM}$ over the modern satellite era. Furthermore, the computation of PFP uses profiles for H₂O and HNO₃ for the contemporary stratosphere (see "Methods").

Values of PFP for the SSP5-8.5 run of 16 of the 20 GCMs that submitted results for all four SSPs highlighted in our study (SSP5-8.5, SSP3-7.0, SSP2-4.5, and SSP1-2.6) are shown in Fig. 4. PFP for the remaining SSP5-8.5 GCM runs are shown either in Fig. 5 or in the Supplementary Information (SI). The suggestion that the coldest Arctic winters are getting colder is also apparent in GCM simulations without adjusting the PSC temperature threshold (see SI). We highlight results with adjusted thresholds to place all of the GCMs on a common scale for assessing PFP in the Arctic stratosphere.

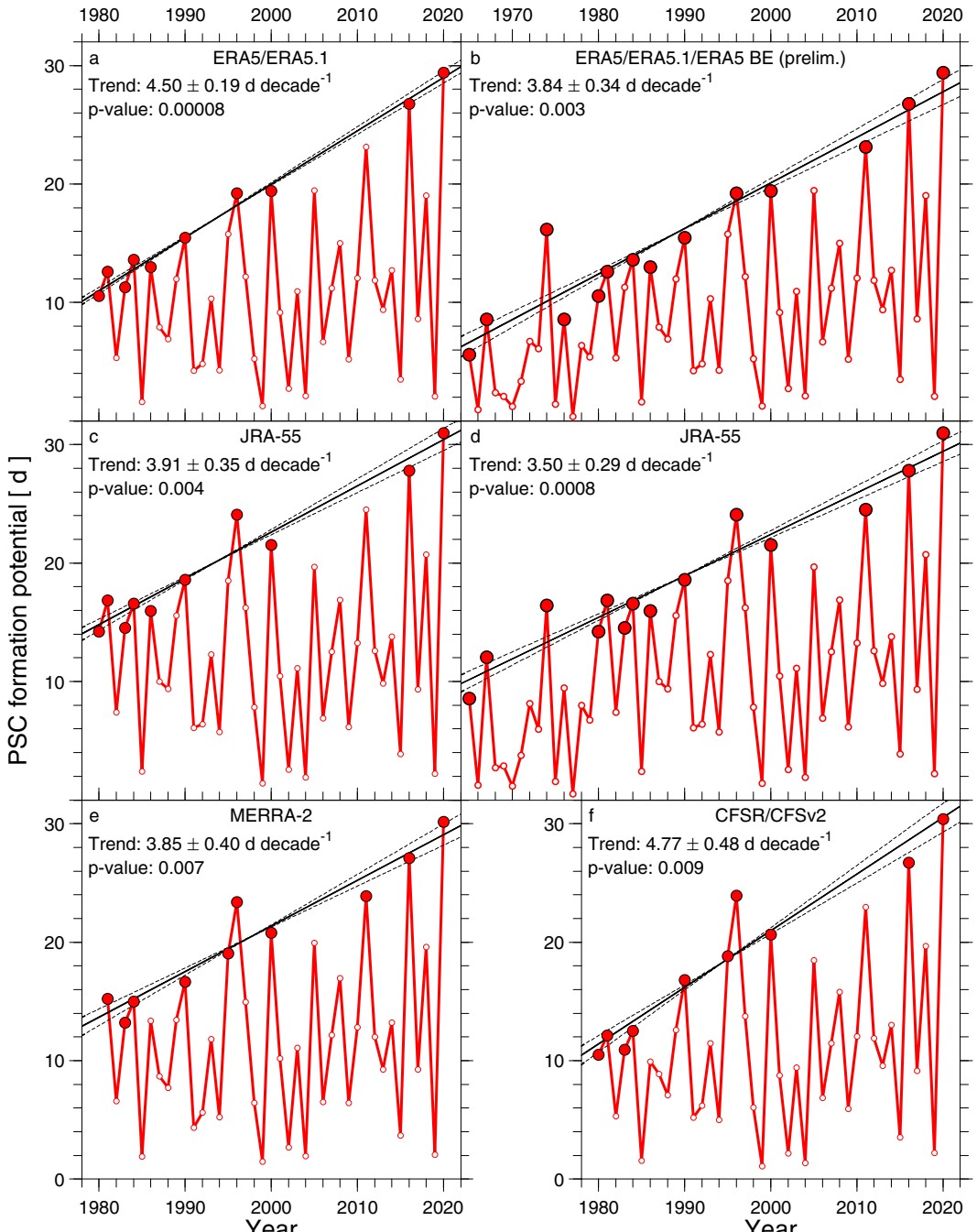

**Fig. 3 PFP as a function of time. a–f** Time series of PSC formation potential (PFP) for reanalysis data from: ERA5/ERA5.1 from 1980 to 2020 (**a**) and ERA5/ERA5.1 combined with the ERA5 back extension (BE) (preliminary version) from 1965 to 2020 (**b**); JRA-55 from 1980 to 2020 (**c**) and from 1965 to 2020 (**d**); MERRA-2 from 1981 to 2020 (**e**); CFSR/CFSv1 from 1980 to 2020 (**f**). The solid red circles indicate the coldest winters in the record selected using the ISA trend detection procedure (see "Methods"). A linear, least-squares fit (solid line) and 1σ uncertainty of the fit (dashed lines) to the solid red circles are shown in each panel, along with numerical values of the slopes ($S_{PFP-LM}$), the 1σ uncertainties of these fits ($\Delta S_{PFP-LM}$), as well as p-values for the quantity $S_{PFP-LM}/\Delta S_{PFP-LM}$ (last column, Table 1).

Values of $S_{PFP-LM}$ found for each of the 26 GCM simulations with archived results for SSP5-8.5 are all positive, ranging from a high of $3.66 \pm 0.16$ d decade$^{-1}$ (IITM-ESM) to a low of $0.62 \pm 0.09$ d decade$^{-1}$ (BCC-CSM2-MR) (Table 1). The majority of these slopes lie between about 1.0 and 2.5 d decade$^{-1}$; statistical significance at better than the 2σ level is exhibited for $S_{PFP-LM}$ in 16 and for $S_{PFP-LM}/\Delta S_{PFP-LM}$ in 24 of these 26 runs. The similarity of the long-term running mean of PFP and regression of PFP versus RF in each of the panels (Fig. 5) suggests the Arctic

stratosphere is cooling in a manner that follows the rise in RF of climate. This provides further support that rising GHGs are the primary factor driving increasing PFP. Nearly all of the GCMs exhibit maximum values of PFP towards the end of the century.

The progressive tendency towards colder Arctic winters is also exhibited in GCMs that participated in the earlier CMIP5 project[44]. For CMIP5, archived output from 27 GCM simulations that ran the Representative Concentration Pathway (RCP) 8.5 (ref. [45]) is considered. The frequency distribution function of the

**Table 1 PFP$^{LM}$ trend results for the reanalyses and CMIP6 GCM output.**

| Reanalysis/GCM | Time range | $W$ | $S$ | $T_{OFFSET}$ (K) | $S_{PFP-LM} \pm \Delta S_{PFP-LM}$ (d decade$^{-1}$) | *p*-value for $S_{PFP-LM}$ | $\frac{S_{PFP-LM}}{\Delta S_{PFP-LM}}$ | *p*-value for $\frac{S_{PFP-LM}}{\Delta S_{PFP-LM}}$ |
|---|---|---|---|---|---|---|---|---|
| ERA5/ERA5.1 | 1980–2020 | 41 | 10 | 0 | 4.50 ± 0.19 | 0.18 | 23.4 | $8 \times 10^{-5}$ |
| ERA5/ERA5.1/ ERA5 BE (prelim.) | 1965–2020 | 56 | 14 | 0 | 3.84 ± 0.34 | 0.09 | 11.1 | $3 \times 10^{-3}$ |
| MERRA-2 | 1981–2020 | 40 | 10 | 0 | 3.85 ± 0.40 | 0.17 | 9.7 | $7 \times 10^{-3}$ |
| CFSR/CFSv2 | 1980–2020 | 41 | 10 | 0 | 4.77 ± 0.48 | 0.14 | 9.9 | $9 \times 10^{-3}$ |
| JRA55 | 1980–2020 | 41 | 10 | 0 | 3.91 ± 0.35 | 0.15 | 11.1 | $4 \times 10^{-3}$ |
| JRA55 | 1965–2020 | 56 | 14 | 0 | 3.50 ± 0.29 | 0.06 | 12.1 | $8 \times 10^{-4}$ |
| BCC-CSM2-MR | 1951–2100 | 150 | 38 | −6 | 0.62 ± 0.09 | $4 \times 10^{-3}$ | 7.1 | $2 \times 10^{-3}$ |
| CanESM5 | 1951–2100 | 150 | 38 | 1 | 1.13 ± 0.09 | $3 \times 10^{-3}$ | 13.1 | $<10^{-6}$ |
| CESM2 | 1951–2100 | 107 | 27 | −6 | 1.22 ± 0.11 | 0.08 | 10.7 | $8 \times 10^{-4}$ |
| CESM2-WACCM | 1951–2100 | 150 | 38 | 2 | 1.70 ± 0.23 | $9 \times 10^{-3}$ | 7.5 | 0.03 |
| CNRM-CM6-1 | 1951–2100 | 150 | 38 | −4 | 2.08 ± 0.12 | $3 \times 10^{-3}$ | 18.1 | $<10^{-6}$ |
| CNRM-CM6-1-HR | 1951–2100 | 150 | 38 | −1 | 1.34 ± 0.09 | 0.06 | 14.2 | $4 \times 10^{-5}$ |
| CNRM-ESM2-1 | 1951–2100 | 150 | 38 | −4 | 2.08 ± 0.14 | $3 \times 10^{-3}$ | 14.9 | $<10^{-6}$ |
| EC-Earth3 | 1951–2100 | 150 | 38 | 3 | 2.27 ± 0.13 | 0.05 | 17.6 | $<10^{-6}$ |
| EC-Earth3-Veg | 1951–2100 | 150 | 38 | 4 | 2.05 ± 0.10 | 0.02 | 21.4 | $<10^{-6}$ |
| FGOALS-g3 | 1951–2100 | 150 | 38 | −1 | 0.82 ± 0.22 | 0.11 | 3.8 | 0.11 |
| GFDL-CM4 | 1951–2100 | 150 | 38 | 5 | 2.33 ± 0.10 | 0.02 | 22.3 | $<10^{-6}$ |
| HadGEM3-GC31-LL | 1951–2100 | 150 | 38 | 1 | 1.73 ± 0.09 | 0.03 | 19.1 | $<10^{-6}$ |
| HadGEM3-GC31-MM | 1951–2100 | 150 | 38 | 1 | 1.53 ± 0.10 | 0.06 | 15.9 | $<10^{-6}$ |
| IITM-ESM | 1951–2099 | 149 | 37 | 4 | 3.66 ± 0.16 | 0.03 | 23.7 | $<10^{-6}$ |
| INM-CM4-8 | 1951–2100 | 150 | 38 | −1 | 2.37 ± 0.16 | $5 \times 10^{-3}$ | 14.8 | $4 \times 10^{-5}$ |
| INM-CM5-0 | 1951–2100 | 150 | 38 | 2 | 1.88 ± 0.09 | 0.06 | 20.2 | $<10^{-6}$ |
| IPSL-CM6A-LR | 1951–2100 | 150 | 38 | 6 | 1.99 ± 0.11 | $8 \times 10^{-3}$ | 18.8 | $<10^{-6}$ |
| MIROC6 | 1951–2100 | 150 | 38 | 3 | 2.91 ± 0.16 | 0.05 | 18.3 | $<10^{-6}$ |
| MIROC-ES2L | 1951–2100 | 150 | 38 | 1 | 3.41 ± 0.08 | $4 \times 10^{-5}$ | 44.8 | $<10^{-6}$ |
| MPI-ESM1-2-HR | 1951–2100 | 150 | 38 | 4 | 0.84 ± 0.11 | $3 \times 10^{-3}$ | 6.9 | $2 \times 10^{-3}$ |
| MPI-ESM1-2-LR | 1951–2100 | 150 | 38 | 2 | 1.63 ± 0.16 | $5 \times 10^{-4}$ | 10.5 | $5 \times 10^{-4}$ |
| MRI-ESM2-0 | 1951–2100 | 150 | 38 | 2 | 1.09 ± 0.10 | $8 \times 10^{-5}$ | 11.0 | $<10^{-6}$ |
| NESM3 | 1951–2100 | 150 | 38 | 2 | 1.36 ± 0.08 | $5 \times 10^{-3}$ | 17.0 | $<10^{-6}$ |
| NorESM2-LM | 1951–2100 | 150 | 38 | 4 | 2.27 ± 0.18 | $3 \times 10^{-3}$ | 12.4 | $3 \times 10^{-4}$ |
| NorESM2-MM | 1951–2100 | 150 | 38 | 3 | 1.01 ± 0.20 | 0.18 | 5.1 | 0.14 |
| UKESM1-0-LL | 1951–2100 | 150 | 38 | 1 | 1.75 ± 0.12 | $7 \times 10^{-3}$ | 14.6 | $<10^{-6}$ |

Slopes ($S_{PFP-LM}$) and corresponding uncertainties ($\Delta S_{PFP-LM}$) of linear least-squares fit through data points selected using ISA, for four reanalyses and 26 CMIP6 GCMs, as well as the ratio $S_{PFP-LM}/\Delta S_{PFP-LM}$. The number of winters ($W$), number of winters selected as local maxima for trend analysis ($S$), and the temperature threshold offset for the formation of PSCs applied to the GCM output ($T_{OFFSET}$) are given, as well as the *p*-values for $S_{PFP-LM}$ and $S_{PFP-LM}/\Delta S_{PFP-LM}$ found using Monte-Carlo simulations. A *p*-value of $<10^{-6}$ is given when fewer than ten of the ten million artificial data sets yield a value of $S_{PFP-LM}/\Delta S_{PFP-LM}$ that is larger than the observed value.

ISA-based value of $S_{PFP-LM}$ over 1950–2100, for 26 CMIP6 GCMs and 27 CMIP5 GCMs, is shown in Fig. 6. The mean and standard deviation of $S_{PFP-LM}$ are $1.71 \pm 0.7$ d decade$^{-1}$ and $1.48 \pm 1.0$ d decade$^{-1}$ for the CMIP6 and CMIP5 GCMs, respectively (Fig. 6b). The CMIP5 GCMs exhibit a greater tendency towards both low and high values of $S_{PFP-LM}$ compared to the CMIP6 GCMs. Most importantly, values of $S_{PFP-LM}$ over 1950–2100 are positive for 52 of the 53 CMIP5/6 GCM simulations forced by an 8.5 W m$^{-2}$ rise in RF by end of the century. These GCM runs provide numerical support for the contention that rising levels of GHGs will lead to cooler conditions in the polar stratosphere that are conducive to the chemical loss of ozone by anthropogenic halogens. The GCM simulations in Figs. 4 and 5 also show a tendency for PFP associated with the warmer Arctic winters (open circles at bottom of the data envelope) to rise slightly over time, a projected trend not yet apparent in observations[46] due perhaps to the generally small values of PFP for the warmest winters over the observational period as well as the lower limit of zero for PFP.

The mean and standard deviation of the empirical value of $S_{PFP-LM}$ over 1980 to 2020 from the four reanalysis datasets is compared to GCM-based values (for the same time period) in Fig. 6a. The rationale for this comparison is the models have undergone a similar rise in the RF of climate over these four decades as the atmosphere. The observationally based trend lies near the upper 1σ value of the GCMs. Over this short period internally generated climate variability may play a substantial role and the one realisation that developed in earth's climate system may have coincidentally followed a path that led to $S_{PFP-LM}$ at the upper range of the GCM values.

On the other hand, tropospheric climate exhibited a shift in the early 2000s that weakened the intensity of planetary wave activity propagating into the stratosphere[47], which could be responsible for a portion of the larger observed value of $S_{PFP-LM}$ compared to results from GCMs. Shifts in patterns of sea surface temperature in the North Pacific have also been implicated as a causal factor in decreased planetary wave activity and the strengthening of the Arctic vortex[48]. The potential association of these drivers of Arctic, stratospheric temperature with climate change is an area of active research[47]. We interpret the results in Fig. 6a as follows: there is a strong similarity in the four observationally based estimates of $S_{PFP-LM}$, and this value is consistent with a subset of the GCMs (i.e., those with the largest values of $S_{PFP-LM}$). It is difficult to attach further meaning to this comparison; because of the potential role of internal variability in planetary wave activity, we caution against asserting that GCMs with the best match to the empirically based value $S_{PFP-LM}$ will provide a more realistic forecast of the future.

As further support for the notion that larger values of PFP towards the end of the century are driven by rising levels of GHGs, we analyse results for the 20 GCM simulations that have provided an output for SSP5-8.5, SSP3-7.0, SSP2-4.5, and

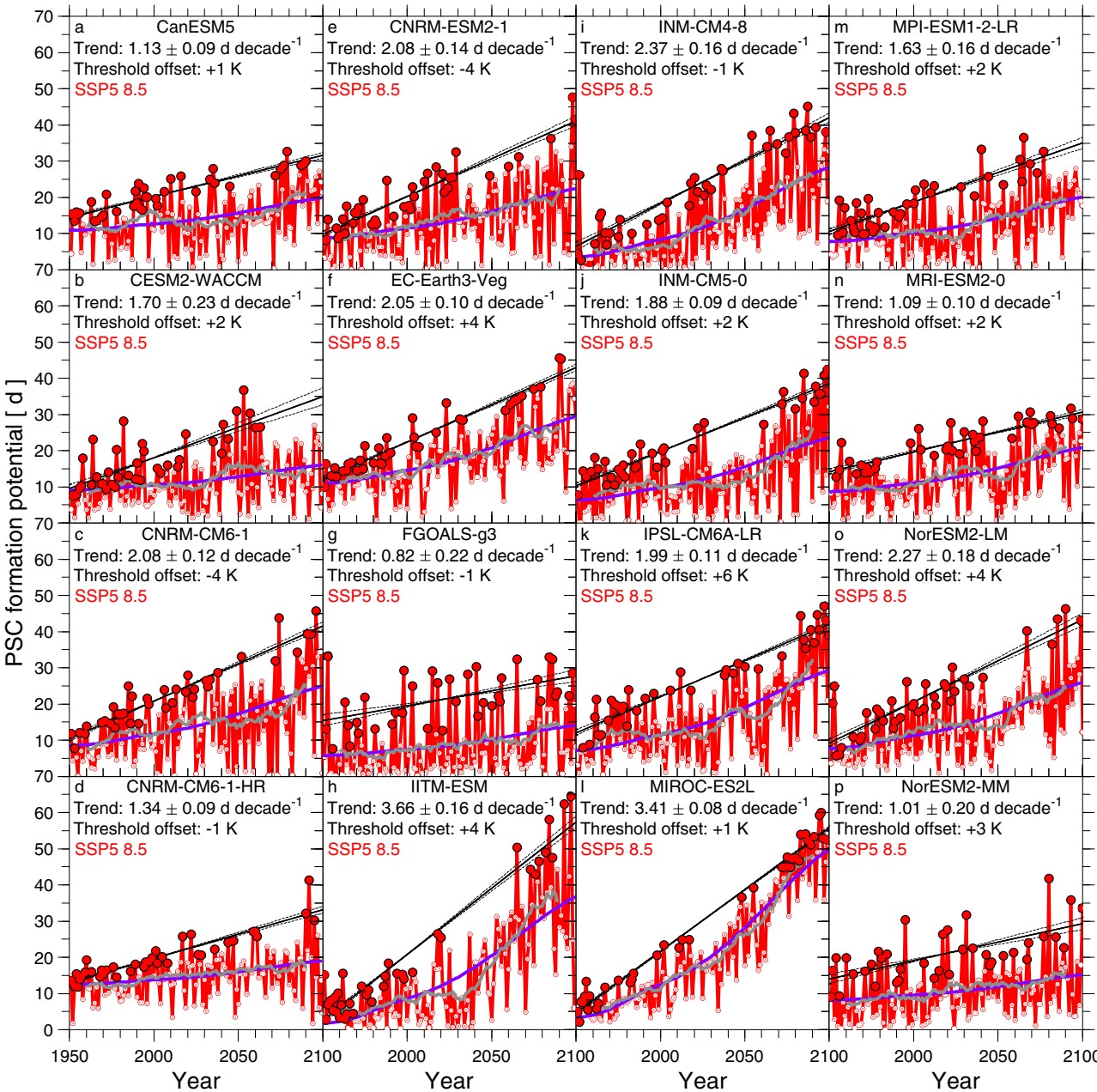

**Fig. 4 PFP, 1950–2100, from CMIP6 GCMs for SSP5-8.5 scenario and time-invariant H$_2$O. a–p** Time series of PSC formation potential (PFP) from 16 CMIP6 GCMs (as indicated on top of each panel), based on archived output from the SSP5-8.5 scenario (2015–2100) combined with output from the historical scenario (1950–2014). The solid circles indicate the coldest winters in the record (local maxima) selected using the ISA trend detection procedure (see "Methods"). A linear, least-squares fit (solid line) and 1σ uncertainty (dashed lines) to the solid red circles are shown in each panel, along with numerical values of the slopes ($S_{PFP-LM}$) and 1σ uncertainties of these fits. The blue line shows the best fit to PFP of the radiative forcing time series for each model run, and the grey line is a 21-year running mean (±10 years) to PFP from each GCM. The temperature threshold for the formation of PSCs has been offset by a constant number, specific to each model, so that the overall magnitude of PFP$^{LM}$ in the GCM matches the observed magnitude of PFP$^{LM}$, over the modern satellite era (see "Methods" and Table 1).

SSP1-2.6 (ref. [41]). A comparison of PFP for four of these GCMs is shown in Fig. 5. Results for the other 16 GCMs exhibit similar behaviour, as shown below using the multi-model ensemble mean projections. Nearly without exception, the ISA-based value of $S_{PFP-LM}$ over 1950–2100 for a particular GCM is largest for the SSP5-8.5 simulation and lowest (in many cases, near zero) for the SSP1-2.6 run. This finding provides further evidence that stratospheric cooling caused by the human release of GHGs is the primary driver of rising LM values of PFP within these GCMs.

The projections of PFP shown in Fig. 5 have been found assuming profiles for H$_2$O and HNO$_3$ appropriate for the contemporary atmosphere. However, future levels of stratospheric H$_2$O will likely rise due to increasing tropospheric CH$_4$ as well as the warming of the tropical tropopause[49,50]. Figure 2 shows estimates of polar, stratospheric H$_2$O for changes driven by the oxidation of CH$_4$ (Fig. 2b), warming of the tropical tropopause (Fig. 2c), and the combination of both effects (Fig. 2d). Our CH$_4$-based estimate is derived from the relation between CH$_4$ and H$_2$O in the contemporary Arctic stratosphere[51]

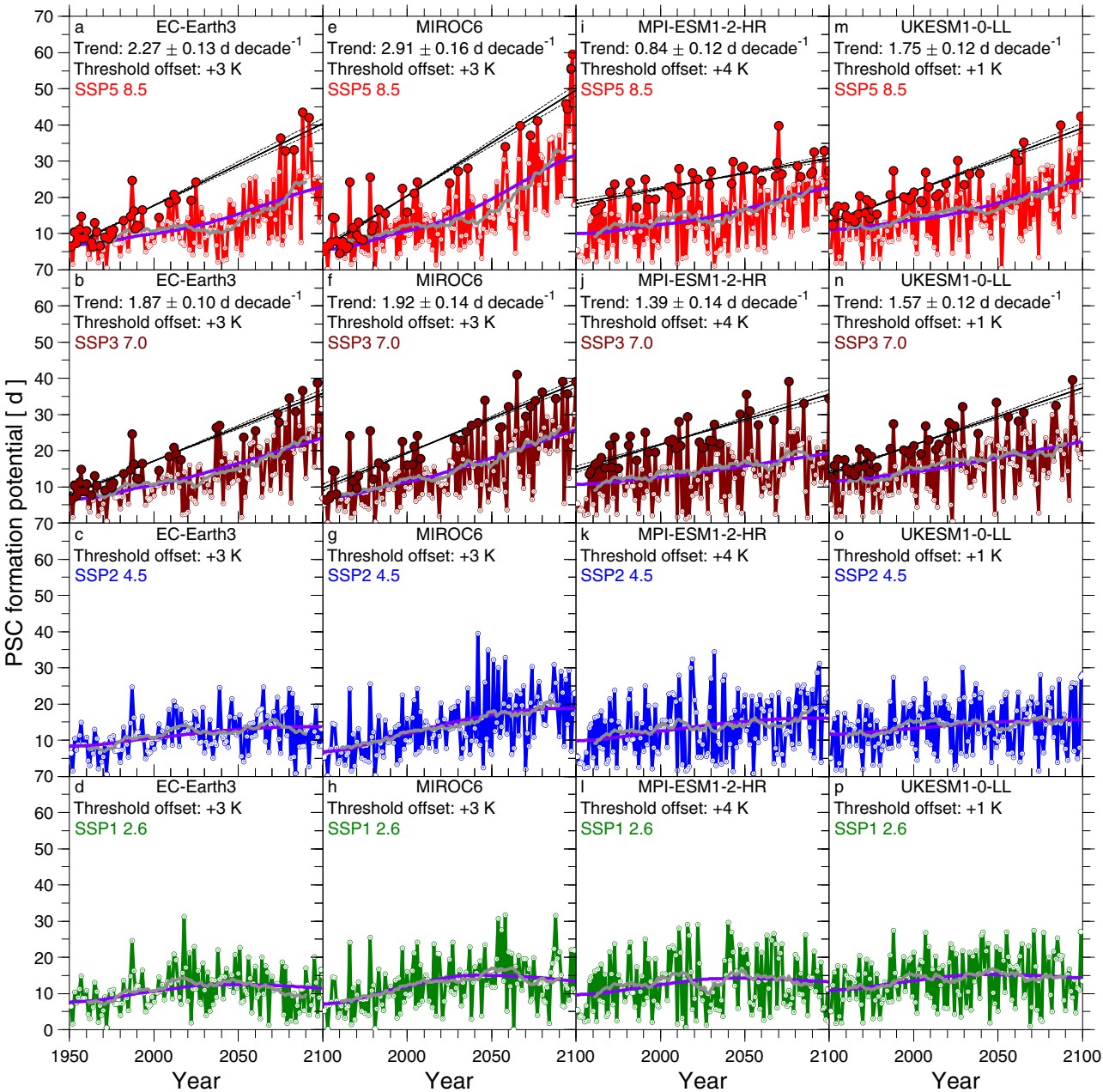

**Fig. 5 PFP, 1950–2100, from CMIP6 GCMs for various SSP scenarios and time-invariant H₂O. a–p** Time series of PSC formation potential (PFP) from 4 CMIP6 GCMs (as indicated on top of each panel), based on archived output from various historical (1950–2014) and SSP scenarios (2015–2100) for radiative forcing of climate. See Fig. 4 for more details.

combined with historical and future projections of CH₄ from the SSP-database, and the thermodynamic-based estimate results from an analysis of CMIP6 GCM output[43] (see "Methods").

Accounting for the future rise in stratospheric water for the computation of $T_{NAT}$ has a profound effect on PFP as well as $S_{PFP-LM}$. Figure 7 shows results from one of the four GCMs highlighted in Fig. 5. The first column of Fig. 7 shows the effect on PFP and $S_{PFP-LM}$ of projected future increases in stratospheric H₂O due to CH₄, the second shows the effect due to thermodynamics, and the third column shows the full effect of rising stratospheric H₂O. The sensitivity of future PFP to the projected change in H₂O is large within the EC-Earth3 GCM, as shown by comparing the first three columns of Fig. 7 (variable H₂O) to the first column of Fig. 5 (time-invariant H₂O), particularly for SSP5-8.5 and SSP3-7.0. The trend in $S_{PFP-LM}$

found using archived output from the EC-Earth3 GCM for SSP5-8.5 increases from $2.27 \pm 0.13$ d decade$^{-1}$ for time-invariant H₂O (Fig. 5a) to $3.93 \pm 0.13$ d decade$^{-1}$ when both of the factors driving the potential future rise in stratospheric H₂O are considered (Fig. 7i), because a more humid future stratosphere is more conducive to the chlorine activation and the formation of PSCs. Conversely, as expected, the impact of future stratospheric H₂O on PFP and $S_{PFP-LM}$ is small for SSP2-4.5 and SSP1-2.6. The other GCMs that have archived results for all four SSPs exhibit similar behaviour (see SI).

**Projections of conditions conducive to Arctic ozone loss.** As shown in Fig. 1b, measured and modelled values of the chemical loss of column ozone in the Arctic stratosphere are well described by

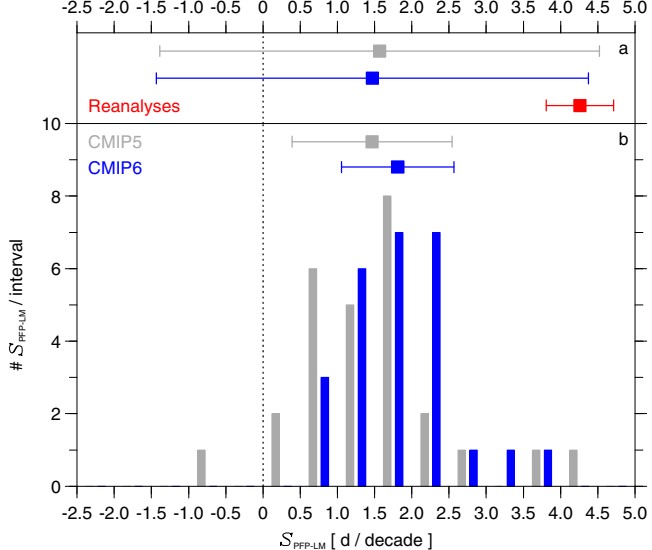

**Fig. 6 Modelled and measured values of $S_{PFP-LM}$.** **a** Mean and 1σ standard deviation of the slope of local maxima of PFP ($S_{PFP-LM}$) selected using the ISA trend detection procedure, for 1980–2020, based upon analysis of output from 26 CMIP6 GCM simulations (blue), 27 CMIP5 GCM runs (grey) (see "Methods") as well as reanalysis data from four meteorological centres (red) **b**, Mean and 1σ standard deviation of $S_{PFP-LM}$ selected using the ISA trend detection procedure, for 1950–2100, based upon analysis of output from 26 CMIP6 GCM simulations (blue points with error bars) and 27 CMIP5 GCM runs (grey points with error bars) as well as the frequency distribution of $S_{PFP-LM}$ from the individual CMIP6 simulations (blue vertical bars) and CMIP5 runs (grey vertical bars).

OLP. For the EC-Earth3 GCM constrained by GHGs abundances for SSP5-8.5 and SSP3-7.0, the largest values of OLP occur towards the latter half of this century, particularly when the full effect of rising stratospheric $H_2O$ is considered (Fig. 7m, n). This projection suggests stratospheric cooling combined with moister conditions, driven by future rises in the atmospheric abundance of anthropogenic GHGs, could prolong the conditions that lead to significant chemical loss of column $O_3$ within the Arctic vortex until late in this century. Conversely, if GHGs follow either the SSP2-4.5 or SSP1-2.6 scenario, the value of OLP is projected to decline from close to present time until the end of the century (Fig. 7o, p).

We now turn to the multi-model ensemble mean values of PFP, rather than the LM of PFP from a single GCM. Figure 8 shows the time series of ensemble-mean values of $\Delta O_3^{REG}$ and OLP from the 20 CMIP6 GCMs that have archived output for GHG abundances from SSP5-8.5, SSP3-7.0, SSP2-4.5, and SSP1-2.6, assuming constant stratospheric $H_2O$. Commonly, year 1980 is used as a benchmark for studies of polar ozone recovery[23]. For fixed $H_2O$, the multi-model mean value of OLP remains well above the 1980 level until the end of the century for SSP5-8.5 and SSP3-7.0, approaches the 1980 level for SSP2-4.5, and reaches the 1980 level at end of the century for SSP1-2.6. For SSP5-8.5 and SSP3-7.0, the seasonal loss of ozone (i.e., $\Delta O_3^{REG}$) in the range of 70–100 DU persists until the end of this century at an amount comparable to contemporary values.

Stratospheric humidity is expected to rise due to an increased source from the oxidation of $CH_4$ and a warmer tropical tropopause, particularly for climate scenarios with high RF of climate towards the end of the century, which will lead to further increases in $\Delta O_3^{REG}$ and OLP. Figure 9 shows ensemble mean values of $\Delta O_3^{REG}$ and OLP for the GCMs also represented in Fig. 8, allowing for variations in stratospheric $H_2O$ in addition to temperature. When the effect of rising $H_2O$ on the future

occurrence of PSCs is considered, $\Delta O_3^{REG}$ and OLP at end of the century are higher than contemporary values of these quantities for the SSP5-8.5 and SSP3-7.0 simulations. This analysis suggests that despite a projected decline in stratospheric halogen loading, the potential for significant chemical loss of Arctic column ozone could not only persist until the end of the century but might actually exceed contemporary loss if the atmospheric abundance of GHGs follows either SSP5-8.5 or SSP3-7.0 (Fig. 9a, b). The multi-model mean values of $\Delta O_3^{REG}$ and OLP at end of the century for SSP2-4.5 (Fig. 9c) also lie above the 1980 levels. Both quantities drop below the 1980 level for SSP1-2.6 (Fig. 9d), because the suppressed abundance of $CH_4$ towards the end of the century within this scenario leads to a decline in stratospheric $H_2O$ relative to today (Fig. 2d).

The multi-model ensemble values of $\Delta O_3^{REG}$ and OLP shown in Figs. 8 and 9 capture the general tendency of projections of stratospheric temperature within 20 GCMs, the result of an enormous computational effort by the climate modelling community. On the other hand, this averaging procedure masks the strong year to year variability in Arctic conditions conducive for major ozone depletion, as represented in Fig. 7m–p (for EC-Earth3) and in SI for other GCMs, and as noted by an analysis of a seven-member ensemble from the United Kingdom Chemistry and Aerosols (UM-UKCA) CCM[23].

## Discussion

There are a number of factors that affect the accuracy of lower stratospheric temperature within GCMs, such as the maximum altitude and vertical resolution[52] as well as model representation of planetary wave activity that transports energy from equatorial to poleward regions[53]. One important marker of the usefulness of a GCM to simulate stratospheric dynamics is whether the model generates an oscillation of the direction of the zonal wind in the tropical lower stratosphere with a period of about 28 months, known as the quasi-biennial oscillation (QBO)[53]. Our examination of the tropical zonal wind from the models suggests CMIP6 GCMs tend to provide a better representation of the QBO than was evident in CMIP5 GCMs (see "Methods"), consistent with the more formal analysis of Richter et al.[54]. We see little difference in our projections of column ozone loss for the Arctic stratosphere ($\Delta O_3^{REG}$) (Figs. 8 and 9) when the CMIP6 GCM output is examined in groups of models that provide a reasonable representation of the QBO versus other models (see "Methods"). Richter et al.[54] note that while the number of models with an internally generated QBO has increased substantially from CMIP5 to CMIP6, the multi-model mean amplitude for atmospheric levels below a pressure of 20 hPa is still much lower than observed. Given the importance of the QBO in stratospheric dynamics, substantial effort is being directed towards improving the representation of this process within GCMs[55].

Ideally, GCMs would include interactive chemistry, as there are numerous feedbacks and interactions between the photochemical processes that regulate stratospheric ozone and the dynamical and radiative drivers of PFP. Four of the 20 CMIP6 GCMs considered above have fully interactive chemistry: the other 16 models use prescribed fields of ozone. The temporal evolution of OLP found using results from the four GCMs with interactive chemistry is about 20–25% lower at end of century than that found for the other 16 GCMs; nonetheless, $\Delta O_3^{REG}$ remains close to the contemporary value until the end of century for the SSP3-7.0 and SSP5-8.5 simulations conducted using these interactive GCMs (see "Methods").

Finally, CCMs that have been used to assess the evolution of Arctic ozone have interactive chemistry with vertically resolved stratospheres and better spatial resolution than most of the

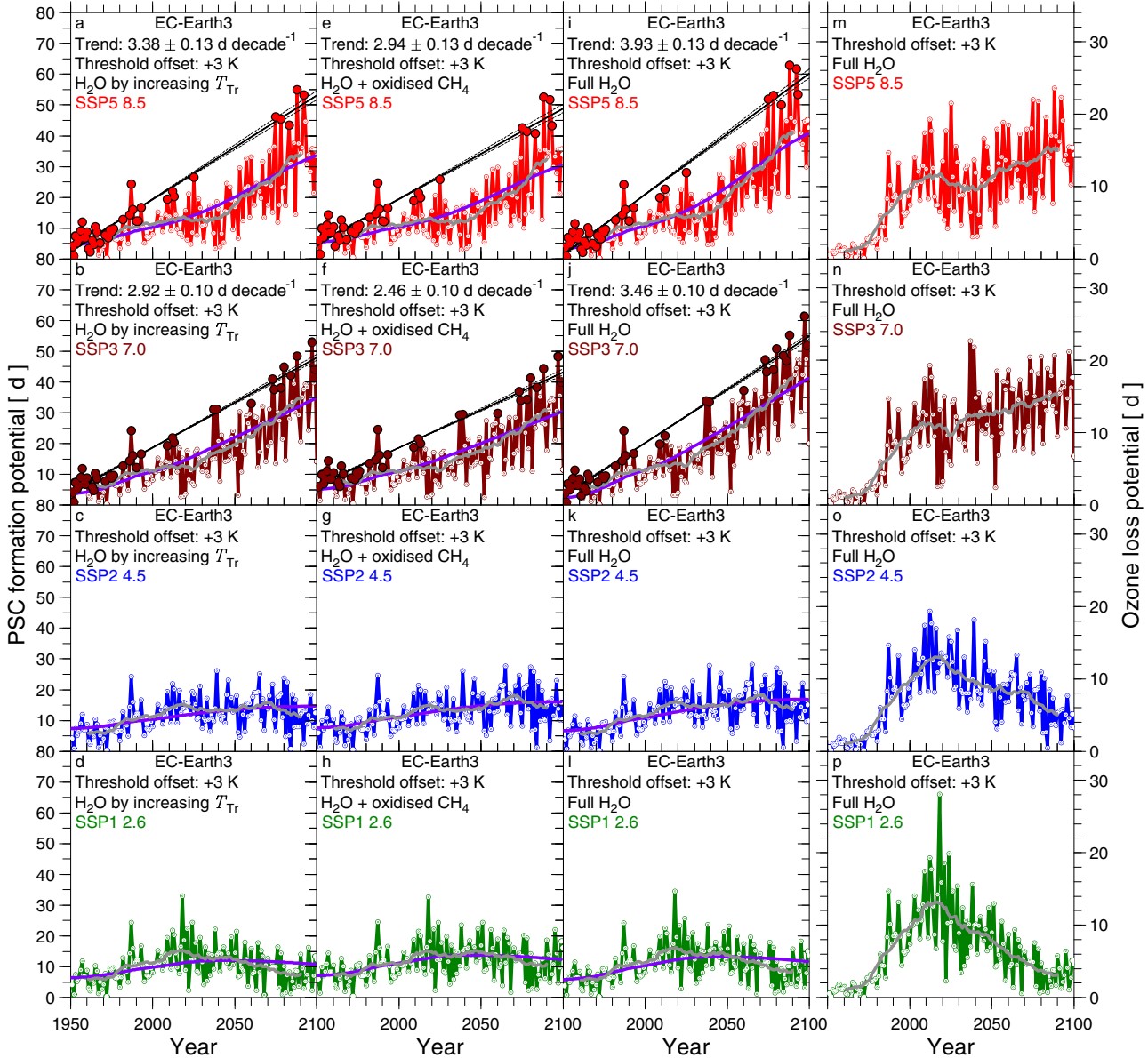

**Fig. 7 PSC formation potential (PFP) and Ozone Loss Potential (OLP), 1950–2100, from EC-Earth3 model for variable H₂O, various SSP scenarios. a–l** Same as Fig. 5 for the EC-Earth3 GCM, for variable H₂O accounting for: tropical tropopause warming (**a–d**), changes in atmospheric CH₄ (**e–h**), and both effects (**i–l**). **m–p** OLP from the EC-Earth3 GCM, for variable H₂O due to both tropopause warming and CH₄ oxidation. The grey line shows a 21-year running mean (±10 years) to OLP from each simulation, conducted for various SSPs. Figures showing results for the other GCMs that appear in Fig. 5 are included in the SI.

CMIP6 GCMs[56]. These CCMs tend to exhibit a more realistic representation of planetary wave activity and are capable of representing the impact of the intensification of the Brewer Dobson Circulation (BDC) and upper stratospheric cooling on ozone, two factors that result in the projection of future increases in Arctic column ozone during winter and spring[56]. However, the multi-model mean of CCMs used to project the future evolution of Arctic column ozone significantly underestimates prior observed ozone depletion, particularly during cold winters with extensive PSC activity[56].

Values of $\Delta O_3^{REG}$ shown in Figs. 8 and 9 represent the seasonal loss of column ozone that may occur for various GHG scenarios, rather than resulting column ozone. Future levels of Arctic column ozone during late winter and early spring are expected to increase due to factors such as intensification of the BDC, upper stratospheric cooling, as well as possible changes in

planetary and gravity wave activity that exert a strong influence on the abundance of column ozone within the Arctic vortex during its formation in early winter and dynamically induced increases during winter[22,23,56]. Langematz et al.[22] project maximum $V_{PSC}$ to occur around 2060 with a subsequent decline due to enhanced dynamical warming of the Arctic vortex in February and March, based on simulations conducted with their CCM. Finally, future levels of N₂O are expected to rise[41], leading to higher levels of HNO₃ that will lead to more favourable conditions for the formation and existence of PSCs[9]. Future total column ozone during spring will reflect a balance between the initial abundance, dynamical transport, and chemical loss that is driven by a large number of factors.

The strong dependence of the ensemble mean value of OLP towards the end of the century on radiative forcing of climate suggests that large, seasonal loss of column ozone in the Arctic

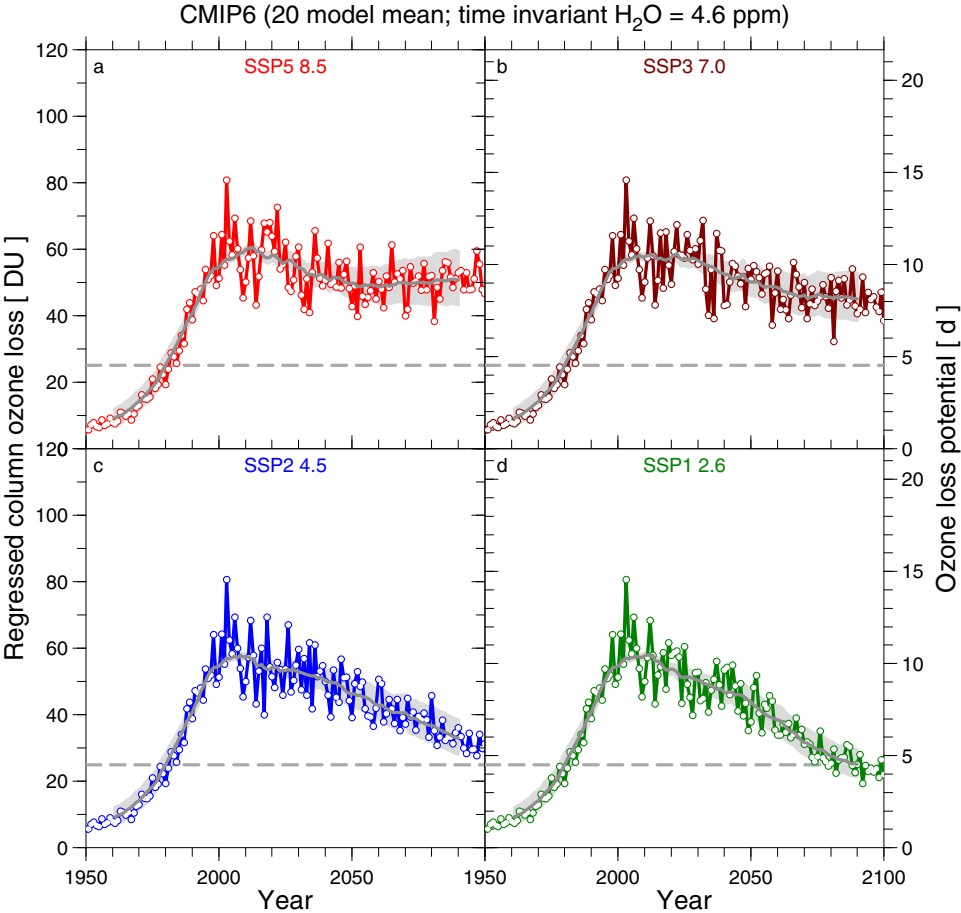

**Fig. 8 Ensemble model mean regressed column ozone loss and Ozone Loss Potential (OLP), time-invariant H₂O.** The value of OLP (right ordinate) and $\Delta O_3^{REG}$ computed from OLP (left ordinate) from the 20 CMIP6 GCMs (CanESM5, CESM2-WACCM, CNRM-CM6-1, CNRM-CM6-1-HR, CNRM-ESM2-1, EC-Earth3, EC-Earth3-Veg, FGOALS-g3, IITM-ESM, INM-CM4-8, INM-CM5-0, IPSL-CM6A-LR, MIROC6, MIROC-ES2L, MPI-ESM1-2-HR, MPI-ESM1-2-LR, MRI-ESM2-0, NorESM2-LM, NorESM2-MM, UKESM1-0-LL) that archived results for the SSP5-8.5 (**a**), SSP3-7.0 (**b**), SSP2-4.5 (**c**), and SSP1-2.6 (**d**) scenarios, computed assuming a constant volume mixing ratio for stratospheric H₂O of 4.6 ppmv. The same temperature threshold offsets specified in Table 1 and Figs. 4 and 5 have been used. The grey solid line shows a 21-year running mean (±10 years) to the ensemble mean of $\Delta O_3^{REG}$ for each SSP, the grey shaded area represents a 21-year running mean of the range in $\Delta O_3^{REG}$ for exponents of 1 (upper boundary) and 1.4 (lower boundary) of the expression for OLP, and the grey dashed horizontal lines denoted the 1980 value of $\Delta O_3^{REG}$. The right-hand ordinate shows the scale of the multi-model mean values of OLP, which are the initial quantities computed from the GCM output. Note, this right-hand ordinate does not correspond to the grey shaded area, since an exponent different from 1.2 was used.

could persist for much longer than is commonly appreciated[56]. If stratospheric H₂O rises as projected in Fig. 2d and GHGs follow a trajectory similar to either SSP5-8.5 or SSP3-7.0, chemical loss of Arctic ozone could even be larger by end of the century than has occurred in the past. Consequently, anthropogenic climate change has the potential to partially counteract the positive effects of the Montreal Protocol in protecting the Arctic ozone layer.

## Methods

**Computation of PFP.** The temperature at which nitric acid trihydrate (NAT) becomes thermodynamically stable, $T_{NAT}$, is governed by the vapour pressure of nitric acid (HNO₃) and water (H₂O)[9]. Here, we use a constant volume mixing ratio of stratospheric H₂O equal to 4.6 parts per million (ppmv) and a profile of HNO₃, both based on satellite observations, to find $T_{NAT}$. We compute $T_{NAT}$ using the saturation vapour pressure of H₂O and HNO₃ over NAT measured by Hanson and Mauersberger[9]. A volume mixing ratio for H₂O of 4.6 parts per million (ppmv) is used at all pressure levels, consistent with observations reported by the U.S. National Aeronautics and Space Administration Microwave Limb Sounder instrument for the lower stratosphere of the Arctic[57], as input to the calculation of $T_{NAT}$. The specified mixing ratio profile of HNO₃, which varies as a function of pressure, is based on measurements acquired in the Arctic during January 1979 by the Limb Infrared Monitor of the Stratosphere (LIMS) on board Nimbus 7 (ref. [58]).

The quantity $V_{PSC}$ represents the volume of air for which temperature is less than $T_{NAT}$, evaluated between potential temperatures of 400 and 700 K. The formation of

PSCs in the Arctic stratosphere also depends on factors such as cooling rate, the degree of super-saturation, the chemical composition of pre-existing nuclei, as well as the surface coating of condensed particles[10–12,59]. During cold Arctic winters, the profile of HNO₃ will be altered by the sedimentation of nitrate-bearing PSCs, termed denitrification[11,12,59,60]. Nonetheless, our approach captures the primary factor that drives the chemical loss of Arctic O₃: that is, temperatures low enough to allow for the existence of PSCs. As described in the main paper and detailed below, we arrive at remarkably similar conclusions based upon consideration of the temperature at which chlorine is activated on aerosols[6,32], rather than $T_{NAT}$, because these two temperature thresholds are similar.

Our analysis requires definition of the area and volume of the Arctic polar vortex, denoted $A_{VORTEX}$ and $V_{VORTEX}$. The horizontal boundary of the vortex is based on the value of $36\,s^{-1}$ for normalized potential vorticity (nPV), which is found from the horizontal wind and temperature fields and then scaled to account for the steep altitude dependence of PV. The value of $36\,s^{-1}$ for normalized PV (nPV) is used to define the edge of the polar vortex, as described in section 3.3 of Rex et al.[26]. Other studies utilize the maximum gradient in PV to define the boundary of the polar vortex[61]. We use nPV = $36\,s^{-1}$ to define the vortex boundary because on some days the gradient method introduces a level of complexity, due to the existence of multiple maximum gradients of nearly equal magnitude separated by a considerable distance, which requires human judgement.

We have examined maps of nPV and temperature plotted for 1 February of the years 1960–2100, in increments of every 10 years, for all 26 CMIP6 GCMs that archived results for SSP5-8.5. These maps show that the nPV = $36\,s^{-1}$ boundary for the Arctic vortex is not greatly affected by climate change until the end of the century; maps for the four CMIP6 GCMs highlighted in Fig. 5 of the paper are

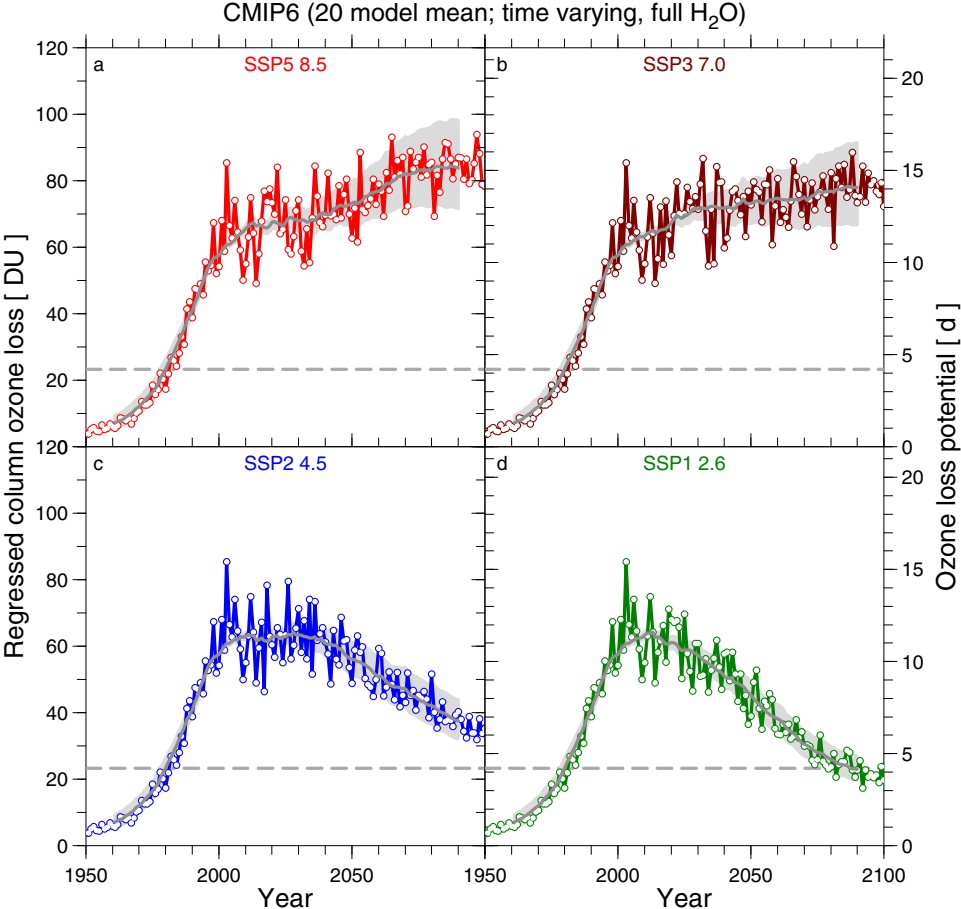

**Fig. 9 Ensemble mean regressed column ozone loss and Ozone Loss Potential (OLP), variable H$_2$O.** Same as Fig. 8, except OLP from the archived GCM output of each GCM has been computed using the time series for polar stratospheric H$_2$O shown in Fig. 2d, which accounts for increasing stratospheric humidity due to both variable CH$_4$ and warming of the tropical tropopause. **a** SSP5-8.5, **b** SSP3-7.0, **c** SSP2-4.5, and **d** SSP1-2.6 scenarios.

shown in Supplementary Fig. 1. Since PV from four reanalyses that span many decades and model output from 53 GCM simulations that span more than a century and a half are examined, it is preferable to implement a method that requires no human intervention.

The next step for the computation of PFP involves calculation of the area over which temperature is below the threshold for the existence of PSCs, $A_{PSC}$, as well as $A_{VORTEX}$. The area for which $T < T_{NAT}$ and the area enclosed by the nPV $= 36\,s^{-1}$ contour are found on various potential temperature ($\theta$) surfaces for each time step of the analysis, which are evaluated to yield $A_{PSC}(\theta, t)$ and $A_{VORTEX}(\theta, t)$. Next, $V_{PSC}(t)$ and $V_{VORTEX}(t)$ are computed for each time step by evaluating:

$$V_{PSC}(t) = \int_{400\,K}^{700\,K} c(\theta) A_{PSC}(\theta, t)\, dt \quad (2)$$

$$V_{VORTEX}(t) = \int_{400\,K}^{700\,K} c(\theta) A_{VORTEX}(\theta, t)\, dt \quad (3)$$

where $c(\theta)$ is a factor that converts intervals of potential temperature to geometric altitude (numerical values provided in a data repository). The next step in the calculation of PFP involves evaluating the integral of the ratio of $V_{PSC}(t)$ and $V_{VORTEX}(t)$ over the Arctic ozone loss season of each winter, which are combined to yield:

$$PFP(yr) = \int_{1\,Nov}^{30\,Apr} \frac{V_{PSC}(t)}{V_{VORTEX}(t)}\, dt \quad (4)$$

1 November (prior year) and 30 April (specified year) are used as limits of integration because these dates encompass the time period of possible PSC activity among reanalysis and GCM-based temperature fields.

A grid for $\theta$ from 400 to 700 K, in 5 K increments, is used for the computation of $V_{PSC}$ from each reanalysis data set, all of which are provided at 6 h time steps. At each time step the value of the ratio $V_{PSC}/V_{VORTEX}$ is capped at unity, because in rare instances the volume for PSC temperatures is larger than the volume of the vortex defined using the 36 s$^{-1}$ boundary. The GCM output is generally available on a daily basis, although some modelling groups have archived output every 6 h; details are provided in Supplementary Table 1. The models that archive output

every 6 h provide high model vertical resolution fields on the native model grid, whereas the daily output is generally provided for only a limited number of pressure levels (i.e., 100, 50, and 10 hPa). In cases where the output for the SSP1-2.6, SSP2-4.5, and SSP3-7.0 scenarios are available only in low resolution (daily), we use low resolution for the SSP5-8.5 scenario from the corresponding GCM run, even if a higher resolution is available for SSP5-8.5.

Values of $V_{PSC}(t)$ and $V_{VORTEX}(t)$ found using Eqs. (2) and (3) as well as the ratio of these terms are shown in Supplementary Fig. 2. The unusual behaviour of Arctic winter 2020, such as record high values for $V_{PSC}$ in March and $V_{VORTEX}$ in March and April, is readily apparent. $V_{PSC}(t)$ and $V_{VORTEX}(t)$ are used in Eq. (4) to determine PFP. All reanalyses and GCM fields are analysed on the native horizontal resolution of the product. Finally, the 1σ uncertainty of PFP shown in Fig. 1 is based on perturbation of the reanalysis temperature field by ±1 K; this magnitude of the offset is based on our analysis of the approximate 1σ standard deviation about the mean of stratospheric temperature from the four data centres, over the modern satellite era.

In the main article, we estimate PFP using the JRA-55 and ERA5/ERA5.1/ERA5 BE (preliminary version) reanalysis products over 1965–2020, as well as 1980–2020. Meteorological data in the Arctic stratosphere acquired prior to 1979 mainly rely on radiosonde measurements, and 1965 marked the beginning of regular radiosonde coverage of the Arctic stratosphere. Luers and Eskridge[62] quantified the bias in temperature reported by ten of the most common radiosondes used throughout the world since 1960, for use in climate studies. The JRA-55 reanalysis makes use of the Radiosonde Observation Correction using Reanalysis (RAOBCORE) version 1.4 (ref. [63]) bias correction procedure for radiosonde temperature until the end of 2006, and RAOBCORE version 1.5 (ref. [64]) thereafter. As an important check on the temporal integrity of the reanalyses prior to 1979, in Supplementary Fig. 3 we show an update to the radiosonde temperature time series acquired at Sodankylä, Finland, for each winter since 1965 (ref. [65]). This figure shows the time evolution of the percentage of observations of temperature < −77.9 °C at 50 hPa over the months of December (prior year) and January, February, and March (indicated year) from regular radiosonde launches from Sodankylä. Supplementary Fig. 3 supports our conclusion, shown in Fig. 3d of the main article, that conditions conducive for the existence of PSCs tended to be less common between 1965 and 1979, compared to the past few decades.

In the main article, we discuss an application of a threshold for the existence temperature of PSCs applied to output from the CMIP5 and CMIP6 GCMs, such that the magnitude of the LM in PFP matches the observed magnitude over the modern satellite record. Details of the specific GCMs[66–107] are given in the Supplement. We compute PFP from these GCMs in a similar way to that applied to the computation of PFP from meteorological data, except for the application of a temperature offset to account for either warm or cold bias. The offsets for $T_{NAT}$ used for CMIP6 GCMs are given in Table 1. These offsets have been determined based on the criterion that a trend line fit to the LM of PFP (PFP$^{LM}$) from the GCM over 1980–2020 using the ISA selection procedure (described below), should have a value in year 2000 (mid-point of the data record) that lies closest to the value of the fit to PFP$^{LM}$ data from ERA5/ERA5.1 in year 2000, among all possible 1 K incremental offsets to $T_{NAT}$ (including no offset) ranging from −9 to +9 K. For CMIP6, 19 of the 26 GCMs required a positive temperature offset for the PSC threshold (Table 1), indicating temperature conditions computed within these GCMs tend to be warmer than climatology, particularly for winters with cold, isolated Arctic vortices.

Supplementary Fig. 4 shows comparisons of PFP for each CMIP6 GCM, with and without application of this threshold. Supplementary Table 2 is similar to Table 1 of the main article, except values and statistical analysis of value $S_{PFP-LM}$ and $\Delta S_{PFP-LM}$ are shown without application of any adjustment for the PSC temperature threshold. It is evident from Supplementary Table 2 and Supplementary Fig. 4 that the main thesis of our study, the coldest winters in the Arctic stratosphere are getting colder due to rising GHGs, is apparent in GCM simulations with and without this adjustment. We have chosen to show estimates of $S_{PFP-LM}$ upon application of a threshold correction in the main article because this is a more realistic metric to examine within the models, particularly those GCMs that have very warm biases and thus exhibit unrealistically small values of PFP.

**Trend detection procedures**. We utilise several procedures to assess the trend in LM of PFP. First, we describe the ISA, which we apply to the 41-year time series from ERA5/ERA5.1. Following the computation of PFP for all Arctic winters, all of the data are fit using a linear least-squares regression line (Supplementary Fig. 5a). We then compute the vertical distance (i.e., the difference in PFP) between the fit line and each data point. The point (in blue) with the largest distance below the line, the most warm winter relative to the current trend line, is omitted from the subsequent analysis. The remaining data points are then fit with another linear least squares regression line (Supplementary Fig. 5b). The same procedure of finding and removing the point (blue) with the greatest distance below the fit line is repeated leading to Supplementary Fig. 5c. The procedure is repeated until one-quarter of the points (termed the upper quartile relative to the trend line) remain; Supplementary Fig. 5d–f shows results of iterations number 28, 29, and 30. The slope ($S_{PFP-LM}$) of 4.50 d decade$^{-1}$ and 1σ uncertainty ($\Delta S_{PFP-LM}$) of 0.19 d decade$^{-1}$ given for the least-squares fit of data shown on Supplementary Fig. 5f are the same as that shown in Fig. 3a.

Next, we describe the Maximum in Interval Method (MIM) for assessing trends in PFP. Rex et al.[13] applied this selection procedure to their analysis of $V_{PSC}$. They quantified the slope in the maximum values of $V_{PSC}$ that had occurred over successive 5-year long, independent time intervals. Their analysis considered 37 years of data spanning winters of 1966–2003, from which eight values of $V_{PSC}$ were selected. Supplementary Fig. 6b shows the resulting selections of LM (red solid points), which yields values of $S_{PFP-LM}$ and $\Delta S_{PFP-LM}$ of 4.24 ± 0.34 d decade$^{-1}$ for the trend in LM from the ERA5/ERA5.1 time series of PFP. Clearly, the results are quite similar to the value of $S_{PFP-LM}$ found using the ISA procedure, even though some of the data points selected as LM by these two techniques differ (Supplementary Fig. 6a and b). Our development of the ISA selection procedure, rather than MIM, was also driven by our analysis of GCM output that shows steadily rising values of PFP until the end of this century when models are driven by either RCP 8.5 or SSP5-8.5 GHG scenarios, for which some models are interspersed with gaps >5 years for LM in PFP. The time interval of the MIM procedure could have been altered, but rather we offer the ISA procedure as a more robust method for the selection of LM of PFP.

Supplementary Fig. 6c illustrates the value above sigma (VAS) selection procedure used by Rieder and Polvani[108] to address trends in $V_{PSC}$. For VAS, one first computes the mean and standard deviation about the mean (σ) using all values of the PFP time series. Next, the slope in PFP is found using only those data points that lie 1σ above the mean. The VAS selection yields seven selected points, resulting in a slope of 3.06 ± 1.51 d decade$^{-1}$ for a fit to these selected points. The selection of PFP from Arctic winter 2018 and lack of selection of any data points prior to 1995 by the VAS selection procedure illustrates the problem with this method: by design, only the highest values are selected. To test the hypothesis that the LM of a quantity has risen over time, one should apply a time-varying statistical method for the selection of points. A static selection such as VAS is not an appropriate means to assess whether the coldest winters are getting colder because VAS tends to select only the highest values, rather than the LM, from the time series of PFP.

In order to further assess the selection of PFP$^{LM}$ by the ISA, MIM, and VAS trend detection procedures a set of MC simulations was conducted for a dataset with an imposed, positive trend in PFP. For this set of MC simulations, one million time series of PFP were generated for a 41-year long record (matching the time period 1980–2020), each with PFP distributed between a lower bound of 0 and an upper bound that starts at 13.6 d (first winter) and rises with a slope of 4.59 d decade$^{-1}$.

Each PFP data point is uniformly, randomly distributed between the time-varying upper bound and the lower bounds, chosen to match the lower and upper limits of PFP from ERA5 in a statistical fashion.

Supplementary Table 3 summarizes the results of this first set of MC simulations. This table gives the mean value of the slope ($\overline{S_{PFP-LM}}$) and 1σ uncertainty ($\overline{\Delta S_{PFP-LM}}$) of the fits to the maxima in PFP of these one million randomly generated time series. The table also provides the mean number ($\overline{k}$) and minimum number ($k_{MIN}$) of LM points from which the slopes and uncertainties are computed. Use of the ISA approach yields a value for $\overline{S_{PFP-LM}}$ of 4.50 d decade$^{-1}$ upon selection of the upper quartile of LM points relative to the trend line. The fact this value of $\overline{S_{PFP-LM}}$ lies within 2% of the slope of the design value of the upper bound attests to the robust accuracy of the ISA approach. The MIM selection procedure with 5-year intervals (the last interval covers 6 years) results in the selection of eight points from which $S_{PFP-LM}$ is computed, for each of the million cases. The MIM approach results in a value for $\overline{S_{PFP-LM}}$ of 3.95 d decade$^{-1}$, which is 14% lower than the upper bound of the experimental design. Numerous values of $S_{PFP-LM}$ from the MIM ensemble are greater than the upper bound design value of 4.59 d decade$^{-1}$. Nonetheless, on average, the MIM approach tends to underestimate the true value of the prescribed upper bound of the experimental design, due to gaps in the true LM of PFP that sometimes exceed five years. Finally, for the VAS approach, the number of selected points can often be low, which is reflected in the value of $\overline{k}$ given for the VAS entries in Supplementary Table 3. Therefore, we have imposed criteria that VAS must select either a minimum of three, five, or seven points from each of the million artificial time series. For a final test of VAS, we have imposed a requirement that ten points (that is, the ten largest values of PFP) must be used for the computation of $S_{PFP-LM}$ for each time series. Values of $\overline{S_{PFP-LM}}$ returned by VAS range from 2.08 to 2.30 d decade$^{-1}$, a factor of two less than the upper bound of the experimental design, because as noted above the VAS procedure selects the highest values rather than LM. As such, the ISA selection procedure provides a more accurate representation of the design of the underlying model than the MIM approach and a much more accurate representation than that provided by the VAS selection procedure.

**Statistical significance**. The fitting uncertainty ($\Delta S_{PFP-LM}$) in the regression lines for PFP$^{LM}$ is not a true measure of the significance of the trend ($S_{PFP-LM}$), because $\Delta S_{PFP-LM}$ does not consider the selection process for obtaining LM of PFP. Therefore, we assess the statistical significance of $S_{PFP-LM}$ and $\Delta S_{PFP-LM}$ using another set of MC simulations. In these MC simulations, we work with actual data for PFP from either a reanalysis or GCM to assure the basis set of our randomly generated time series are identical to the PFP time series. The time series for PFP shown in Fig. 3a consists of 41 data points, which could be arranged in more than $3 \times 10^{49}$ possible combinations. We use a random number generator to place these 41 PFP data points into 10 million combinations.

The ISA selection algorithm is applied to each of the 10 million combinations of PFP, resulting in a selection of the upper quartile (that is, 10 and usually 38 points for the reanalyses and GCMs, respectively) relative to the trend line, following the same algorithm used to select the PFP$^{LM}$ in the main article. The corresponding slope ($S_{PFP-LM}$) and uncertainty ($\Delta S_{PFP-LM}$) is found for each of these combinations. The p-values given in Table 1 for $S_{PFP-LM}$ are equal to the probability that the slope of these random fits exceeds the slope determined from the data. In other words, 18% of the randomly generated combinations of PFP for the ERA5/ERA5.1 basis set (over the 1980–2020 time period) yield a value for $S_{PFP-LM}$ larger than 4.50 d decade$^{-1}$. However, for the vast majority of the time series that yields a value of $S_{PFP-LM}$ larger than 4.50 d decade$^{-1}$, the value of $\Delta S_{PFP-LM}$ associated with the fit is larger than the ±0.19 d decade$^{-1}$ uncertainty found from the ERA5/ERA5.1 time series. High slopes with large uncertainty are usually dominated either by several low values of PFP$^{LM}$ at the start of the time series of the selected points or a couple of high values of PFP$^{LM}$ towards the end of the time series. As explained below, very few of the randomly generated time-series yield a high value of $S_{PFP-LM}$ in combination with a low value of $\Delta S_{PFP-LM}$.

We therefore examine the quantity $S_{PFP-LM}/\Delta S_{PFP-LM}$ as a measure of the statistical significance of both the temporal rise in PFP$^{LM}$ as well as the uncertainty in this rise. Of the randomly generated time series, 99.992% yield a value of $S_{PFP-LM}/\Delta S_{PFP-LM}$ that is smaller than the actual value of 23.6 (4.50 d decade$^{-1}$ divided by 0.19 d decade$^{-1}$). Consequently, a p-value of $8 \times 10^{-5}$ is associated with the entry for $S_{PFP-LM}/\Delta S_{PFP-LM}$ based upon ERA5/ERA5.1 data in Table 1 and we state, in the main article, that the value of $S_{PFP-LM}$ and the associated uncertainty are statistically significant at better than the 2σ confidence level. While the shape of the probability density functions of $S_{PFP-LM}$ and $S_{PFP-LM}/\Delta S_{PFP-LM}$ are not strictly Gaussian, the fall-offs of the tail of both functions are Gaussian-like (i.e., kurtosis close to 3; more specifically, the kurtosis for $S_{PFP-LM}$ is 2.1 and for $S_{PFP-LM}/\Delta S_{PFP-LM}$ is 3.4). Therefore, we are comfortable assessing better than 2σ confidence to $S_{PFP-LM}/\Delta S_{PFP-LM}$ since $8 \times 10^{-5}$ is so much less than 0.05, the 2σ confidence marker for a strictly Gaussian distribution. We have similarly estimated the statistical likelihood of achieving the reported values of $S_{PFP-LM}$ and $S_{PFP-LM}/\Delta S_{PFP-LM}$ from the 150-year time series of PFP from each CMIP6 GCM simulation constrained by SSP5-8.5, again using 10 million of the possible combinations of PFP from each basis set. The vast majority of the resulting p-values indicate statistical significance at close to or better than the 2σ level of confidence for both GCM-based values of $S_{PFP-LM}$ as well as $S_{PFP-LM}/\Delta S_{PFP-LM}$ (Table 1).

**Vortex boundary**. The vortex boundary used throughout our study is based on the value of $36\,s^{-1}$ for nPV. This definition of the vortex boundary is commonly used in other studies of Arctic ozone, because nPV $= 36\,s^{-1}$ tends to be closely associated with the maximum, horizontal gradient of potential vorticity[20,26,109]. To check whether other definitions of the vortex boundary would yield insignificant results Supplementary Fig. 7 shows trends in $S_{PFP-LM}$ found by the ISA algorithm applied to data from ERA5/ERA5.1 combined with ERA5 BE (preliminary version) from 1965 to 2020 for four alternate definitions of the vortex boundary, along with the resulting trends and p-values for the quantity $S_{PFP-LM}/\Delta S_{PFP-LM}$. For each alternate vortex boundary definition, the resulting trends in $S_{PFP-LM}$ are positive and highly statistically significant. The numerical values for PFP do vary based on how the boundary is specified and differ from those shown in Fig. 3b of the main paper, due largely to the use of the volume of the Arctic vortex in the denominator of the definition of PFP (Eq. 4).

**SSU & TOVS versus AMUS & ATOVS**. In the paper, we state that similar results are obtained for trends in PFP (differences within respective uncertainties) when considering temperature from the SSU and TOVS space-borne systems versus AMSU and ATOVS systems. The transition occurred in the years 1998–1999 (ref. [40]). Supplementary Fig. 8 shows that similar results for trends in $PFP^{LM}$ (differences within respective uncertainties) are found when considering data obtained only prior and only after this transition.

**Aerosol reactivity potential**. The main article states: we arrive at remarkably similar conclusions based upon consideration of the temperature at which chlorine is activated on aerosols[6,32], rather than $T_{NAT}$, because these two temperature thresholds are so similar. The term aerosol reactivity potential (ARP) is similar to PFP, except in Eq. (1) the quantity $T_{NAT}$ is replaced by $T_{ACL}$, which represents the temperature at which chlorine is activated. Values of $T_{ACL}$ are computed as a function of $H_2O$ and sulphate surface area density at 210 K using Eq. (1) and information in the caption of Fig. 5 of Drdla and Müller[6]. We use potential temperature as the vertical coordinate and the values of coefficients given in Table 1 to find $T_{ACL}$. The entire analysis is then repeated (i.e., analogues of $A_{PSC}$ and $V_{PSC}$, termed $A_{ACL}$ and $V_{ACL}$, are computed and used as in Eq. (1)), resulting in the term ARP being computed using an Eq. (2) with $V_{ACL}$ rather than $V_{PSC}$.

Supplementary Fig. 9 shows measured and modelled $\Delta O_3$ as a function of ARP (panel a) and OLP found using ARP rather than PFP (panel b). Supplementary Figure 9 shows trends in ARP from the four reanalysis data centres used in Fig. 3. The numerical values for the slope of the LM of ARP ($S_{ARP-LM}$) differ by only a small amount (typically 10%) compared to those given for $S_{PFP-LM}$ in the main article. Finally, Supplementary Fig. 11 shows the time series of ARP and the local maximum in ARP selected using ISA, for the 4 GCMs highlighted in Fig. 5. The results shown in Supplementary Figs. 9 and 10 are quite similar to those shown in Figs. 1 and 5 of the main article because $T_{NAT}$ is so similar to $T_{ACL}$. In the actual Arctic stratosphere, denitrification (the removal of $HNO_3$ by the physical sedimentation of PSCs) will prolong ozone loss[60] and alter $T_{NAT}$ due to suppression of gas-phase $HNO_3$ (ref. [57]). However, the volume of air for which chlorine is activated by heterogeneous chemistry is governed most strongly by temperature. The close visual relation between Figs. 1, 3 and 5 and Supplementary Figs. 9, 10, and 11 support the validity of the definition of OLP used in the main paper, which does not explicitly represent denitrification for the computation of $T_{NAT}$.

**Stratospheric $H_2O$**. Figure 2 contains our projections of stratospheric $H_2O$ accounting for contributions from the oxidation of $CH_4$ (Fig. 2b), warming of the tropical tropopause (Fig. 2c), and the sum of both forcings (Fig. 2d). The effect of oxidation of $CH_4$ on stratospheric $H_2O$ is based upon analysis of satellite observations of $CH_4$ obtained by the HALOE instrument in the Arctic polar vortex, as shown in Figure 12 of Müller et al.[51] for April 1993. In the Arctic stratosphere, between about 450 and 600 K potential temperature, the HALOE measurement of $CH_4$ exhibits a near-constant (with respect to altitude) value of ~0.5 ppmv. The age of air in the Arctic, lower stratosphere (i.e., the mean transit time from the tropical tropopause to the polar, lower stratosphere) tends to be about 6 years[30]. Hence, the appropriate comparison for surface conditions is the global mean abundance of $CH_4$ in January 1987, which was 1.639 ppmv according to https://www.esrl.noaa.gov/gmd/ccgg/trends_ch4. Consequently, we infer that about 70% of the available $CH_4$ (at the time this air parcel entered the stratosphere) has been converted to $H_2O$, based on the simple calculation fraction = (1.67 ppmv−0.5 ppmv)/(1.67 ppmv) = 0.70. The time series for $H_2O$ shown in Fig. 2b is found from:

$$\Delta H_2O(yr) = 2 \times 0.7 \times CH_4^{SURFACE}(yr - 6) \qquad (5)$$

$$H_2O(yr) = \Delta H_2O(yr) + 2.306\ ppmv \qquad (6)$$

where the leading 2 in Eq. (5) accounts for the production of two $H_2O$ molecules upon loss of every $CH_4$ molecule, the factor of 0.7 and the 6-year lag have been explained just above, and the constant value of 2.306 ppmv is used to force polar stratospheric $H_2O$ to equal 4.6 ppmv in year 1990. The historical and future surface $CH_4$ time series that underlie Fig. 2b have been obtained from the various SSP scenarios[41]. Since the numerical value of the 0.7 terms in Eq. (5) depends on stratospheric OH (mainly), stratospheric Cl (second order), and the strength of the BDC,

this conversion factor could change over time. Our approach is simplistic, yet captures the primary first-order effect of changing $CH_4$ on polar stratospheric $H_2O$. The satellite-based data record for $H_2O$ that affords coverage of the polar regions starts in 1984, but trends are difficult to discern due to offsets between retrievals from various instruments that are commonly larger than the expected increase in polar $H_2O$ since 1984 (ref. [110]).

The projections for the effect of warming of the tropical tropopause shown in Fig. 2c are based on the analysis of output from CMIP6 GCMs shown in Figure 15 of Keeble et al.[43]. They document results from ten CMIP6 GCMs, for the four SSP scenarios shown in Fig. 2c, plus a few additional SSPs. We have computed a multi-model mean from the time series for nine of the ten GCMs, neglecting results from the UKESM1-0-LL GCM, because the results from this GCM seem to be an outlier (large future rise in stratospheric $H_2O$) compared to results from the other nine GCMs. We then apply a time-invariant, constant offset to this time series such that stratospheric $H_2O$ equals 4.60 ppmv in 1990. A few of the GCMs did not archive output for all four of the SSP scenarios used in our paper; in this case, we simply averaged output from all available GCMs. These ten CMIP6 GCMs tend, on average, to underestimate observed $H_2O$ in the tropical lower stratosphere[110] by nearly 1 ppmv from 1984 to the present, as shown in the upper panel of Figure 12 of Keeble et al.[43]. The abundance of $H_2O$ in the tropical lower stratospheric is governed by thermodynamics, whereas the abundance of $H_2O$ in the polar stratosphere is driven by this process as well as the oxidation of $CH_4$. This forecast of rising polar stratospheric $H_2O$ shown in Fig. 2c is consistent with a recent theoretical analysis of the future evolution of height and temperature of the tropical tropopause associated with global warming[50].

**ATLAS chemical transport model**. Simulations are performed with the ATLAS global Lagrangian Chemistry and Transport Model (CTM)[27,109]. Model runs are driven by meteorological data from the ERA5 reanalysis[34]. Descent rates are calculated directly from the heating rates provided by the ERA5 reanalysis. From the two different options provided by ECMWF, we use the total (all sky) heating rates and not the clear sky heating rates. The vertical range of the model domain is 350–1900 K and the horizontal resolution is 150 km. The run for winter 2020 starts on 1 September 2019 and ends on 1 May 2020, with the first 30 days consisting of model spin up. Additional runs with a similar setup for the Arctic winters of 2005, 2010, and 2011 are performed. Model values of $O_3$, $H_2O$, HCl, $N_2O$, $HNO_3$ and CO are initialized from the measurements obtained by the MLS instrument for the particular year (data obtained from https://mls.jpl.nasa.gov/data), and $ClONO_2$ is initialized from a climatology provided by the ACE-FTS instrument at http://www.ace.uwaterloo.ca/data.php. Initialization of $CH_4$, $NO_x$ and $Br_y$ are as described in Wohltmann et al.[109]. Reaction rates and absorption cross sections are from the 2015 NASA Chemical Kinetics and Photochemical Data for Use in Atmospheric Studies compendium https://jpldataeval.jpl.nasa.gov/pdf/JPL_Publication_15-10.pdf. A common deficiency of CTMs is a pronounced discrepancy between measured and modelled HCl mixing ratios in the Antarctic polar vortex, as described in section 6.1 of Wohltmann et al.[109]. Therefore, a temperature offset of −3 K was used for the calculation of the Henry constant of HCl, which improves this discrepancy.

Two additional ATLAS runs were started with the meteorological data of 2019/2020 and with scaling factors for chlorine and bromine relative to 2020, intended to simulate conditions for 2060 and 2100, respectively (Fig. 2a). The scaling factors for chlorine were 0.667 and 0.455 for 2060 and 2100, respectively, and the scaling factors for bromine were 0.778 and 0.694 for these two years. These scaling factors are based on the contributions of chlorine and bromine to polar EESC, found as described in the caption of Fig. 2.

The main article states: we use an exponent of 1.2 for EESC because this choice leads to the largest value of $r^2$ for the six ATLAS runs shown in Fig. 1b. Supplementary Fig. 12 illustrates the value of $r^2$ found as a function of the exponent $\eta$ in the expression:

$$\frac{EESC(yr)^{\eta}}{EESC_{MAX}^{\eta}} \times PFP(yr) \qquad (7)$$

The ATLAS runs for winters 2005, 2010, 2011, 2020, 2060, and 2100 exhibit a well-defined maximum in $r^2$ at $\eta = 1.2$, due to the large variation of EESC over these years. Conversely, the ozonesonde determinations of $\Delta O_3$ cannot be used to constrain $\eta$ because EESC varies by only ~15% from 1993 to 2020. The ozonesonde data are quite valuable for showing the near-linear dependence of $\Delta O_3$ with PFP (Fig. 1a). Values of $r^2$ as a function of $\eta$, for the expression $EESC^{\eta} \times ARP$, are also shown in Supplementary Fig. 12. The simulation of $\Delta O_3$ by the ATLAS model also exhibits a maximum near $\eta$ of 1.2 when $T_{ACL}$ is used rather than $T_{NAT}$, reinforcing the statement in the main article: remarkably similar conclusions based upon consideration of the temperature at which chlorine is activated on aerosols[6,32], rather than $T_{NAT}$.

**Exponent for EESC**. In the main article, we assess the uncertainty in $\Delta O_3^{REG}$ using lower and upper limits of 1 and 1.4 as the exponent for EESC in the expression for OLP. The lower limit of 1 corresponds to a linear dependence of chemical loss of Arctic $O_3$ on EESC, based upon the work of Douglass et al.[111] who showed that $\Delta O_3$ for the Arctic vortex varies linearly with EESC for fixed values of $V_{PSC}$, for values of EESC spanning 1990–2016. The upper limit of 1.4 was chosen because $r^2$ has the same value for $\eta = 1$ and $\eta = 1.4$ in Supplementary Fig. 12, and also

because Jiang et al.[112] showed that the variation of the chemical loss of Antarctic ozone varies as a function of chlorine loading to the power of 1.4 for 1980–1990, a period of rapid rise in the chlorine component of EESC.

**General circulation models (GCMs) and the QBO of zonal wind.** This paper relies extensively on archived GCM output. The computation of PFP is based upon analysis of horizontally and vertically resolved fields of temperature and pressure from 26 CMIP6 GCM simulations constrained by SSP5-8.5 projections of GHGs and 27 CMIP5 GCM runs constrained by RCP 8.5. Supplementary Fig. 13 shows the time series of PFP from CMIP5 GCMs in a manner analogous to Fig. 4 of the main article, which provides results for CMIP6 GCMs; Supplementary Table 4 provides tabular information regarding $S_{PFP−LM}$, $\Delta S_{PFP−LM}$, the temperature threshold offset for the existence of PSCs, and $p$-values for CMIP5 GCMs in a manner analogous to Table 1. The modelling centre and literature reference for each of these GCM simulations are given in Supplementary Table 1. On the CMIP5 archive model output is stored using a nomenclature of rLiMpN, where r refers to realization, i refers to initialization method, p refers to physics version, and L, M, and M are integers used to distinguish results from different runs from a particular GCM. Based upon file availability, we have used r1i1p1 output for all GCM runs except for r6i1p1 from CCSM4 for both the historical and RCP 8.5 simulations, r6i1p1 for the historical and r2i1p1 for the RCP 8.5 runs from GISS-E2-H as well as GISS-ER-2. For CMIP6 output, the nomenclature of rLiMpNfO is used, where r, i, and p are the same as described above, and f refers to the forcing index and O is a fourth integer. In this study, all output is from r1i1p1f1 files except for the use of r1i1p1f2 for historical and SSP runs from the CNRM-CM6-1, CNRM-CM6-1-HR, CNRM-ESM2-1, MIROC-ES2L, and UKESM1-0-LL GCMs, and the use of r1i1p1f3 for the historical and SSP runs from the HadGEM3-GC21-LL and HadEM3-GC21-MM GCMs.

Figure 7 of the paper shows the effect of time-dependent stratospheric $H_2O$ on the time series of PFP and OLP from the EC-Earth3 GCM. Supplementary Fig. 14 shows the effect of variable $H_2O$ on PFP and OLP from the other three GCMs that appear in Fig. 5. These other GCMs exhibit similar behaviour to the results from EC-Earth3 illustrated in Fig. 7, supporting the robustness of the time series for PFP and OLP across numerous GCMs.

In the main article, we state that examination of the tropical zonal wind from the GCMs indicates that the CMIP6 models tend to provide a better representation of the QBO than was evident in output from CMIP5 GCMs. This feature of the GCMs is illustrated in Supplementary Fig. 15 (reanalysis data) and 16 (GCMs). The model output shown in Supplementary Fig. 16 was mainly based upon archived monthly mean zonal wind fields from each GCM and complemented above 10 hPa by corresponding computed monthly means from daily/six-hourly data where needed; data for each panel are shown up to the highest altitude of each GCM. As can be seen from this figure, the representation of the QBO is considerably more realistic within the CMIP6 GCMs than the CMIP5 models.

Supplementary Fig. 17 is similar to Fig. 9, except trends are shown for $\Delta O_3^{REG}$ and OLP from the 20 CMIP6 GCMs that submitted results for all four SSPs to the CMIP6 archive that either: (a) exhibit a realistic QBO based upon our cursory examination or (b) do not exhibit a rendering of the QBO. There is little difference in the behaviour of $\Delta O_3^{REG}$ and OLP among these two groupings of the CMIP6 GCMs. As noted in the Main article, a more quantitative analysis of the representation of the QBO in these models reveals deficiencies in the mean amplitude below 20 hPa[54] and substantial effort is currently being directed towards improving the representation of the QBO within GCMs[55].

**Further considerations.** In the main article we state: the temporal evolution of $\Delta O_3^{REG}$ and OLP found using results from the four GCMs with interactive chemistry is about 20–25% lower at end of century than that found for the other 16 CMIP6 GCMs; nonetheless, $\Delta O_3^{REG}$ remains close to the contemporary value until the end of the century for the SSP3-7.0 and SSP5-8.5 simulations conducted using these interactive GCMs. This finding is illustrated by Supplementary Fig. 18, similar to Fig. 9 except results are shown only for the four CMIP6 GCMs with fully interactive stratospheric chemistry.

Supplementary Fig. 19 is also similar to Fig. 9, except trends are shown for the quantity:

$$\frac{EESC(yr)^{\eta}}{EESC_{MAX}^{\eta}} \times ARP(yr) \qquad (8)$$

computed from CMIP6 GCM output. This figure reinforces the notion that remarkably similar conclusions are found upon consideration of the temperature at which chlorine is activated, rather than the PSC existence temperature.

Finally, Supplementary Fig. 20 shows results similar to Figs. 8a and 9a, in this case illustrating how $\Delta O_3^{REG}$ and OLP vary as a function of time for a multi-model mean of the 27 CMIP6 GCMs that archived results for RCP 8.5, the 26 CMIP6 GCMs that recorded output for SSP 5-8.5, and a grand multi-model ensemble of all 53 GCM runs conducted using an end of century RF of climate equal to 8.5 W m$^{-2}$. Supplementary Figures 17 to 20 provide further evidence that the future rise in GHGs has the potential to cause a significant cooling of the Arctic stratosphere leading to conditions conducive to large, seasonal loss of Arctic $O_3$, particularly with future levels of stratospheric $H_2O$ as shown in Fig. 2d.

## Data availability

The data that support the findings of this study are available in Zenodo with the identifier https://doi.org/10.5281/zenodo.4414822. ERA5/ERA5.1/ERA5 BE (preliminary version) data are available at https://cds.climate.copernicus.eu/cdsapp#!/search?text=ERA5 (ERA5) as well as https://confluence.ecmwf.int/display/CKB/How+to+download+ERA5#Howtodownload ERA5-OptionB:DownloadERA5familydatathatisNOTlistedintheCDSonlinecatalogue-SLOW ACCESS (ERA5 BE prelim). CFSR and CFSv2 data are provided by NOAA's National Centers for Environmental Prediction and are available at https://climatedataguide.ucar.edu/climate-data/climate-forecast-system-reanalysis-cfsr (CFSR) and https://cfs.ncep.noaa.gov/cfsv2/downloads.html (CFSv2). MERRA-2 data are provided by the Global Modeling and Assimilation Office at NASA Goddard Space Flight Center and are available at https://disc.gsfc.nasa.gov/datasets/M2I3NVASM_5.12.4/summary (https://doi.org/10.5067/WWQSXQ8IVFW8). The Japanese 55-year Reanalysis (JRA-55) project was carried out by the Japan Meteorological Agency and the data are available at https://jra.kishou.go.jp/JRA-55/index_en.html. That dataset was collected and provided under the Data Integration and Analysis System (DIAS, Project No. JPMXD0716808999), which has been developed and operated by the Ministry of Education, Culture, Sports, Science and Technology. CMIP5 and CMIP6 GCM output are provided by the World Climate Research Programme's Working Group on Coupled Modelling and are available at https://esgf-node.llnl.gov/projects/cmip5 and https://esgf-node.llnl.gov/search/cmip6.

## Code availability

Code relating to this study is available from the corresponding author on request.

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

## Acknowledgements

Work conducted at the University of Maryland was supported by the U.S. National Aeronautics and Space Administration Atmospheric Composition and Modeling Program (ACMAP) under Grant No. 80NSSC19K0983. This study has been partly supported by the MOSAiC project under the ID: AWI_PS122_00. We appreciate the help of Laura A. McBride for facilitating access to radiative forcing of climate time series from the RCP and SSP databases. We thank the meteorological centres, i.e., ECMWF for ERA5, Japan Meteorological Agency (JMA) for JRA-55, Global Modeling and Assimilation Office (GMAO) at NASA Goddard Space Flight Center for MERRA2, NOAA's National Centers for Environmental Prediction (NCEP) for CFSR/CFSv2 data products, the World Climate Research Programme's Working Group on Coupled Modelling for coordinating and promoting CMIP5 & CMIP6, all climate modelling groups for producing and making available their model output, and the Earth System Grid Federation (ESGF) for archiving the data and providing access to the GCM output.

## Author contributions

M.R. and P.v.d.G. developed the code used to guide the acquisition of ozonesonde measurements and analyse the resulting data; P.v.d.G. developed the code to analyse the reanalyses and GCM model output and designed the ISA trend detection method; R.K. provided the radiosonde measurements from Sodankylä and performed the related analyses; I.W. conducted the ATLAS runs and led the interpretation of these results; R.J.S. and P.v.d.G. wrote the original draft that was reviewed and edited by M.R., R.K., and I.W. Finally, M.R. supervised and administrated the project.

## Funding

## Competing interests

The authors declare no competing interests.
