## [Peer Review File · Nature Communications]

Reviewer comments, first round –

Reviewer #1 (Remarks to the Author):

One of the main drivers of chemical loss of Arctic ozone due to anthropogenic halogens is temperature. In particularly cold winters, polar stratospheric clouds (PSC) are favoured and release of active chlorine and denitrification are enhanced. It is a well established result - reported also here in fig. 1 - how the ozone loss correlates with the potential for PSC formation (PFP), this latter parameter defined as the volume of air cold enough for PSC formation, normalized to the volume of air inside the polar vortex.

Hence the interest to study the evolution of this parameter, both from reanalysis of past meteorological data and projections from CCM, which is the theme of this paper.

A novel method, the Iterative Selection Approach (ISA) to detects trends in the occurrence of extreme cold winter events is suggested; this is compared to other methods used in the past, the Maximum in Interval Method (MIM) and the Value Above Sigma (VAS), this latter criticized for its non-applicability to the study of extremal value trends for non-stationary time series. The proposed novel method is extensively validated over randomized simulations, and is convincingly more robust and reliable.

The method is effective in finding a positive, statistically significant rise in the maxima of PFP over the past four decades in the meteorological data and

General Circulation Model (GCM) (projected) simulations which, as the authors suggest, seems driven in part by the build-up of anthropogenic GHGs.

The paper is clear and well referenced, the topic is interesting and timely, so I think it deserves to be published. Below are some comments the authors may address before publication.

1. To my knowledge, only a few of the CMIP5 models ran their simulations with interactive chemistry, while the vast majority uses prescribed ozone fields. A similar consideration applies to some of the CMIP6 models too. Which of the models whose results are here reported, run a full stratospheric chemistry and which not? Does that has an impact on the projections? This is not clear and the authors may specify and comment on such aspect.

2. In the method section the authors claim that "...while the shape of the probability density functions ... are not strictly Gaussian, the fall-offs of the tail of both functions are Gaussian-like" (382) Is this quantitatively supported? Have the authors computed the Pearson index to evaluate the kurtosis of the distribution?

3. The authors focus on the detection of trends in the occurrence of the coldest vortexes. These trends may arise when the time serie is increasing its variance, its (running) average, or both. It would be of interest to look more carefully at that. As instance they may apply (a sort of) ISA method to detect trends in the evolution of the warmest winters. Are they becoming less warm, as it would seem by taking the time series at face value, or is the system simply more variable? I think it would be interesting to comparre the trends of the coldest and warmest winters.

As stated above, the paper is well-laid and clear, its results are of interest for a vast community, so it surely deserves publication after addressing these few remarks.

Reviewer #2 (Remarks to the Author):

Review of the manuscript "The influence of climate change on chemical loss of ozone in the Arctic stratosphere" by Gathen et al.

This study challenges the view expressed, e.g. in Rieder and Polvani (2013) (RP13) and Rieder et al. (2014), that anthropogenic well-mixed greenhouse gases do not play a significant role in the past and projected evolution of lower-stratospheric polar temperature and ozone depletion, which are, instead, controlled mainly by the concentrations of ozone depleting substances. Specifically, through a careful analysis of the diagnostic PSC formation potential (PFP), this study provides evidence from reanalyses and model simulations for the "cold Arctic winters are getting colder"

paradigm consistent with Rex et al. (2006) among others. Furthermore, it demonstrates that the upward trend in the local maximum PFP projected for the 21st century correlates very well with radiative forcing due to well-mixed GHG. The authors argue that the analysis presented in RP13 was not sound from the statistical perspective and that it was overly simplistic in that it was done on the entire zonally symmetric polar region, which cannot be identified with the highly variable polar vortex where ozone depletion occurs.

The paper is very clearly written and easy to follow. The methods section provides all the definitions and calculations needed to reproduce the results and contains helpful discussions that justify the choices made by the authors (primarily the use of the iterative selection approach). Statistical significance estimation is discussed in detail.

I find the argument pertaining to the long-term PFP behavior convincing and interesting. It certainly makes for a valuable contribution to the debate. However, the connection between the "coldest Arctic winters are getting colder" finding and the future of ozone is entirely speculative in the paper, necessitating the use of phrases such as "could conceivably" (L206) and "may have been" (the abstract). See my general comment (2) for some discussion. This is disappointing, because at the same time, the prominent role that the subject of ozone recovery plays in Concluding Remarks suggests that it is meant as the central motivation for this work. I can think of two ways to remedy this: (1) if the connection with ozone is made more solid and quantitative or (2) if it is explicitly relegated to future work (it would be hard not to mention ozone at all!) and the speculative discussion is dropped. In my opinion, the PFP results are important and interesting enough to stand on their own.

General comments

1. The paper explicitly disputes the findings of Rieder and Polvani (2013) (RP13). In contrast to RP13, the approach here is more refined in two ways; (1) only VPSC and the related diagnostic, PFP inside the polar vortex is considered (RP13 base their analysis on data from latitudes $\geq 60^{\circ}\text{N}$ that don't necessarily coincide with the vortex, especially in the NH). (2) the use of ISA instead of VAS. However, RP13 note: "To confirm this, we have computed trends in several different ways and found nothing that is statistically significant on a rigorous test level (i.e., 95%). Specifically, we have computed the trend over (1) the entire time series, (2) the first and second halves separately, and (3) 5 year and 3 year maximum and minimum values. The p values for all of these are reported in Table S1: no trends are significant on a 95% level." Indeed, comparing the results here (Figs 4 and 4) and Figure 3 in RP13 that shows extended wintertime VPSC from a CAM ensemble, there is no clear trend in the latter, independent of how the maxima are selected. I can think of at least three (not necessary mutually exclusive) possibilities: (1) the dynamical response to increasing GHGs in CAM is fundamentally different than in the models discussed here; (2) the use of 60°N – 90°N range in RP13 (as opposed to the vortex interior) leads to trend cancelations; (3) It's because the metric is different and the trends seen in PFPLM reflect a decrease in the vortex volume rather than an increase in VPSC. It would be nice to see some discussion of this. I admit I find this discrepancy puzzling. While I absolutely agree that 60°N – 90°N cannot be identified with the polar vortex in general, is it really such a bad vortex proxy during very cold winters when the vortex is near-zonal and relatively undisturbed?

2. This one is crucial. While the analysis of PFP^{LM} trends using the ISA method appears solid (while reconciliation of it with RP13 may not be quite sufficient, see (1)), the connection that the paper makes with the future of polar ozone is tenuous. The authors demonstrate, assuming continuing GHG increase, that if a very cold winter does happen in the future it will likely be colder than in the past, and the polar vortex will host more PSCs. That's a solid conclusion from the CMIP5/6 results but the following sentence from the concluding remarks section, that justifies the importance of this study remains unsubstantiated: "Should stratospheric chlorine levels continue to decline at a slower than expected rate, the GHG-driven tendency for the coldest Arctic winters to get colder over time could conceivably drive even larger chemical loss of Arctic ozone than has previously occurred." What is meant by this? That "Arctic ozone holes" will occasionally continue to happen? How frequently? That polar ozone trends will be affected? That trend detection will be affected? Polar ozone loss is influenced by a number of factors, of which this paper discusses only one. To be precise about this the authors would have to actually show what happens to ozone

under, say SSP5-8.5 scenario together with some pessimistic ODS scenario. What about the frequency of strong/cold vortex events? What about projected changes to ozone resupply from the enhanced Brewer-Dobson circulation under increased GHG? The paper does mention the recent slowdown of the decline of halogens but its consequences are discussed only speculatively. By comparison, Keeble et al., 2020 demonstrates that the worst-case scenario (in some sense) is a 10-year delay in ozone recovery but not so much in the NH high latitudes. (their Figs 8 & 9). Other studies talk about the possibility of "super-recovery" (WMO 2018) resulting from higher GHG concentrations in the future, particularly at midlatitudes upper stratosphere. One might expect that this would also enhance polar springtime ozone through resupply. Even if we accept the hypothesis that the extreme ozone loss in 2020 is at least in part attributable to climate change (which is possible but not demonstrated in the paper) that does not automatically imply that an even colder winter a decade from now will have the same effect on ozone. Perhaps regressing Arctic ozone loss or vortex ozone content on PFP and halogen loading could help substantiate the connection with ozone.

Specific and technical comments

LL17-18. This sounds like an attribution result, but no quantitative attribution analysis is presented.

LL 20-22 If you say "interannual variability" do you need to say "for particular winter-spring" seasons? This reads a bit strange to me. Maybe I misunderstood something.

LL. 42-43: Wohltmann et al. doesn't appear to be a published paper or the reference given is insufficient to find it. Note that there is this new paper that discusses 2020 Arctic ozone: Manney et al. (2020)

L 58. "since this is straightforward to describe and implement" is not a good enough reason (L220 explains it better, though). I would just drop this sentence and simply refer the reader to Methods.

L62-65. Correct but then one (not me!) might ask: is it OK to use the reanalyses? The same reference (Lawrence et al., 2018) may be of help here as it provides specific recommendations for the use of polar processing diagnostics derived from reanalysis, based on a very thorough vetting process. That paper doesn't talk about ERA5 but the recommendations still apply, I think, and could be referenced very briefly (use modern reanalyses, use a number of them, not just one, etc., which is what you, in fact, do) There's also an important caveat here: stratospheric temperature-derived diagnostics (such as PFP) are more reliable after 1998/1999 when the reanalyses start assimilating AMSU-A data (see also Long et al., 2017). Also note the following cautionary note from Lawrence et al. (2018): "Since many of the major changes in data inputs are made at approximately the same time in each reanalysis, the agreement or lack thereof between the reanalyses does not provide the information to assess the degree to which these changes are caused by changes to the assimilated observations. We thus emphasize here that (...) use of reanalyses in trend studies should be regarded with skepticism and only attempted after rigorous assessment of the relationships of temperature changes to observations assimilated." I don't think this means that one absolutely shouldn't look at trends derived from reanalyses and that those trends are necessarily artifacts of observing system changes (in fact we know that in many instances that is not the case, and reanalyses have gone a long way toward being "trend-friendly" scientific tools; I myself have been known for using them for trend studies in some limited context) but it would be good to justify the validity of what you're doing here a little more. I realize this is a tall order for a diagnostic for which we don't have direct measurements, but a few words of justification would go a long way. One simple suggestion: repeat the trend calculations for the period 1999–2020. ATOVS radiance data during that recent period provide a strong constraint and studies don't show significant spurious discontinuities in stratospheric temperatures in the ATOVS period (Long et al. 2017, Lawrence et al., 2018). Statistical significance will go down, of course, but it may be instructive to compare what you get with the trends computed over the 40-year period. This comment may sound like I fundamentally mistrust reanalyses. This is not the case. But being close to the reanalysis world I have a tendency to be extra careful in this area, that's all.

L66 The canonical reference for ERA5 is Hersbach et al (2020).

L85. "because these days encompass the full range of PSC activity among all meteorological fields" I'm struggling to understand what that means. Please, consider revising. In addition, I presume these dates were chosen because significant amounts of PSCs occur in that period (supplementary Fig. 1). But is that true just for the historical period or for the 21st century simulations as well? If, for example, there were significant PSCs in some of the simulations outside of that time window that could distort the results.

L94 What does "accurate" mean in this context? I realize this is explained in Methods but it would be good to have a sentence of explanation here.

L114. Personally, I would change that title to something that reflects that this section is about 20-21st century model simulations. "Computed PFP" suggests that the PFP in the preceding section wasn't computed;)

LL127-130. Nice!

L146. But dependent on the future trajectory of the concentrations of those halogens!

LL152-155. Or the reanalyses may overestimate the trends due to discontinuities induced by step changes in their observing systems

LL157-160. Nice. I would expand it just a tad: "...role of internal variability and uncertainties in trends derived from reanalyses..." or something to that effect.

LL163-167. Nice!

LL170–175 and LL410–418. I suggest making a stronger connection here. Why is the representation of the QBO a good metric for polar processes (e.g. Holton-Tan mechanism)? Also please add some references for QBO in CMIP (the SPARC-QBOi activity: Butchart et al, 2020, Richter et al., 2020a, b). For example, Richter et al. (2020b) performed a detail comparisons of the QBO representation in different versions of CMIP, including 5 and 6. They state "Despite the tripling of the number of models that are able to simulate the QBO, the quality of the simulation of the QBO from CMIP5 to CMIP6 has not changed." LL410-412 state that the latter simulate the QBO better than the former. Looking at Supplementary Figures 11 and 12 this appears to simply reflect the fact that most of the CMIP5 didn't generate any QBO at all and that situation improved in CMIP6. If the correct representation of the QBO is important then wouldn't it make sense to separate the results shown in Figs 3 and 4, and supplementary Figs 8 and 9 into groups of models that do and do not generate the QBO? For example, a cursory comparison of Fig. 3 and supplementary figure 12 reveals that the models that produce PFPLM trends in excess of 2 d/decade are those without the QBO, and two of them have a very low top.

LL199-200. See my general comment 2.

LL200-201. The "and" there sounds a little awkward; China is in eastern Asia. Please, consider revising.

LL206-207. There are several trends happening here with opposite impacts on ozone: ODS declining, cold winters getting colder, etc. It is not at all clear which wins and under what scenarios. See my general comment 2.

L215. Is the assumption of constant H2O valid for the 21st century projections? How sensitive are TNAT to possible water vapor increase due to, e.g. increasing upper stratospheric methane?

LL 329-329. "13.6 d decade⁻¹". Aren't the units of PFP days, [d]?

LL260–266. Supplementary Figure 5 shows a rapid increase in maximum frequencies starting in the mid-1990s. Can you comment on that? Is the temporal resolution of the sonde launches uniform throughout the entire period? Was there a change in instrumentation? Also, are the

Sodankylä sondes assimilated in JRA-55?

L264. launched  "launches" (?)

L270. "criteria"  "criterion"

Supplementary material

L22. Comparing with Fig.3 in the main text it seems to me that it's the other way around: the lower panels are with adjustments. Am I misreading this figure?

L22. "and adjustment"  "an adjustment"

References

Butchart, N., Anstey, J. A., Kawatani, Y., Osprey, S. M., Richter, J. H., & Wu, T. (2020). QBO changes in CMIP6 climate projections. *Geophysical Research Letters*, 47, e2019GL086903. <https://doi.org/10.1029/2019GL086903>

Hersbach, H, Bell, B, Berrisford, P, et al. The ERA5 global reanalysis. *Q J R Meteorol Soc.* 2020; 1–51. <https://doi.org/10.1002/qj.3803>

Keeble, J., Abraham, N. L., Archibald, A. T., Chipperfield, M. P., Dhomse, S., Griffiths, P. T., and Pyle, J. A.: Modelling the potential impacts of the recent, unexpected increase in CFC-11 emissions on total column ozone recovery, *Atmos. Chem. Phys.*, 20, 7153–7166, <https://doi.org/10.5194/acp-20-7153-2020>, 2020.

Long, C. S., Fujiwara, M., Davis, S., Mitchell, D. M., and Wright, C. J.: Climatology and Interannual Variability of Dynamic Variables in Multiple Reanalyses Evaluated by the SPARC Reanalysis Intercomparison Project (S-RIP), *Atmos. Chem. Phys.*, 17, 14593–14629, 2017.

Manney, G. L., Livesey, N. J., Santee, M. L., Froidevaux, L., Lambert, A., Lawrence, Z. D., et al.. (2020). Record-low Arctic stratospheric ozone in 2020: MLS observations of chemical processes and comparisons with previous extreme winters. *Geophysical Research Letters*, 47, e2020GL089063. <https://doi.org/10.1029/2020GL089063>

Richter, J. H., Butchart, N., Kawatani, Y., Bushell, A. C., Holt, L., Serva, F., Anstey, J., Simpson, I., Osprey, S., Hamilton, K., Braesicke, P., Cagnazzo, C., Chen, C.-C., Garcia, R., Grey, L., Kerzenmacher, T., Lott, F., McLandress, C., Naoe, H., Scinocca, J., Stockdale, T., Versick, S., Watanabe, S., Yoshida, K., & Yukimoto, S. (2020a). Response of the Quasi-Biennial Oscillation to a warming climate in global climate models. *Quarterly Journal of the Royal Meteorological Society*, 1–29. <https://doi.org/10.1002/qj.3749>

Richter, J. H., Anstey, J. A., Butchart, N., Kawatani, Y., Meehl, G. A., Osprey, S., & Simpson, I. R. (2020b). Progress in simulating the quasi-biennial oscillation in CMIP models. *Journal Geophysical Research: Atmospheres*, 125, e2019JD032362. <https://doi.org/10.1029/2019JD032362>

WMO (World Meteorological Organization) (2018). *Scientific Assessment of Ozone Depletion: 2018. Global Ozone Research and Monitoring Project–Report No. 58*, 588 pp., Geneva, Switzerland.

Kris Wargan

Reviewer #3 (Remarks to the Author):

Review of “The influence of climate change on chemical loss ...”

General

Let me start with my overall conclusion first: this is a strong paper that deserves to be published in *Nat. Commun.* – the paper finds that conditions allowing heterogeneous chlorine activation and thus polar ozone loss are increasingly favourable for ozone loss over the past four decades. As a result, increasingly severe Arctic ozone loss has occurred over these time scales. In simple words, the message of the paper is that cold Arctic winters are getting colder.

This time evolution of Arctic temperatures is analysed based on reanalyses from four meteorological centres and on the results from a large number of general circulation model simulations. The paper concludes that stratospheric cooling caused by human release of greenhouse gases likely drives the change in Arctic temperature patterns and thus the increasingly severe Arctic ozone loss. These results are clearly of significance to the field of stratospheric ozone loss, but also have various implications for the science of global change (see also discussion below).

Earlier work in this field (e.g. Tilmes et al., 2006; Rex et al., 2004, 2006) had used different data sets (that are partly superseded by more modern products); a strong point of the recent study is the consistent use of reanalysis schemes from four different centres. I think one can say that the currently best available data sets have been employed here. An important aspect is further the use of the results from general circulation model simulations. In this way, the authors can make an attribution statement about the likely cause of changes in Arctic temperature patterns and chemical ozone loss. This is important.

Overall, the methodology is sound and certainly meets the expected standards in the field. I have a few points (see below) where I think the paper would gain from a somewhat extended discussion. (I would also suggest providing more results in tabulated form (electronically) to make the use and reproduction of the present results easier.) I think this can be done.

A point, where I am critical is the focus on the cold winters. While the cold winters are relevant for Arctic ozone loss, the investigation of the causes of the variability and the attribution of trends to increasing concentrations of greenhouse gases requires a look at the warmer winters too. How are the warmer winters affected by greenhouse gases and how relevant are changes in stratospheric wave activity? (See discussion below.)

Finally. I am citing references in this review to support the points I am making; I do *not* mean to suggest that those references should be cited in the manuscript.

In summary, although I am suggesting some revisions, this paper deserves to be published in *Nat. Commun.*.

Discussion

Polar stratospheric cloud (PSC) formation potential: PSC formation potential (PFP)

The analysis here is based on a particular way of calculating PFP. However there are other alternatives which should be discussed. Tilmes et al. (2007, 2006) used a quantity referred to as ‘potential for the activation of chlorine’, which is similar to PFP but uses a threshold temperature for chlorine activation on cold sulphate aerosol (Drdla and Müller, 2012) and is a function of temperature, water vapour, and the surface area density of stratospheric sulphate. The potential for the activation of chlorine is thus more dependent on the stratospheric sulphate loading and thus to the occurrence of volcanic eruptions, although for some conditions it is comparable to PFP. It would also be possible to take into account the changing stratospheric halogen loading when defining a quantity like PFP, which might be helpful, when considering model results over 1950 to 2100.

There is some discussion in the paper regarding violation of the Montreal protocol. However, these violations happen against the background of the changing (and regulated by the Montreal protocol) stratospheric halogen loading, which should not be neglected in the paper. Note that Chipperfield et al. (2015) concluded that a very strong Arctic ozone loss, with column ozone values below 120 DU, would have occurred without the Montreal protocol for meteorological conditions as in 2011. Moreover, not only chlorine levels might develop in a different way than predicted, the strength of the chemical ozone loss might also be affected by bromine levels and the contribution of short-lived bromine substances (e.g., Fernandez et al., 2017).

PFP and ozone loss

Figure 1: As Fig. 1 in the submitted manuscript, but for the subset of the five coldest winters (2020, 2011, 2005, 1996, 2000, if I am reading the colour scale correctly).

Figure 1 in the submitted manuscript describes the relation between column ozone loss in the vortex and PFP. First, I think the paper could be clearer about the fact that the ozone loss reported here has been determined with the

Match method. The paper should also cite the papers, where all ozone loss values except 2020 have been published (assuming this is true). If there is anything particular to say about the ozone loss determination for 2020 it could be discussed in the appendix (or perhaps a preprint could be cited). Another question that arises is how well the ozone loss determined from Match compares with other ozone loss estimates. Second, if one only considers the five coldest winters (Fig. 1) the correlation between ozone loss and PFP is less obvious.

I think we all agree that the ozone loss in 2020 was particularly strong as it was cold enough for heterogeneous chemistry for an extended period in March and April. A hypothetical winter/spring with a very cold vortex from December to February but with a warming in (say) early March, would certainly have much less ozone depletion than observed in 2020 even if the PFP value were the same as in 2020 (about 29 d). On the other hand, there could be a small volume of the polar vortex below the threshold for heterogeneous chemistry, which nonetheless is responsible for activating a large fraction of the Arctic vortex (Wegner et al., 2016). Because the typical life cycle of the polar vortex incorporates implicitly a link between a cold and a long-lived vortex, concepts like PFP are useful indicators of polar ozone loss. However, I suggest a bit more careful discussion of the subject in the manuscript.

Warm winters

Somewhat crudely speaking the message of this manuscript is that “cold Arctic winters are getting colder”. However most winters in the analysed time series are not ‘cold’ but are rather ‘warm’. If the message of the paper is that the “colder cold winters” are driven by the increase in anthropogenic GHGs, why does this not affect the ‘warm’ winters? This question is left open in the manuscript, isn’t it?

I suggest applying the ISA technique put forward here (or some variant of it) also to warm winters, i.e. follow a similar strategy as outlined in lines 287-298, but remove the ‘cold’ years from the time series. The analysis has to be done first, but I would expect (see Fig. 2 of the manuscript) that the conclusion is that the frequency of warm winters has not changed significantly over the time period of 1989 to 2020. On the other hand, from inspecting Fig. 2e, it seems that what used to be a cold winter in the late 1960s is considered a warm winter after the year 2000 by the ISA method. I think adding such a discussion would allow a weak point of this paper (what happens to the warm winters?) to be addressed.

Changes in stratospheric wave intensity

I suggest that the paper gives a bit more background and discussion on the cases of the variability (which is evident e.g. in Figures 2, 3, and 4 in the manuscript) in vortex temperatures, vortex strength and vortex position. For example, studies have reported that the tropospheric climate shifted around the year 2000 and that the December stratospheric wave intensity during 2001-2015 has weakened, in contrast to that during 1979-2000, implying a shift of stratospheric wave intensity around 2000 (Hu et al., 2019, and references therein). Hu et al. (2019) further find that the shift in stratospheric wave intensity leads to a shift in stratospheric Arctic temperatures, that is, to a warming during 1979-2000 and

to a cooling during 2001-2015. This could imply that similar shifting phenomena may appear regardless of the continued greenhouse gas emissions. Could such findings have an impact on the conclusions of the present manuscript?

Further, the variability of Arctic ozone is also related to sea surface temperatures; e.g., Liu et al. (2020) find that the distribution of ozone is related to the positive phase of sea surface temperatures over the North Pacific, which could also have an impact on the stratospheric Arctic vortex. And there could be a feedback between stratospheric ozone depletion and the position of the polar vortex. Studies (e.g., Zhang et al., 2020, and references therein) have found a shift of the Arctic stratospheric polar vortex toward Siberia during late winter since 1980 and there are reports that zonally asymmetric stratospheric ozone depletion gives a significant feedback on the position of the polar vortex (with changes happening over the time period of interest here).

Summary Figure

Figure 2: Similar as Fig. 2 in the submitted manuscript, but focusing on the ‘cold’ winters only. Blue symbols show ERA5 data. Red symbols show JRA-55 data. Shown are the ‘cold winter’ data and the corresponding fits. (Note that the data points have been crudely extracted from the plot provided in the manuscript; this is only meant as a sketch.)

One point that I think is missing in the paper is a good summary (iconic) figure for the message of the paper. In Figure 2 of this review I have tried to provide a sketch how such a figure could look. I suggest focusing on the ‘cold’ winters only, which is the focus of the manuscript. One could try including also the information (fits?) from Merra2 and CFSR. I would definitely suggest including information (grey scale?) from the models (Fig. 3 of the main paper). My suggestion would be to focus on ERA5 and JRA-55, but this is obviously a choice of the authors. However, I note that in Fig. 2 the JRA-55 information is shown twice (this is not referring to the fit), so there could be space for showing a summary figure.

Details

- l. 10, 11: This sounds as if the formation of PSCs is a requirement for the onset of polar ozone loss. However, polar ozone loss is really driven by the onset of heterogeneous chlorine activation, which is not the same things as the formation of PSCs.
- l. 34-42: you discuss here the results from three particular GCMs – how relevant is this discussion given the fact that later in the paper you discuss the results of a number of CMIP model results and furthermore point to the importance of a well resolved stratosphere (e.g. quality of the QBO representation).
- l 54: The formation mechanisms of ternary solution droplets (STS) and NAT are rather different, NAT formation requires formation of crystals and thus supersaturation, while STS are liquid particles; the formation mechanism of large NAT particles (ref. 20) is likely a different issue again (e.g., Molleker et al., 2014; Hoffmann et al., 2017; Pitts et al., 2018). I suggest being a bit more specific here.
- l. 58: I suggest being more specific here, although I agree in principle. But heterogeneous chlorine activation requires neither NAT nor STS to be present; it can start on cold sulphate aerosol (e.g., Wegner et al., 2012; Drdla and Müller, 2012, refs. 21 and 23 of the paper). Further HNO_3 is likely not constant in a cold Arctic vortex (e.g., Waibel et al., 1999). Finally, T_{NAT} depends on water vapour and HNO_3 , whereas HNO_3 is not relevant for estimating the threshold for activation on cold sulphate aerosol, so the two thresholds cannot always agree.
- l. 72: note that the ECMWF says: “The entire ERA5 dataset from 1950 to present is expected to be available for use in 2020”.
- l. 78: boundary of the vortex (see also Methods): I would state the value of 36 sec^{-1} for nPV in the main paper. But PV is strongly altitude dependent (which is why nPV is used), so how well does nPV ‘correct’ for this altitude dependence? How does nPV compare to other forms of scaled PV (e.g., Lait, 1994)? Perhaps more importantly; the paper states that temperatures are changing in the stratosphere because of global change; at the same time the assumption is that PV is not affected so that using the constant vortex boundary value of 36 sec^{-1} for nPV is correct. How well is this assumption justified? (see also Liu et al., 2020)
- l. 88: why do you start in 1993?
- l. 111: the meaning of ‘p-values’ is not clear here
- l. 119: quantify ‘substantial’
- l. 126: I think I know what you mean, but this sentence is not really clear
- l. 146: do you mean ‘stratosphere’ here or ‘polar stratosphere’?
- l. 158: (and the discussion in the entire paragraph). This discussion sounds a bit speculative, I suggest to shorten. The discussion focuses on ‘internal

variability’, however, given the large range of the temperature (and thus PFP) mismatch in the CCM results, I could imagine other inadequacies of the models leading to problems simulating $S_{\text{PFP-LM}}$ than just ‘internal variability’.

- l. 183: what is shown in this paper cannot have affected the ‘reluctance’ in the past mentioned here.
- l. 186: the authors of this paper clearly do not agree with the results of Rieder and Polvani (2013) and Rieder et al. (2014) – I think that for the present paper a consensus on this issue (e.g., VAS against ISA) should not be required; I would expect further discussion on this issue in the scientific literature
- Ref. 16: see also the special issue in JGR/GRL on the ozone loss in the Arctic in 2020 (preprints available at: <https://www.essoar.org/>) and the discussion paper by Dameris et al. (2020).
- l. 200: See Rigby et al. (2019) for the origin of illegal recent production of CFC-11; also note that stratospheric bromine levels might change (e.g., Fernandez et al., 2017)
- l. 203: can you quantify ‘significant’?
- l. 211: I agree, but assuming that society takes action in reducing the emissions of GHGs, what would be the timescale over which stratospheric temperatures might change (e.g. compared to the expected decline in the stratospheric halogen loading).
- l. 222: you say ‘we might have’: what would be the expected consequences if you did so.
- l. 226: as discussed above how well is the constant vortex boundary value of 36 sec^{-1} for nPV justified? Over which altitude regime is it valid?
- l. 241: it would be good if the value $c(\theta)$ is available in the paper (e.g., for reproducing the results described here)
- I suggest to be more explicit here what is meant with ‘anomalous behaviour’.
- l. 248: what about the vertical resolution?
- l. 273, 274: This is not clear to me, what is a ‘possible 1 K offset’?
- Fig. 1: the colour scale used here is not ideal, some of the years are difficult to distinguish
- Data availability: I suggest also to make the data (Ozone loss, PFP etc) use to produce the figures in the paper easily available as tables in the electronic supplement. This would considerably help making the results of this paper easily reproducible in follow up studies.

References

- Chipperfield, M. P., Dhomse, S. S., Feng, W., McKenzie, R. L., Velders, G. J. M., and Pyle, J. A.: Quantifying the ozone and ultraviolet benefits already achieved by the Montreal Protocol, *Nat. Commun.*, 6, 7233, <https://doi.org/10.1038/ncomms8233>, 2015.
- Dameris, M., Loyola, D. G., Nützel, M., Coldewey-Egbers, M., Lerot, C., Romahn, F., and van Roozendaal, M.: First description and classification of the ozone hole over the Arctic in boreal spring 2020, *Atmos. Chem. Phys. Discuss.*, 2020, 1–26, <https://doi.org/10.5194/acp-2020-746>, URL <https://acp.copernicus.org/preprints/acp-2020-746/>, 2020.
- Drdla, K. and Müller, R.: Temperature thresholds for chlorine activation and ozone loss in the polar stratosphere, *Ann. Geophys.*, 30, 1055–1073, <https://doi.org/10.5194/angeo-30-1055-2012>, 2012.
- Fernandez, R. P., Kinnison, D. E., Lamarque, J.-F., Tilmes, S., and Saiz-Lopez, A.: Impact of biogenic very short-lived bromine on the Antarctic ozone hole during the 21st century, *Atmos. Chem. Phys.*, 17, 1673–1688, <https://doi.org/10.5194/acp-17-1673-2017>, URL <http://www.atmos-chem-phys.net/17/1673/2017/>, 2017.
- Hoffmann, L., Spang, R., Orr, A., Alexander, M. J., Holt, L. A., and Stein, O.: A decadal satellite record of gravity wave activity in the lower stratosphere to study polar stratospheric cloud formation, *Atmos. Chem. Phys.*, 17, 2901–2920, <https://doi.org/10.5194/acp-17-2901-2017>, URL <http://www.atmos-chem-phys.net/17/2901/2017/>, 2017.
- Hu, D., Guo, Y., and Guan, Z.: Recent Weakening in the Stratospheric Planetary Wave Intensity in Early Winter, *Geophys. Res. Lett.*, 46, 3953–3962, <https://doi.org/10.1029/2019GL082113>, URL <https://agupubs.onlinelibrary.wiley.com/doi/abs/10.1029/2019GL082113>, 2019.
- Lait, L. R.: An alternative form for potential vorticity, *J. Atmos. Sci.*, 51, 1754–1759, 1994.
- Liu, M., Hu, D., and Zhang, F.: Connections Between Stratospheric Ozone Concentrations Over the Arctic and Sea Surface Temperatures in the North Pacific, *J. Geophys. Res.*, 125, e2019JD031690, <https://doi.org/10.1029/2019JD031690>, URL <https://agupubs.onlinelibrary.wiley.com/doi/abs/10.1029/2019JD031690>, e2019JD031690 2019JD031690, 2020.
- Molleker, S., Borrmann, S., Schlager, H., Luo, B., Frey, W., Klingebiel, M., Weigel, R., Ebert, M., Mitev, V., Matthey, R., Woiwode, W., Oelhaf, H., Dörnbrack, A., Stratmann, G., Groß, J.-U., Günther, G., Vogel, B., Müller, R., Krämer, M., Meyer, J., and Cairo, F.: Microphysical properties of synoptic-scale polar stratospheric clouds: in situ measurements of unexpectedly large HNO₃-containing particles in the Arctic vortex, *Atmospheric Chemistry and Physics*, 14, 10 785–10 801, <https://doi.org/10.5194/acp-14-10785-2014>, URL <http://www.atmos-chem-phys.net/14/10785/2014/>, 2014.

- Pitts, M. C., Poole, L. R., and Gonzalez, R.: Polar stratospheric cloud climatology based on CALIPSO spaceborne lidar measurements from 2006 to 2017, *Atmos. Chem. Phys.*, 18, 10 881–10 913, <https://doi.org/10.5194/acp-18-10881-2018>, URL <https://www.atmos-chem-phys.net/18/10881/2018/>, 2018.
- Rex, M., Salawitch, R. J., von der Gathen, P., Harris, N. R. P., Chipperfield, M. P., and Naujokat, B.: Arctic ozone loss and climate change, *Geophys. Res. Lett.*, 31, L04116, <https://doi.org/10.1029/2003GL018844>, 2004.
- Rex, M., Salawitch, R. J., Deckelmann, H., von der Gathen, P., Harris, N. R. P., Chipperfield, M. P., Naujokat, B., Reimer, E., Allaart, M., Andersen, S. B., Bevilacqua, R., Braathen, G. O., Claude, H., Davies, J., De Backer, H., Dier, H., Dorokov, V., Fast, H., Gerding, M., Godin-Beekmann, S., Hoppel, K., Johnson, B., Kyrö, E., Litynska, Z., Moore, D., Nakane, H., Parrondo, M. C., Risley Jr., A. D., Skrivankova, P., Stübi, R., Viatte, P., Yushkov, V., and Zerefos, C.: Arctic winter 2005: Implications for stratospheric ozone loss and climate change, *Geophys. Res. Lett.*, 33, L23808, <https://doi.org/10.1029/2006GL026731>, 2006.
- Rieder, H. E. and Polvani, L. M.: Are recent Arctic ozone losses caused by increasing greenhouse gases?, *Geophys. Res. Lett.*, 40, 4437–4441, <https://doi.org/10.1002/grl.50835>, URL <https://agupubs.onlinelibrary.wiley.com/doi/abs/10.1002/grl.50835>, 2013.
- Rieder, H. E., Polvani, L. M., and Solomon, S.: Distinguishing the impacts of ozone-depleting substances and well-mixed greenhouse gases on Arctic stratospheric ozone and temperature trends, *Geophys. Res. Lett.*, 41, 2652–2660, <https://doi.org/10.1002/2014GL059367>, URL <https://agupubs.onlinelibrary.wiley.com/doi/abs/10.1002/2014GL059367>, 2014.
- Rigby, M., Park, S., Saito, T., Western, L. M., Redington, A. L., Fang, X., Henne, S., Manning, A. J., Prinn, R. G., Dutton, G. S., Fraser, P. J., Ganesan, A. L., Hall, B. D., Harth, C. M., Kim, J., Kim, K.-R., Krummel, P. B., Lee, T., Li, S., Liang, Q., Lunt, M. F., Montzka, S. A., Mühle, J., O’Doherty, S., Park, M.-K., Reimann, S., Salameh, P. K., Simmonds, P., Tunnicliffe, R. L., Weiss, R. F., Yokouchi, Y., and Young, D.: Increase in CFC-11 emissions from eastern China based on atmospheric observations, *Nature*, 569, 546–550, URL <https://doi.org/10.1038/s41586-019-1193-4>, 2019.
- Tilmes, S., Müller, R., Engel, A., Rex, M., and Russell III, J.: Chemical ozone loss in the Arctic and Antarctic stratosphere between 1992 and 2005, *Geophys. Res. Lett.*, 33, L20812, <https://doi.org/10.1029/2006GL026925>, 2006.
- Tilmes, S., Kinnison, D., Müller, R., Sassi, F., Marsh, D., Boville, B., and Garcia, R.: Evaluation of heterogeneous processes in the polar lower stratosphere in the Whole Atmosphere Community Climate Model, *J. Geophys. Res.*, 112, D24301, <https://doi.org/10.1029/2006JD008334>, 2007.
- Waibel, A. E., Peter, T., Carslaw, K. S., Oelhaf, H., Wetzell, G., Crutzen, P. J., Pöschl, U., Tsias, A., Reimer, E., and Fischer, H.: Arctic Ozone Loss Due to Denitrification, *Science*, 283, 2064–2069, 1999.

- Wegner, T., Grooß, J.-U., von Hobe, M., Stroh, F., Sumińska-Ebersoldt, O., Volk, C. M., Hösen, E., Mitev, V., Shur, G., and Müller, R.: Heterogeneous chlorine activation on stratospheric aerosols and clouds in the Arctic polar vortex, *Atmos. Chem. Phys.*, 12, 11 095–11 106, <https://doi.org/10.5194/acp-12-11095-2012>, 2012.
- Wegner, T., Pitts, M. C., Poole, L. R., Tritscher, I., Grooß, J.-U., and Nakajima, H.: Vortex-wide chlorine activation by a mesoscale PSC event in the Arctic winter of 2009/10, *Atmos. Chem. Phys.*, 16, 4569–4577, <https://doi.org/10.5194/acp-16-4569-2016>, URL <https://www.atmos-chem-phys.net/16/4569/2016/>, 2016.
- Zhang, J., Tian, W., Xie, F., Pyle, J. A., Keeble, J., and Wang, T.: The Influence of Zonally Asymmetric Stratospheric Ozone Changes on the Arctic Polar Vortex Shift, *J. Climate*, 33, 4641–4658, <https://doi.org/10.1175/JCLI-D-19-0647.1>, URL <https://doi.org/10.1175/JCLI-D-19-0647.1>, 2020.

REVIEWER COMMENTS

We sincerely appreciate the comments of the three reviewers, which have resulted in a major revision to the paper, leading to what we feel is a much stronger manuscript.

When the original paper had been submitted, it was first sent to *Nature Climate Change (NCC)*, which has a limit of 3000 words, 50 citations, and 6 display items. The editors of NCC decided not to send the submitted paper out for review and gave us the option of passing the manuscript along to *Nature Communications (NC)*. At this stage, we then had the option of either revising the manuscript, since NC allows for longer papers than NCC, or having the paper submitted to NCC be considered “as is” for NC. For various reasons, one of which was time expediency, we decided on the second option. Since NC allows for much longer papers than NCC (5000 words; 70 citations; 10 display items), we actually have considerable space to address the excellent set of comments from the three reviewers.

The revised paper builds upon the original paper by assessing how trends in stratospheric H₂O induced by rising CH₄ and the thermodynamics of the tropical tropopause will affect the future of PSC formation potential (PFP) in the Arctic vortex, shows that the observed chemical loss of column ozone in the Arctic (ΔO_3) is well described by the product of Equivalent Effective Stratospheric Chlorine (EESC) raised to the 1.2 power multiplied by PFP ($EESC^{1.2} \times PFP$). We now examine trends in $EESC^{1.2} \times PFP$ from the ensemble average of CMIP6 GCM output, in addition to trends in the local maximum of PFP (PFP^{LM}) that was the focus of the original paper. We have also added results from the ATLAS Lagrangian model that includes a comprehensive treatment of stratospheric chemistry, to provide theoretical support for the relation between ΔO_3 and $EESC^{1.2} \times PFP$. As a result, Ingo Wohltmann has joined the author team.

Generally, we have strived to keep our responses short when the text has been revised in response to the comments, focusing mainly on quoting how the text has changed. **All line numbers are for the version of the manuscript for which changes have not been tracked, since this might be easier to read, given the large number of changes.** We eagerly await a new round of reviews of the revised manuscript, and sincerely appreciate the time and effort of the reviewers.

Reviewer #1 (Remarks to the Author):

One of the main drivers of chemical loss of Arctic ozone due to anthropogenic halogens is temperature. In particularly cold winters, polar stratospheric clouds (PSC) are favoured and release of active chlorine and denitrification are enhanced. It is a well established result - reported also here in fig. 1 - how the ozone loss correlates with the potential for PSC formation (PFP), this latter parameter defined as the volume of air cold enough for PSC formation, normalized to the volume of air inside the polar vortex.

Hence the interest to study the evolution of this parameter, both from reanalysis of past meteorological data and projections from CCM, which is the theme of this paper.

A novel method, the Iterative Selection Approach (ISA) to detects trends in the occurrence of extreme cold winter events is suggested; this is compared to other methods used in the past, the Maximum in Interval Method (MIM) and the Value Above Sigma (VAS), this latter criticized for its non-applicability to the study of extremal value trends for non-stationary time series. The proposed novel method is extensively validated over randomized simulations, and is convincingly more robust and reliable.

The method is effective in finding a positive, statistically significant rise in the maxima of PFP over the past four decades in the meteorological data and

General Circulation Model (GCM) (projected) simulations which, as the authors suggest, seems driven in part by the build-up of anthropogenic GHGs.

The paper is clear and well referenced, the topic is interesting and timely, so I think it deserves to be published. Below are some comments the authors may address before publication.

Much thanks for these kind words.

1. To my knowledge, only a few of the CMIP5 models ran their simulations with interactive chemistry, while the vast majority uses prescribed ozone fields. A similar consideration applies to some of the CMIP6 models too. Which of the models whose results are here reported, run a full stratospheric chemistry and which not? Does that have an impact on the projections? This is not clear and the authors may specify and comment on such aspect.

Much thanks; this is an important point.

For the revised version of the paper, we show results using archived output from 27 CMIP5 GCM simulations that ran RCP 8.5 (same set of models used in the original submission) and for 26 CMIP6 GCMs. 20 of which archived results for SSP5-8.5, SSP3-7.5, SSP2-4.5, and SSP1-2.6. Of these 20 CMIP6 GCMs, four use full interactive stratospheric chemistry: CESM2-WACCM, CNRM-ESM2-1, MRI-ESM2-0, UKESM1-0-LL ESM. In the revised version of the paper, we have added a discussion to the main article about these models that reads:

Ideally, GCMs would include interactive chemistry, as there are numerous feedbacks and interactions between the photochemical processes that regulate stratospheric ozone and the dynamical and radiative drivers of PFP. Four of the 20 CMIP6 GCMs considered above have fully interactive chemistry: the other 16 models use prescribed fields of ozone. The temporal evolution of $EESC^{1.2} \times PFP$ and ΔO_3 found using results from the four GCMs with interactive chemistry is somewhat lower (i.e., about 20 to 25% at end of century) than that found for the other 16 CMIP6 GCMs, but not appreciably different (see Methods). Therefore, we conclude that GCMs with full stratospheric chemistry exhibit similar behavior for future winter, Arctic stratospheric temperature as the other GCMs.

Furthermore, whereas our new Figure 9 includes the multi-model GCM ensemble mean values of $EESC^{1.2} \times PFP$ from all 20 of these CMIP6 GCMs, we include in Supplemental Figure 17 an alternate version that shows $EESC^{1.2} \times PFP$ from the four GCMs with interactive chemistry, to support the statement noted above.

2. In the method section the authors claim that "...while the shape of the probability density functions ... are not strictly Gaussian, the fall-offs of the tail of both functions are Gaussian-like" (382) Is this quantitatively supported? Have the authors computed the Pearson index to evaluate the kurtosis of the distribution?

We have computed the kurtosis of the distribution of the Monte-Carlo simulations of the quantities S_{PFP-LM} and $S_{PFP-LM}/\Delta S_{PFP-LM}$ for both the reanalysis products and output of each GCM. For the re-analysis products, the S_{PFP-LM} and $S_{PFP-LM}/\Delta S_{PFP-LM}$ have a range of 2.1 to 2.8 and 3.1 to 4.1, respectively.

For the CMIP5 output, S_{PFP-LM} and $S_{PFP-LM}/\Delta S_{PFP-LM}$ have mean values of 3.0 and 2.8, respectively, with a range of 1.8 to 5.1 and 1.9 to 4.9. For CMIP6 output, S_{PFP-LM} and $S_{PFP-LM}/\Delta S_{PFP-LM}$ both have mean values of 3.2 and 2.8, respectively, with a range of 1.6 to 6.7 and 1.6 to 4.8.

We have changed the text in Methods which is related to ERA5/ERA5.1 to read:

While the shape of the probability density functions of S_{PFP-LM} and $S_{PFP-LM}/\Delta S_{PFP-LM}$ are not strictly Gaussian, the fall-offs of the tail of both functions are Gaussian-like (i.e., kurtosis close to 3; more specifically, the kurtosis for S_{PFP-LM} is 2.1 and for $S_{PFP-LM}/\Delta S_{PFP-LM}$ is 3.4). Therefore, we are comfortable assessing better than 2σ confidence to $S_{PFP-LM}/\Delta S_{PFP-LM}$ since 8×10^{-5} is so much less than 0.05, the 2σ confidence marker for a strictly Gaussian distribution.

3. The authors focus on the detection of trends in the occurrence of the coldest vortexes. These trends may arise when the time series is increasing its variance, its (running) average, or both.

It would be of interest to look more carefully at that. As instance they may apply (a sort of) ISA method to detect trends in the evolution of the warmest winters. Are they becoming less warm, as it would seem by taking the time series at face value, or is the system simply more variable? I think it would be interesting to compare the trends of the coldest and warmest winters.

Much thanks for this suggestion.

We have added the following text starting on line 202:

The GCM simulations for SSP5-8.5 shown in Figs. 4 and 5 also show a tendency for PFP associated with the warmer Arctic winters (bottom of the data envelope) to rise over time, a projected trend that is not yet apparent in observations⁴⁷ due perhaps to the generally lower values in the observational period and the lower limit of zero.

with a citation to:

⁴⁷Rieder, H. E., Polvani, L. M. & Solomon, S. Distinguishing the impacts of ozone-depleting substances and well-mixed greenhouse gases on Arctic stratospheric ozone and temperature trends. *Geophys. Res. Lett.* 41, 2652–2660 (2014).

As stated above, the paper is well-laid and clear, its results are of interest for a vast community, so it surely deserves publication after addressing these few remarks.

Much thanks; we hope the revised paper will be evaluated in a similar manner.

Reviewer #2 (Remarks to the Author):

Review of the manuscript “The influence of climate change on chemical loss of ozone in the Arctic stratosphere” by Gathen et al.

This study challenges the view expressed, e.g. in Rieder and Polvani (2013) (RP13) and Rieder et al. (2014), that anthropogenic well-mixed greenhouse gases do not play a significant role in the past and projected evolution of lower-stratospheric polar temperature and ozone depletion, which are, instead, controlled mainly by the concentrations of ozone depleting substances. Specifically, through a careful analysis of the diagnostic PSC formation potential (PFP), this study provides evidence from reanalyses and model simulations for the “cold Arctic winters are getting colder” paradigm consistent with Rex et al. (2006) among others. Furthermore, it demonstrates that the upward trend in the local maximum PFP projected for the 21st century correlates very well with radiative forcing due to well-mixed GHG. The authors argue that the analysis presented in RP13 was not sound from the statistical perspective and that it was overly simplistic in that it was done on the entire zonally symmetric polar region, which cannot be identified with the highly variable polar vortex where ozone depletion occurs.

The paper is very clearly written and easy to follow. The methods section provides all the definitions and calculations needed to reproduce the results and contains helpful discussions that justify the choices made by the authors (primarily the use of the iterative selection approach). Statistical significance estimation is discussed in detail.

Much thanks for these kind words.

I find the argument pertaining to the long-term PFP behavior convincing and interesting. It certainly makes for a valuable contribution to the debate. However, the connection between the “coldest Arctic winters are getting colder” finding and the future of ozone is entirely speculative in the paper, necessitating the use of phrases such as “could conceivably” (L206) and “may have been” (the abstract). See my general comment (2) for some discussion. This is disappointing, because at the same time, the prominent role that the subject of ozone recovery plays in Concluding Remarks suggests that it is meant as the central motivation for this work. I can think of two ways to remedy this: (1) if the connection with ozone is made more solid and quantitative or (2) if it is explicitly relegated to future work (it would be hard not to mention ozone at all!) and the speculative discussion is dropped. In my opinion, the PFP results are important and interesting enough to stand on their own.

We have made major revisions to the paper in light of this comment, and similar words from Reviewer 3. New figures 2, 7, 8, and 9 are used to highlight how the chemical loss chemical loss of column ozone in the Arctic (\$\Delta O_3\$ ) will respond to future values of the PSC formation potential (PFP), Equivalent Effective Stratospheric Chlorine (EESC), as well as future trends in stratospheric \$H_2O\$ induced by rising \$CH_4\$ and the thermodynamics of the tropical tropopause according to various SSP scenarios.

General comments

1. The paper explicitly disputes the findings of Rieder and Polvani (2013) (RP13). In contrast to RP13, the approach here is more refined in two ways; (1) only V_{PSC} and the related diagnostic, PFP inside the polar vortex is considered (RP13 base their analysis on data from latitudes $\geq 60^\circ N$ that don't necessarily coincide with the vortex, especially in the NH). (2) the use of ISA instead of VAS. However, RP13 note: “To confirm this, we have computed trends in several different ways and found nothing that is statistically significant on a rigorous test level (i.e., 95%). Specifically, we have computed the trend over (1) the entire time series, (2) the first and second halves separately, and (3) 5 year and 3 year maximum and minimum values. The p values for all of these are reported in Table S1: no trends are significant on a 95% level.” Indeed, comparing the results here (Figs 4 and 4) and Figure 3 in RP13 that shows extended wintertime V_{PSC} from a CAM ensemble, there is no clear trend in the latter, independent of how the maxima are selected. I can think of at least three (not necessary mutually exclusive) possibilities: (1) the dynamical response to increasing GHGs in CAM is fundamentally different than in the models discussed here; (2) the use of $60^\circ N$ - $90^\circ N$ range in RP13 (as opposed to the

vortex interior) leads to trend cancelations; (3) It's because the metric is different and the trends seen in PFP^{LM} reflect a decrease in the vortex volume rather than an increase in V_{PSC} . It would be nice to see some discussion of this. I admit I find this discrepancy puzzling. While I absolutely agree that 60°N-90°N cannot be identified with the polar vortex in general, is it really such a bad vortex proxy during very cold winters when the vortex is near-zonal and relatively undisturbed?

Point 1:

RP13 performed several analyses. In the beginning they repeated the studies on V_{PSC} of Rex et al., 2004, 2006 (R04, R06) on observational data. R04 and R06 used in addition to ECMWF analysis products the FU Berlin data analysis based on radio soundings (Fig. 3 in R04 and Fig. 4 in R06). The use of the radio sounding data allowed them to cover the time period from 1966 to 2003/2005, i.e. 38/40 winters. RP13 criticized the inhomogeneity in the used data sets of R04 and R06 and used instead the more homogeneous NASA-MERRA, ERA-Interim, and NCEP-NCAR data sets. Those data sets covered the time period 1980 to 2011, i.e. 32 winters. RP13 did report a positive trend in V_{PSC} in their data sets, but it was not statistically significant. Therefore, they stated "... data sets show no trends in the volume of polar stratospheric clouds ...". In our humble opinion a better statement would have been: "... show no significant trends ...". Using the more homogeneous products (RP13) instead of the merged data set (R04, R06) had the cost of using less winters leading to a low number for their statistics, an issue not been addressed by RP13.

In our new study we are able to use homogeneous data sets comparable to those of RP13 and more important again a longer time period (1965/80/81 – 2020) with 56/41/40 winters including two more record breaking winters. Therefore, it is not surprising that the trend calculation got significant again. In response to this person's ATOVS discussion we have computed trends for almost halved parts of the data sets, now (see below).

Point 2:

We have made a concerted effort to obtain CMAM output to assess the three possibilities noted above, but unfortunately the only archived results of T, u-wind, and v-wind from the Canadian Middle Atmosphere Model (CMAM) are sporadic with respect to time.

We exchanged a series of emails with Charles McLandress (Univ of Toronto), Ted Shepherd (Univ of Reading) and David Plummer (Environment and Climate Change Canada) during September 2020. This led to our access of archived output from the CMAM model from http://data.ceda.ac.uk/badc/ccmval/data/CCMVal-2/Sensitivity_Runs/SCN-B2b

The archived data span the years 1960 to 2098, but cover only one time step in a range of three successive days followed by large gaps until the next recorded time step. The highest density of archived model output is for these three ranges:

First-last day of a month, 1st and 2nd day of the following month.

Second-10th, 11th, 12th of each month.

Third-20th, 21th, 22th of each month.

However, even this above mentioned level of coverage (three dates per month for all six winter months) is only provided for 15 winters: 1990/91 to 2004/05.

Archived output for all other winters have additional gaps in time, with some repetition every three winters, for instance:

Winters 1960/61, 1963/64, 1966/67 have output for Nov/Dec and no output for Jan - Apr

Winters 1961/62, 1964/65, 1967/68, have only one data point (1 Nov)

Winters 1962/63, 1965/66, 1968/69, have data for 1 Nov, no other data in Nov and Dec, followed by data for Jan - Apr.

All three of the CMAM runs on this CEDA archive show the same temporal gaps. We attempted to obtain higher temporal resolution output via email exchange with the above mentioned persons, but these fields had not been archived.

Point 3:

On many days during Arctic winter, such as this map of total column ozone on 15 March 2020 obtained from <http://www.cpc.ncep.noaa.gov/products/stratosphere/temperature/50mbnhlo.png>:

the vortex is so disturbed that, yes, one would expect major differences between quantities calculated within the boundary of the polar vortex (evident from the ozone contours) and an average between 60° and 90°N. We can easily envision that this total ozone pattern is a result of the Brewer Dobson circulation whose last downward wing more or less stopped before the vortex boundary, leading to two completely different temperature regimes inside and outside of the vortex. Therefore, our best guess is: yes, polar cap temperature means can be such a bad vortex proxy even during very cold winters when the vortex is near-zonal and relatively undisturbed. One should keep in mind that the largest extent of the Arctic polar vortex covers only about half of the polar cap region. Therefore, it seems to us that it is the duty of those who use an average poleward of 60° N as a proxy to address the validity of this approach. Quantification of differences between vortex averaged quantities and 60° to 90°N quantities is an important research task, but one we feel lies outside the scope of our paper.

2. This one is crucial. While the analysis of PFP^{LM} trends using the ISA method appears solid (while reconciliation of it with RP13 may not be quite sufficient, see (1)), the connection that the paper makes with the future of polar ozone is tenuous. The authors demonstrate, assuming continuing GHG increase, that if a very cold winter does happen in the future it will likely be colder than in the past, and the polar vortex will host more PSCs. That's a solid conclusion from the CMIP5/6 results but the following sentence from the concluding remarks section, that justifies the importance of this study remains unsubstantiated: "Should stratospheric chlorine levels continue to decline at a slower than expected rate, the GHG-driven tendency for the coldest Arctic winters to get colder over time could conceivably drive even larger chemical loss of Arctic ozone than has previously occurred." What is meant by this? That "Arctic ozone holes" will occasionally continue to happen?

How frequently? That polar ozone trends will be affected? That trend detection will be affected? Polar ozone loss is influenced by a number of factors, of which this paper discusses only one. To be precise about this the authors would have to actually show what happens to ozone under, say SSP5-8.5 scenario together with some pessimistic ODS scenario. What about the frequency of strong/cold vortex events? What about projected changes to ozone resupply from the enhanced Brewer-Dobson circulation under increased GHG? The paper does mention the recent slowdown of the decline of halogens but its consequences are discussed only speculatively. By comparison, Keeble et al., 2020

demonstrates that the worst-case scenario (in some sense) is a 10-year delay in ozone recovery but not so much in the NH high latitudes. (their Figs 8 & 9). Other studies talk about the possibility of “super-recovery” (WMO 2018) resulting from higher GHG concentrations in the future, particularly at midlatitudes upper stratosphere. One might expect that this would also enhance polar springtime ozone through resupply. Even if we accept the hypothesis that the extreme ozone loss in 2020 is at least in part attributable to climate change (which is possible but not demonstrated in the paper) that does not automatically imply that an even colder winter a decade from now will have the same effect on ozone. Perhaps regressing Arctic ozone loss or vortex ozone content on PFP and halogen loading could help substantiate the connection with ozone.

Great comment. This critique is addressed by our new analysis of trends in the quantity $EESC^{1.2} \times PFP$, which as noted in the opening paragraph of our reply, serves as a good proxy for the chemical loss of column ozone within the Arctic vortex. We also quantify the impact of trends in stratospheric H_2O induced by rising CH_4 and the thermodynamics of the tropical tropopause, in terms of both PFP and $EESC^{1.2} \times PFP$. Most importantly, we now use the global ATLAS Chemistry and Transport Model that includes a comprehensive treatment of stratospheric chemistry (Wohltmann and Rex, GMD, doi:10.5194/gmd-2-153-2009, 2009), along with our analysis of ozonesonde data to show that the chemical loss of column ozone in the Arctic (ΔO_3) is well described by the quantity $EESC^{1.2} \times PFP$ (new Fig 1b). New Figures 2 and 7 and the associated text describe various aspects of the trends in $EESC^{1.2} \times PFP$, and new Figures 8 and 9 show how $EESC^{1.2} \times PFP$ computed from the ensemble average of CMIP6 GCM output, for four SSP scenarios.

We have added the following text starting on line 334 to address the Brewer Dobson Circulation (BDC):

Projections of ΔO_3 shown in Figs. 8 and 9 represent the seasonal loss of column ozone that may occur for various GHG scenarios, rather than resulting column ozone. Future levels of Arctic column ozone during late winter and early spring are expected to increase due to factors such as intensification of the BDC, upper stratospheric cooling, and possible changes in planetary wave activity that will exert a strong influence on the abundance of column ozone within the Arctic vortex during its formation in early winter^{13,59}. Future total column ozone during spring will reflect a balance between the initial abundance and chemical loss.

with citations to:

13. Bednarz, E. M. *et al.* Future Arctic ozone recovery: the importance of chemistry and dynamics. *Atmos. Chem. Phys.* **16**, 12159–12176 (2016).
59. Dhomse, S. S. *et al.* Estimates of ozone return dates from Chemistry-Climate Model Initiative simulations. *Atmos. Chem. Phys.* **18**, 8409–8438 (2018).

Specific and technical comments

LL17-18. This sounds like an attribution result, but no quantitative attribution analysis is presented.

This text, in the original abstract, had read:

Extremely cold conditions in the Arctic during winter 2020 that resulted in extensive regions of low total column ozone may have been driven in part by the build-up of anthropogenic GHGs over the past half-century.

The abstract has since been revised, and no longer contains this sentence.

LL 20-22 If you say “interannual variability” do you need to say “for particular winter-spring” seasons? This reads a bit strange to me. Maybe I misunderstood something.

Original text had read:

Chemical loss of stratospheric ozone in the Arctic polar vortex due to anthropogenic halogens exhibits large inter-annual variability for particular winter-spring seasons (hereafter winter), driven by variations in meteorology.

We have modified the text to read:

Chemical loss of stratospheric ozone in the Arctic polar vortex due to anthropogenic halogens during winter-spring seasons (hereafter winter) exhibits large inter-annual variability, driven by meteorology.

LL. 42-43: Wohltmann et al. doesn't appear to be a published paper or the reference given is insufficient to find it. Note that there is this new paper that discusses 2020 Arctic ozone: Manney et al. (2020).

Since the Wohltmann *et al.* paper is now published, and the lead author of this paper has joined our author team, we have retained this citation. The Manney *et al.* (2020) is also an excellent manuscript that will receive a large number of citations. Since *Nature Communications* limits us to the use of 70 total citations, we have decided to only cite the Wohltmann *et al.* paper. We would be very happy to cite Manney et al. in addition, if the number of 70 is not taken as critical by the editor!

L 58. "since this is straightforward to describe and implement" is not a good enough reason (L220 explains it better, though). I would just drop this sentence and simply refer the reader to Methods.

Great suggestion; we have changed this sentence to read:

Here, we use a constant volume mixing ratio of stratospheric H₂O equal to 4.6 parts per million (ppmv) and a profile of HNO₃, both based on satellite observations, to find T_{NAT} (see Methods).

L62-65. Correct but then one (not me!) might ask: is it OK to use the reanalyses? The same reference (Lawrence et al., 2018) may be of help here as it provides specific recommendations for the use of polar processing diagnostics derived from reanalysis, based on a very thorough vetting process. That paper doesn't talk about ERA5 but the recommendations still apply, I think, and could be referenced very briefly (use modern reanalyses, use a number of them, not just one, etc., which is what you, in fact, do) There's also an important caveat here: stratospheric temperature-derived diagnostics (such as PFP) are more reliable after 1998/1999 when the reanalyses start assimilating AMSU-A data (see also Long et al., 2017). Also note the following cautionary note from Lawrence et al. (2018): "Since many of the major changes in data inputs are made at approximately the same time in each reanalysis, the agreement or lack thereof between the reanalyses does not provide the information to assess the degree to which these changes are caused by changes to the assimilated observations. We thus emphasize here that (...) use of reanalyses in trend studies should be regarded with skepticism and only attempted after rigorous assessment of the relationships of temperature changes to observations assimilated." I don't think this means that one absolutely shouldn't look at trends derived from reanalyses and that those trends are necessarily artifacts of observing system changes (in fact we know that in many instances that is not the case, and reanalyses have gone a long way toward being "trend-friendly" scientific tools; I myself have been known for using them for trend studies in some limited context) but it would be good to justify the validity of what you're doing here a little more. I realize this is a tall order for a diagnostic for which we don't have direct measurements, but a few words of justification would go a long way. One simple suggestion: repeat the trend calculations for the period 1999-2020. ATOVS radiance data during that recent period provide a strong constraint and studies don't show significant spurious discontinuities in stratospheric temperatures in the ATOVS period (Long et al. 2017, Lawrence et al., 2018). Statistical significance will go down, of course, but it may be instructive to compare what you get with the trends computed over the 40-year period. This comment may sound like I fundamentally mistrust reanalyses. This is not the case. But being close to the reanalysis world I have a tendency to be extra careful in this area, that's all.

The original text reads:

Diagnostics for the formation of PSCs can vary substantially between reanalyses, such that conclusions based on the often marginal conditions for the formation of PSCs in the NH could be affected by small differences among the reanalyses²⁵

with citation to:

²⁵Lawrence, Z. D., Manney, G. L. & Wargan, K. Reanalysis intercomparisons of stratospheric polar processing diagnostics. *Atmos. Chem. Phys.* 18, 13547–13579 (2018).

Upon revision, this text was added to the paragraph in which the original text noted above appears (line 91):

Finally, reanalyses transition from the use of space-borne data from SSU and TOVS to AMSU and ATOVS systems in the 1998 to 1999 timeframe²⁷. We obtain similar results for trends in PFP^{LM} (differences within respective uncertainties) when considering data obtained prior and after this transition (see Methods).

with citation to the same Lawrence et al. paper.

New Supplementary Figure 7 has been added to support this statement, and the section in Methods that describes this figure reads as follows:

SSU & TOVS versus AMUS & ATOVS. In the paper we state that similar results are obtained for trends in PFP (differences within respective uncertainties) when considering space-borne temperature obtained from SSU and TOVS systems versus AMSU and ATOVS systems. The transition occurred in the years 1998 to 1999²⁷. Supplementary Figure 7 shows similar results for trends in PFP^{LM} (differences within respective uncertainties) when considering data obtained prior and after this transition.

L66 The canonical reference for ERA5 is Hersbach et al (2020).

Much thanks; this change has been made. This article appeared after our original submission to NCC.

L85. “because these days encompass the full range of PSC activity among all meteorological fields” I’m struggling to understand what that means. Please, consider revising. In addition, I presume these dates were chosen because significant amounts of PSCs occur in that period (supplementary Fig. 1). But is that true just for the historical period or for the 21st century simulations as well? If, for example, there were significant PSCs in some of the simulations outside of that time window that could distort the results.

Again, much thanks.

We have revised the period of temporal integration to be 1 April to 30 November, and have modified the sentence starting on line 103 to read:

1 November (prior year) and 30 April (specified year) are used as limits of integration because these dates encompass the full range of PSC activity among reanalysis and GCM-based temperature fields.

L94 What does “accurate” mean in this context? I realize this is explained in Methods but it would be good to have a sentence of explanation here.

The original text had read:

Since publication of these papers, we have developed a demonstrably more accurate and robust trend detection procedure (see Methods) termed the Iterative Selection Approach (ISA).

We have modified the text starting on line 150 to read:

Since publication of these papers, we have developed a more accurate and robust trend detection procedure as documented by a series of Monte-Carlo (MC) simulations (see Methods), termed the Iterative Selection Approach (ISA).

L114. Personally, I would change that title to something that reflects that this section is about 20-21st century model simulations. “Computed PFP” suggests that the PFP in the preceding section wasn’t computed).

The section title had been:

Computed PSC Formation Potential

and has been changed to:

PSC Formation Potential from GCMs

LL127-130. Nice!

Thanks!

L146. But dependent on the future trajectory of the concentrations of those halogens!

Great comment. This point is now addressed by our analysis of trends in the quantity $EESC^{1,2} \times PFP$, which constitutes a major revision to the paper.

LL152-155. Or the reanalyses may overestimate the trends due to discontinuities induced by step changes in their observing systems

The original text had read:

The observationally-based trend lies near the upper 1σ value of the GCMs. Over this shorter period internally generated climate variability may play a substantial role and during these four decades the one realisation of climate that developed in earth's climate system may have coincidentally followed a path that led to values of S_{PFP-LM} at the upper range of the frequency function from the GCMs.

Based on the work we have conducted in reply to the comment on L62-65, it seems unlikely that the difference between the observationally-based trend and the GCM-based trend in S_{PFP-LM} is caused by discontinuities in the observing system. We hope the changes to the paper in reply to the comment on L62-65 adequately address this comment. We have decided to not alter these two sentences.

LL157-160. Nice. I would expand it just a tad: "...role of internal variability and uncertainties in trends derived from reanalyses..." or something to that effect.

Thanks for the suggestion. Alas, this section has been significantly modified, in response to Major Comment 2 of Reviewer 3, as well as this person's comment on line 158.

If you don't mind, we would appreciate if you could have a look at our reply to Major Comment 2 and comment about line 158, both of Reviewer 3, for a detailed reply.

LL163-167. Nice!

Thanks!

LL170-175 and LL410-418. I suggest making a stronger connection here. Why is the representation of the QBO a good metric for polar processes (e.g. Holton-Tan mechanism)? Also please add some references for QBO in CMIP (the SPARC-QBOi activity: Butchart et al, 2020, Richter et al., 2020a, b). For example, Richter et al. (2020b) performed a detail comparisons of the QBO representation in different versions of CMIP, including 5 and 6. They state "Despite the tripling of the number of models that are able to simulate the QBO, the quality of the simulation of the QBO from CMIP5 to CMIP6 has not changed." LL410-412 state that the latter simulate the QBO better than the former. Looking at Supplementary Figures 11 and 12 this appears to simply reflect the fact that most of the CMIP5 didn't generate any QBO at all and that situation improved in CMIP6. If the correct representation of the QBO is important then wouldn't it make sense to separate the results shown in Figs 3 and 4, and supplementary Figs 8 and 9 into groups of models that do and do not generate the QBO? For example, a cursory comparison of Fig. 3 and supplementary figure 12 reveals that the models that produce PFP^{LM} trends in excess of 2 d/decade are those without the QBO, and two of them have a very low top.

We have used two of the suggested references (Butchart et al, 2020 and Richter et al., 2020b) to revise the text, starting on line 303, to read:

There are a number of factors that affect the accuracy of lower stratospheric temperature within GCMs, such as the maximum altitude and vertical resolution⁵⁵ as well as model representation of planetary wave activity that transports energy from equatorial to poleward regions⁵⁶. One important marker of the usefulness of a GCM to simulate stratospheric dynamics is whether the model generates an oscillation of the direction of the zonal wind in the tropical lower stratosphere with a period of about 28 months, known as the quasi biennial oscillation (QBO)⁵⁶. Our examination of the tropical zonal wind from CMIP5 and CMIP6 GCMs indicates that CMIP6 GCMs tend to provide a better representation of the QBO than was evident in output from CMIP5 GCMs (see Methods), consistent with the more formal analysis of Richter et al⁵⁷. We see little difference in our Figs. 8 and 9 when the CMIP6 GCM output is examined in groups of models that provide a reasonable representation of the QBO versus other models (see Methods). Richter et al.⁵⁷ note that while the number of models with an internally generated QBO has increased substantially from CMIP5 to CMIP6, the multi-model mean amplitude for atmospheric levels below a pressure of 20 hPa is still much lower than observed. Given the importance of the QBO in stratospheric dynamics, substantial effort is being directed towards improving the representation of this process within GCMs⁵⁸.

with citations to:

⁵⁵Charlton-Perez, A. J. *et al.* On the lack of stratospheric dynamical variability in low-top versions of the CMIP5 models. *J. Geophys. Res. Atmos.* **118**, 2494–2505 (2013).

⁵⁶Lott, F. *et al.* Kelvin and Rossby-gravity wave packets in the lower stratosphere of some high-top CMIP5 models. *J. Geophys. Res. Atmos.* **119**, 2156–2173 (2014).

⁵⁷Richter, J. H. *et al.* Progress in Simulating the Quasi-Biennial Oscillation in CMIP Models. *J. Geophys. Res. Atmos.* **125**, e2019JD032362 (2020).

⁵⁸Butchart, N. *et al.* QBO changes in CMIP6 climate projections. *Geophys. Res. Lett.* **47**, e2019GL086903 (2020).

LL199-200. See my general comment 2.

The original text had read:

Stratospheric chlorine has declined in a slower than expected manner due to illegal recent production of CFC-11 (ref. ⁴²) that appears to be originating, at least in part, from eastern Asia⁴² and China⁴³

Upon revision we have decided to remove any mention of the violation of the Montreal Protocol, in part because this discussion is not truly central to the thesis of the paper, and also because one of us (RS) recently reviewed a manuscript, to be published in Nature sometime soon, that indicates the violations of the Montreal Protocol within eastern Asia ceased around the start of 2019 (please note this is confidential information).

LL200-201. The “and” there sounds a little awkward; China is in eastern Asia. Please, consider revising.

The text in question has been removed.

LL206-207. There are several trends happening here with opposite impacts on ozone: ODS declining, cold winters getting colder, etc. It is not at all clear which wins and under what scenarios. See my general comment 2.

Please see our reply to Major Comment 2 for a fuller explanation. Long-story short, we address this comment with by analyzing trends in EESC^{1,2} × PFP, which involves four new figures in main (Figs 2 and Figures 7 to 9), new text spanning lines 261 to 301, as well as a number of figures in the Supplement.

L215. Is the assumption of constant H₂O valid for the 21st century projections? How sensitive are TNAT to possible water vapor increase due to, e.g. increasing upper stratospheric methane?

Another great point. Much thanks.

The additional space available under *Nature Communications* (again, our original paper was a “pass through” of a *Nature Climate Change* submission has) allowed a thorough quantification of the effect of rising stratospheric H₂O on PFP and inferred chemical loss of Arctic O₃.

The issue of time dependent water is first described on line 235 of the revised paper, and constitutes parts of Fig. 2, all of Figs. 7, 8, and 9, as well as numerous figures in the Supplement.

LL 329-329. “13.6 d decade⁻¹”. Aren’t the units of PFP days, [d]?

Great catch!

LL260-266. Supplementary Figure 5 shows a rapid increase in maximum frequencies starting in the mid-1990s. Can you comment on that? Is the temporal resolution of the sonde launches uniform throughout the entire period? Was there a change in instrumentation? Also, are the Sodankylä sondes assimilated in JRA-55?

We believe the reviewer is commenting on Supplementary Figure 2 of the submitted paper, rather than Supplementary Figure 5.

The rapid change in maximum frequencies displayed in Supplementary Figure 2 starting in the mid-1990s cannot be explained by changes in instrumentation. The temporal resolution of the sonde launches was uniform throughout the entire period. We have analysed data obtained at 00 UT and 12 UT for this figure.

At Sodankylä, Vaisala RS80 radiosondes were used for temperature measurements from 1982 to 1999.

The radiosonde models and temperature sensors used in Sodankylä since 1965 are listed below:

Period	Model	Temperature sensor
1965-1966	RS12	bimetal strip
1967-1975	RS13/RS15	bimetal strip
1976-1981	RS18	bimetal ring
1981	RS21	bimetal ring in a duct
1982-1999	RS80	capacitive sensor
2000-2004	RS90	capacitive sensor
2005-2017	RS92	capacitive sensor
2017-	RS41	resistive platinum sensor

All radiosondes were manufactured by Vaisala. Although the radiosonde models have changed over time, however the effect on measured temperature at 50 hPa has been minimal. Gaffen (1994) <https://agupubs.onlinelibrary.wiley.com/doi/abs/10.1029/93JD03179> reported small biases at 50 hPa when comparing various historical radiosonde models (Vaisala RS12-80). The differences were found to be smaller than 0.2 K between the radiosonde models RS12-15, RS18 and RS80 at 50 hPa (Gaffen 1994, Figure 4). The later models of radiosondes have an improved capacitive temperature sensor or resistive platinum sensor (RS41). RS90 and RS92 have both the same type of temperature sensor, which is smaller than the previously used sensor on RS80. However, the changes from RS80 to RS92 have not introduced significant temperature biases at 50 hPa. Dirksen et al. (2014) <https://amt.copernicus.org/articles/7/4463/2014/amt-7-4463-2014.pdf> found that the uncertainty in RS92 night-time temperature measurements is 0.15 K. Hence, we are confident the time series of PSC temperature record over Sodankylä show in Supplementary Figure 2 of the submitted paper is correct. We have added a few sentences to the revised Supplement, starting at line 402, that read:

As an important check on the temporal integrity of the JRA-55 reanalysis prior to 1979, in Supplementary Fig. 3 we show an update to the radiosonde temperature time series acquired at Sodankylä, Finland for each winter since 1965 (ref. ⁶⁶). This figure shows the time evolution of the percentage of observations of temperature < -77.9°C at 50 hPa over the months of December (prior year) and January, February, and March (indicated year) from regular radiosonde launches from Sodankylä. Supplementary Fig. 3 supports our conclusion, shown in Fig 3e of the main article, that conditions conducive for the formation of PSCs were rarely present between 1965 and 1979, the start of the modern satellite era.

with citation to:

⁶⁶Kivi, R., Kyrö, E., Turunen, T., Ulich, T. & Turunen, E. Atmospheric trends above Finland: II. Troposphere and stratosphere. *Geophysica* 35, 71–85 (1999).

to address this comment.

Finally, the Sodankylä radiosonde data are assimilated into JRA-55. As noted in Haimberger, J. *Clim.* 2007 <https://journals.ametsoc.org/view/journals/clim/20/7/jcli4050.1.xml>, data from the Sodankylä are part of the RAOBCORE (Radiosonde Observation Correction Using Reanalyses) dataset, that is used by JRA-55. We show a timeseries of temperature from Sodankylä radiosondes in the Supplement because the long-term integrity of Arctic stratosphere temperature is central to our study.

L264. launched  “launches”(?)

Change made; thanks!

L270. “criteria” “criterion”

Change made; thanks for the very careful read.

Supplementary material

L22. Comparing with Fig.3 in the main text it seems to me that it’s the other way around: the lower panels are with adjustments. Am I misreading this figure?

Change made; great attention to detail. Much thanks.

L22. “and adjustment”  “an adjustment”

Change made; again, thanks for the very careful read.

References

Butchart, N., Anstey, J. A., Kawatani, Y., Osprey, S. M., Richter, J. H., & Wu, T. (2020). QBO changes in CMIP6 climate projections. *Geophysical Research Letters*, 47, e2019GL086903. <https://doi.org/10.1029/2019GL086903>

Hersbach, H, Bell, B, Berrisford, P, et al. The ERA5 global reanalysis. *Q J R Meteorol Soc.* 2020; 1- 51. <https://doi.org/10.1002/qj.3803>

Keeble, J., Abraham, N. L., Archibald, A. T., Chipperfield, M. P., Dhomse, S., Griffiths, P. T., and Pyle, J. A.: Modelling the potential impacts of the recent, unexpected increase in CFC-11 emissions on total column ozone recovery, *Atmos. Chem. Phys.*, 20, 7153-7166, <https://doi.org/10.5194/acp-20-7153-2020>, 2020.

Long, C. S., Fujiwara, M., Davis, S., Mitchell, D. M., and Wright, C. J.: Climatology and Interannual Variability of Dynamic Variables in Multiple Reanalyses Evaluated by the SPARC Reanalysis Intercomparison Project (S-RIP), *Atmos. Chem. Phys.*, 17, 14593-14629, 2017.

Manney, G. L., Livesey, N. J., Santee, M. L., Froidevaux, L., Lambert, A., Lawrence, Z. D., et al.. (2020). Record-low Arctic stratospheric ozone in 2020: MLS observations of chemical processes and comparisons with previous extreme winters. *Geophysical Research Letters*, 47, e2020GL089063. <https://doi.org/10.1029/2020GL089063>

Richter, J. H., Butchart, N., Kawatani, Y., Bushell, A. C., Holt, L., Serva, F., Anstey, J., Simpson, I., Osprey, S., Hamilton, K., Braesicke, P., Cagnazzo, C., Chen, C.-C., Garcia, R., Grey, L., Kerzenmacher, T., Lott, F., McLandress,

C., Naoe, H., Scinocca, J., Stockdale, T., Versick, S., Watanabe, S., Yoshida, K., & Yukimoto, S. (2020a). Response of the Quasi-Biennial Oscillation to a warming climate in global climate models. *Quarterly Journal of the Royal Meteorological Society*, 1-29. <https://doi.org/10.1002/qj.3749>

Richter, J. H., Anstey, J. A., Butchart, N., Kawatani, Y., Meehl, G. A., Osprey, S., & Simpson, I. R. (2020b). Progress in simulating the quasi-biennial oscillation in CMIP models. *Journal Geophysical Research: Atmospheres*, 125, e2019JD032362. <https://doi.org/10.1029/2019JD032362>

WMO (World Meteorological Organization) (2018). *Scientific Assessment of Ozone Depletion: 2018*. Global Ozone Research and Monitoring Project-Report No. 58, 588 pp., Geneva, Switzerland.

Much thanks for this nice list of citations. Upon revision, we have added citations to those papers marked with blue font, as well as a different Keeble et al. paper.

Kris Wargan

Thanks Kris! We sincerely appreciate the time and effort you have put into this review, and have added our thanks to the acknowledgements.

Reviewer #3 (Remarks to the Author):

Review of "The influence of climate change on chemical loss ..."

General

Let me start with my overall conclusion first: this is a strong paper that deserves to be published in Nat. Commun. - the paper finds that conditions allowing heterogeneous chlorine activation and thus polar ozone loss are increasingly favourable for ozone loss over the past four decades. As a result, increasingly severe Arctic ozone loss has occurred over these time scales. In simple words, the message of the paper is that cold Arctic winters are getting colder.

This time evolution of Arctic temperatures is analysed based on reanalyses from four meteorological centres and on the results from a large number of general circulation model simulations. The paper concludes that stratospheric cooling caused by human release of greenhouse gases likely drives the change in Arctic temperature patterns and thus the increasingly severe Arctic ozone loss. These results are clearly of significance to the field of stratospheric ozone loss, but also have various implications for the science of global change (see also discussion below).

Earlier work in this field (e.g. Tilmes et al., 2006; Rex et al., 2004, 2006) had used different data sets (that are partly superseded by more modern products); a strong point of the recent study is the consistent use of reanalysis schemes from four different centres. I think one can say that the currently best available data sets have been employed here. An important aspect is further the use of the results from general circulation model simulations. In this way, the authors can make an attribution statement about the likely cause of changes in Arctic temperature patterns and chemical ozone loss. This is important.

Overall, the methodology is sound and certainly meets the expected standards in the field. I have a few points (see below) where I think the paper would gain from a somewhat extended discussion. (I would also suggest providing more results in tabulated form (electronically) to make the use and reproduction of the present results easier.) I think this can be done.

Thanks for these kind words. We have made a concerted effort to address the excellent points raised in this review. We have placed tabulated data on the web electronically in the data center Zenodo, with the identifier <http://doi.org/10.5281/zenodo.4414823>.

A point, where I am critical is the focus on the cold winters. While the cold winters are relevant for Arctic ozone loss, the investigation of the causes of the variability and the attribution of trends to increasing concentrations of greenhouse gases requires a look at the warmer winters too. How are the warmer winters affected by greenhouse gases and how relevant are changes in stratospheric wave activity? (See discussion below.)

Good point. Since these same thoughts are provided with considerably more detail below in major point 2, we'll provide a detailed reply there.

Finally, I am citing references in this review to support the points I am making; I do not mean to suggest that those references should be cited in the manuscript.

Thanks for providing so many suggested citations. We regret that we cannot cite all of the suggested papers, because Nature Communications has a limit of 70 total citations, but some of the papers noted below have been added.

In summary, although I am suggesting some revisions, this paper deserves to be published in Nat. Commun.

Much thanks!

Discussion

We have taken the liberty to add numbers and highlight the reviewers leading phrases in bold-face.

1. Polar stratospheric cloud (PSC) formation potential: PSC formation potential (PFP)

The analysis here is based on a particular way of calculating PFP. However there are other alternatives which should be discussed. Tilmes et al. (2007, 2006) used a quantity referred to as ‘potential for the activation of chlorine’, which is similar to PFP but uses a threshold temperature for chlorine activation on cold sulphate aerosol (Drdla and Müller, 2012) and is a function of temperature, water vapour, and the surface area density of stratospheric sulphate. The potential for the activation of chlorine is thus more dependent on the stratospheric sulphate loading and thus to the occurrence of volcanic eruptions, although for some conditions it is comparable to PFP. It would also be possible to take into account the changing stratospheric halogen loading when defining a quantity like PFP, which might be helpful, when considering model results over 1950 to 2100.

Thanks for these great suggestions.

We have added a sentence starting on line 71 that reads:

We arrive at remarkably similar conclusions based upon consideration of the temperature at which chlorine is activated on aerosols^{18,26}, rather than T_{NAT} , in the calculations that follow because these two temperature thresholds are so closely related (see Methods).

with citations to:

¹⁸Drdla, K. & Müller, R. Temperature thresholds for chlorine activation and ozone loss in the polar stratosphere. *Ann. Geophys.* **30**, 1055–1073 (2012).

²⁶Tilmes, S. *et al.* Evaluation of heterogeneous processes in the polar lower stratosphere in the Whole Atmosphere Community Climate Model. *J. Geophys. Res. Atmos.* **112**, (2007).

This statement of “remarkably similar conclusions” is supported by Supplementary Figures 8, 9, 10, 11 and 18 all based on T_{ACL} rather than T_{NAT} . These figures, and the new text in the Methods section describing the analysis of Aerosol Reactivity Potential (ARP), support the “remarkably similar conclusions” statement added to the main article.

There is some discussion in the paper regarding violation of the Montreal protocol. However, these violations happen against the background of the changing (and regulated by the Montreal protocol) stratospheric halogen loading, which should not be neglected in the paper. Note that Chipperfield et al. (2015) concluded that a very strong Arctic ozone loss, with column ozone values below 120 DU, would have occurred without the Montreal protocol for meteorological conditions as in 2011. Moreover, not only chlorine levels might develop in a different way than predicted, the strength of the chemical ozone loss might also be affected by bromine levels and the contribution of short-lived bromine substances (e.g., Fernandez et al., 2017).

Upon revision we have decided to remove any mention of the violation of the Montreal Protocol, in part because this discussion is not truly central to the thesis of the paper, and also because one of us (RS) recently reviewed a manuscript, to be published in Nature sometime soon, that indicates the violations of the Montreal Protocol within eastern Asia ceased around the start of 2019 (please note this is confidential information).

2. PFP and ozone loss

Figure 1: As Fig. 1 in the submitted manuscript, but for the subset of the five coldest winters (2020, 2011, 2005, 1996, 2000, if I am reading the colour scale correctly).

Figure 1 in the submitted manuscript describes the relation between column ozone loss in the vortex and PFP. First, I think the paper could be clearer about the fact that the ozone loss reported here has been determined with the Match method. The paper should also cite the papers, where all ozone loss values except 2020 have been published (assuming this is true). If there is anything particular to say about the ozone loss determination for 2020 it could be discussed in the appendix (or perhaps a preprint could be cited). Another question that arises is how well the ozone loss determined from Match compares with other ozone loss estimates. Second, if one only considers the five coldest winters (Fig. 1) the correlation between ozone loss and PFP is less obvious.

In reply to the first point, the text starting on line 107 has been modified to read:

Figure 1a shows values of column ozone loss based on our analyses of ozonesonde measurements in the Arctic vortex, plotted as function of PFP. Data values are shown for all of the cold winters that have occurred since the inception of regular ozonesonde launches, in 1991. The estimates of ΔO_3 are based either on Match events (situations where individual air masses are usually probed twice above different measurement stations)^{2,3,6,34} or on the difference between a passive ozone tracer and the vortex mean, observed profile of ozone¹⁶.

with citations to:

²Rex, M. *et al.* Arctic ozone loss and climate change. *Geophys. Res. Lett.* **31**, (2004).

³Rex, M. *et al.* Arctic winter 2005: Implications for stratospheric ozone loss and climate change. *Geophys. Res. Lett.* **33**, (2006).

⁶Harris, N. R. P., Lehmann, R., Rex, M. & von der Gathen, P. A closer look at Arctic ozone loss and polar stratospheric clouds. *Atmos. Chem. Phys.* **10**, 8499–8510 (2010).

¹⁶Wohltmann, I. *et al.* Near-complete local reduction of Arctic stratospheric ozone by severe chemical loss in spring 2020. *Geophys. Res. Lett.* **47**, e2020GL089547 (2020).

³⁴Rex, M. *et al.* Chemical ozone loss in the Arctic winter 1994/95 as determined by the Match technique. *J. Atmos. Chem.* **32**, 35–59 (1999).

This change provides considerably more detail and documentation for the ozonesonde based loss estimates than had been present in the original submission.

--

In terms of the question “how well the ozone loss determined from Match compares with other ozone loss estimates?” we present below a screen capture of Table 3-2 of the 2002 WMO/UNEP Ozone Assessment Report:

Table 3-2. Comparisons of chemical loss of column ozone, column [$O_3^* - O_3$] (see Rex et al., 2002a), inside the Arctic vortex for the winter of 1999/2000 as of the indicated date. N/A indicates that data for that date are not available.

Data Source	Method	Reference	Chemical Loss of Column Ozone (DU)		
			5 March 2000	15 March 2000	28 March 2000
OMS balloon	Tracer-tracer (O_3 vs. N_2O)	Salawitch et al. (2002a)	61 ± 14	N/A	N/A
HALOE	Tracer-tracer	Müller et al. (2002)	N/A	84 ± 13	N/A
POAM III satellite	Vortex-averaged descent	Hoppel et al. (2002)	51 ± 11	67 ± 11	N/A
SAOZ network	Transport model	EC (2001)	85 ± 24	98 ± 25	101 ± 30
Ozonesondes	Match	Rex et al. (2002a)	53 ± 11	71 ± 12	88 ± 13

Page 3.50 of Chapter 3, available at:

<https://csl.noaa.gov/assessments/ozone/2002/chapters/chapter3.pdf>

states:

Table 3-2 gives an overview of various estimates for the deficit in the total column amount of ozone due to chemical loss of ozone. The numbers given are the difference between the actually observed column amount of ozone and the column amount of ozone that would have been present at a given day in the absence of chemical ozone loss, dynamics being equal. **For comparable time periods the agreement between results from all approaches is within the error bars.** The results from the SAOZ/-transport model study are generally somewhat higher than the other approaches and have larger uncertainties. The results of the other approaches agree to better than 20%. Figure 3-34c shows the evolution of the total column loss from January through March, as determined by Match. By the end of March, the chemically induced ozone deficit amounted to between 90 and 100 DU. This is roughly the amount of total ozone that has been supplied to the polar vortex by dynamic effects during the same time, and therefore the total ozone column remained relatively constant during January to March (Rex et al., 2002a), which is in contrast to the natural, climatological increase of the Arctic ozone column during this season.

We prefer to make no change to the paper in response to this comment, since making a change would require adding a citation to Chapter 3 of this fairly old report. We are presently at the limit of 70 for the total number of citations.

--

Thanks for the excellent second point, accompanied by the figure. First, restricting a correlation to a smaller part will usually reduce the computed and visual correlation. However, this comment, as well as Major Comment 2 from Reviewer 2, prompted us to conduct a new analysis of trends in the quantity $EESC^{1.2} \times PFP$. As noted in the opening paragraph of our reply, and as now shown in our new Fig 1b, the quantity $EESC^{1.2} \times PFP$ serves as a good proxy for the chemical loss of column ozone. We also quantify the impact of trends in stratospheric H_2O induced by rising CH_4 and the thermodynamics of the tropical tropopause, in terms of both PFP and $EESC^{1.2} \times PFP$. Most importantly, we now use the global ATLAS Chemistry and Transport Model that includes a comprehensive treatment of stratospheric chemistry (Wohltmann and Rex, GMD, doi:10.5194/gmd-2-153-2009, 2009), along with our analysis of ozonesonde data to show that the chemical loss of column ozone in the Arctic (ΔO_3) is well described by the quantity $EESC^{1.2} \times PFP$ (new Fig 1b). New Figures 2 and 7 and the associated text describe various aspects of the trends in $EESC^{1.2} \times PFP$, and new Figures 8 and 9 show how $EESC^{1.2} \times PFP$ computed from the ensemble average of CMIP6 GCM output, for four SSP scenarios.

I think we all agree that the ozone loss in 2020 was particularly strong as it was cold enough for heterogeneous chemistry for an extended period in March and April. A hypothetical winter/spring with a very cold vortex from December to February but with a warming in (say) early March, would certainly have much less ozone depletion than observed in 2020 even if the PFP value were the same as in 2020

(about 29 d). On the other hand, there could be a small volume of the polar vortex below the threshold for heterogeneous chemistry, which nonetheless is responsible for activating a large fraction of the Arctic vortex (Wegner et al., 2016). Because the typical life cycle of the polar vortex incorporates implicitly a link between a cold and a long-lived vortex, concepts like PFP are useful indicators of polar ozone loss. However, I suggest an a bit more careful discussion of the subject in the manuscript.

Have changed the text, starting on line 123, to read:

The near linear relation between ΔO_3 and V_{PSC} is a robust relation for the contemporary Arctic stratosphere^{5,6}, despite the fact that in early winter, a small volume of the Arctic vortex can exist below the temperature threshold for chlorine activation and affect a large portion of the vortex³⁸.

with citations to:

⁵ Chipperfield, M. P., Feng, W. & Rex, M. Arctic ozone loss and climate sensitivity: Updated three-dimensional model study. *Geophys. Res. Lett.* **32**, (2005)

⁶ Harris, N. R. P., Lehmann, R., Rex, M. & von der Gathen, P. A closer look at Arctic ozone loss and polar stratospheric clouds. *Atmos. Chem. Phys.* **10**, 8499–8510 (2010).

³⁸ Wegner, T. et al. Vortex-wide chlorine activation by a mesoscale PSC event in the Arctic winter of 2009/10. *Atmos. Chem. Phys.* **16**, 4569–4577 (2016).

3. Warm winters

Somewhat crudely speaking the message of this manuscript is that “cold Arctic winters are getting colder”. However most winters in the analysed time series are not ‘cold’ but are rather ‘warm’. If the message of the paper is that the “colder cold winters” are driven by the increase in anthropogenic GHGs, why does this not affect the ‘warm’ winters? This question is left open in the manuscript, isn’t it?

I suggest applying the ISA technique put forward here (or some variant of it) also to warm winters, i.e. follow a similar strategy as outlined in lines 287-298, but remove the ‘cold’ years from the time series. The analysis has to be done first, but I would expect (see Fig. 2 of the manuscript) that the conclusion is that the frequency of warm winters has not changed significantly over the time period of 1989 to 2020. On the other hand, from inspecting Fig. 2e, it seems that what used to be a cold winter in the late 1960s is considered a warm winter after the year 2000 by the ISA method. I think adding such a discussion would allow a weak point of this paper (what happens to the warm winters?) to be addressed.

Have changed text, starting on line 201, to read:

Our analysis is focused primarily on the coldest Arctic winters because this condition is clearly associated with large ozone loss. The GCM simulations for SSP5-8.5 shown in Figs. 4 and 5 also show a tendency for PFP associated with the warmer Arctic winters (bottom of the data envelope) to rise over time, a projected trend that is not yet apparent in observations⁴⁷ due perhaps to the generally lower values in the observational period and the lower limit of zero.

with a citation to:

⁴⁷ Rieder, H. E., Polvani, L. M. & Solomon, S. Distinguishing the impacts of ozone-depleting substances and well-mixed greenhouse gases on Arctic stratospheric ozone and temperature trends. *Geophys. Res. Lett.* **41**, 2652–2660 (2014).

We would rather not delve any further into the warm winters other than acknowledge they are also getting colder (as reflected in larger PFP), as we are pressed for space and we think the other additions to the paper are more important. If, upon reading the revision you feel more words about the warm winters are warranted, please let us know and we’ll be happy to accommodate.

4. Changes in stratospheric wave intensity

I suggest that the paper gives a bit more background and discussion on the cases of the variability (which is evident e.g. in Figures 2, 3, and 4 in the manuscript) in vortex temperatures, vortex strength and vortex position. For example, studies have reported that the tropospheric climate shifted around the year 2000 and that the December stratospheric wave intensity during 2001-2015 has weakened, in contrast to that during 1979-2000, implying a shift of stratospheric wave intensity around 2000 (Hu et al., 2019, and references therein). Hu et al. (2019) further find that the shift in stratospheric wave intensity leads to a shift in stratospheric Arctic temperatures, that is, to a warming during 1979-2000 and to a cooling during 2001-2015. This could imply that similar shifting phenomena may appear regardless of the continued greenhouse gas emissions. Could such findings have an impact on the conclusions of the present manuscript?

Further, the variability of Arctic ozone is also related to sea surface temperatures; e.g., Liu et al. (2020) find that the distribution of ozone is related to the positive phase of sea surface temperatures over the North Pacific, which could also have an impact on the stratospheric Arctic vortex. And there could be a feedback between stratospheric ozone depletion and the position of the polar vortex. Studies (e.g., Zhang et al., 2020, and references therein) have found a shift of the Arctic stratospheric polar vortex toward Siberia during late winter since 1980 and there are reports that zonally asymmetric stratospheric ozone depletion gives a significant feedback on the position of the polar vortex (with changes happening over the time period of interest here).

Much thanks for these great suggestions.

Have added the following text starting on line 215:

On the other hand, tropospheric climate exhibited a shift in the early 2000s that weakened the intensity of planetary wave activity propagating into the stratosphere⁴⁸, which could be responsible for a portion of the larger observed value of $S_{\text{PFP-LM}}$ compared to results from GCMs. Shifts in patterns of sea surface temperature in the North Pacific have also been implicated as a causal factor in decreased planetary wave activity and the strengthening of the Arctic vortex⁴⁹. The potential association of these drivers of Arctic, stratospheric temperature with climate change is an area of active research⁴⁸.

With citations to:

⁴⁸ Hu, D., Guo, Y. & Guan, Z. Recent Weakening in the Stratospheric Planetary Wave Intensity in Early Winter. *Geophys. Res. Lett.* **46**, 3953–3962 (2019).

⁴⁹ Liu, M., Hu, D. & Zhang, F. Connections between stratospheric ozone concentrations over the Arctic and sea surface temperatures in the North Pacific. *J. Geophys. Res. Atmos.* **125**, e2019JD031690 (2020).

We carefully examined the three suggested papers and came to the conclusion that the Hu et al. and Liu et al. papers were the most important of the three. Since we are only limited to 70 citations total, we have to make some difficult decisions.

5. Summary Figure

Figure 2: Similar as Fig. 2 in the submitted manuscript, but focusing on the ‘cold’ winters only. Blue symbols show ERA5 data. Red symbols show JRA-55 data. Shown are the ‘cold winter’ data and the corresponding fits. (Note that the data points have been crudely extracted from the plot provided in the manuscript; this is only meant as a sketch.)

One point that I think is missing in the paper is a good summary (iconic) figure for the message of the paper. In Figure 2 of this review I have tried to provide a sketch how such a figure could look. I suggest focusing on the ‘cold’ winters only, which is the focus of the manuscript. One could try including also the information (fits?) from Merra2 and CFSR. I would definitely suggest including information (grey scale?) from the models (Fig. 3 of the main paper). My suggestion would be to focus on ERA5 and JRA-55, but this is obviously a choice of the authors. However, I note that in Fig. 2 the JRA-55 information is shown twice (this is not referring to the fit), so there could be space for showing a summary figure.

Much thanks for these great suggestions. Our team had extensive discussions regarding the need for a great summary figure, based on this comment, and we arrived at the new Figures 8 and 9, as well as the new Figure 1b.

We hope these three new graphical elements address this excellent comment.

Details

- l. 10, 11: This sounds as if the formation of PSCs is a requirement for the onset of polar ozone loss. However, polar ozone loss is really driven by the onset of heterogeneous chlorine activation, which is not the same things as the formation of PSCs.

We have re-written the abstract due to modifications to the paper based on this and the other two reviews, and this sentence has been removed.

- l. 34-42: you discuss here the results from three particular GCMs - how relevant is this discussion given the fact that later in the paper you discuss the results of a number of CMIP model results and furthermore point to the importance of a well resolved stratosphere (e.g. quality of the QBO representation).

The original text had read:

An early evaluation using a general circulation model (GCM) with coupled active chemistry (a chemistry climate model, or CCM) suggested decreases in planetary wave activity reaching the mid-latitude stratosphere due to increased westerly winds in the subtropics, driven by rising levels of GHGs, would lead to stronger, colder Arctic vortices¹¹. More recently, a simulation using another CCM suggested future cooling of the Arctic lower stratosphere during early winter would result from direct radiative cooling driven by GHGs and indirect effects related to declining Arctic sea ice and rising sea surface temperatures¹². Simulations conducted using a third CCM showed modest cooling (~ 0.15 K decade⁻¹) of the future Arctic

stratosphere at 50 hPa also driven by GHGs, with high interannual variability that complicates the assessment of statistical significance¹³.

Here, our intent was to cite three important prior papers:

¹¹Shindell, D. T., Rind, D. & Lonergan, P. Increased polar stratospheric ozone losses and delayed eventual recovery owing to increasing greenhouse-gas concentrations. *Nature* 392, 589–592, 445 (1998).

¹²Langematz, U. et al. Future Arctic temperature and ozone: The role of stratospheric composition changes. *J. Geophys. Res. Atmos.* 119, 2092–2112 (2014).

¹³Bednarz, E. M. et al. Future Arctic ozone recovery: the importance of chemistry and dynamics, *Atmos. Chem. Phys.* 16, 12159–12176 (2016).

that address a topic similar to that of our submitted manuscript. We used CCM three times to highlight that prior knowledge in this topical area has been based on results from this modelling tool, and to foreshadow the reliance on CCM output in our paper.

We have greatly expanded the discussion of the quality of QBO representation, as noted in our reply to the comment regarding LL170-175 and LL410-418 of Reviewer 2.

For this revision we have left this block of text unchanged, since we highlight QBO later in the revised paper. We'll be happy to consider a modification if so directed.

- I 54: The formation mechanisms of ternary solution droplets (STS) and NAT are rather different, NAT formation requires formation of crystals and thus supersaturation, while STS are liquid particles; the formation mechanism of large NAT particles (ref. 20) is likely a different issue again (e.g., Molleker et al., 2014; Hoffmann et al., 2017; Pitts et al., 2018). I suggest being a bit more specific here.

We respond to this comment, and the following, just below.

- I. 58: I suggest being more specific here, although I agree in principle. But heterogeneous chlorine activation requires neither NAT nor STS to be present; it can start on cold sulphate aerosol (e.g., Wegner et al., 2012; Drdla and Müller, 2012, refs. 21 and 23 of the paper). Further HNO₃ is likely not constant in a cold Arctic vortex (e.g., Waibel et al., 1999). Finally, T_{NAT} depends on water vapour and HNO₃, whereas HNO₃ is not relevant for estimating the threshold for activation on cold sulphate aerosol, so the two thresholds cannot always agree.

Much thanks!

The text on lines 53 to 60, the entire paragraph of this and the prior comment, had read:

When temperatures fall during Arctic winter, supercooled ternary (H₂SO₄-HNO₃-H₂O) solution droplets¹⁸ and nitric acid trihydrate (NAT) particles^{19,20} are the first types of PSCs to form. The timescale for chemical processing of chlorine reservoir gases on STS droplets transitions from weeks to days near the temperature at which NAT becomes thermodynamically stable (T_{NAT})²¹. T_{NAT} is governed by the vapour pressure of nitric acid (HNO₃) and water (H₂O)¹⁹. We use constant volume mixing ratio profiles of H₂O and HNO₃ to find T_{NAT} (see Methods) since this is straightforward to describe and implement, and also because the actual temperature at which chlorine activation occurs bears a close quantitative relation to T_{NAT} found in this manner^{22–24}.

with citations to:

¹⁹Hanson, D. & Mauersberger, K. Laboratory studies of the nitric acid trihydrate: Implications for the south polar stratosphere. *Geophys. Res. Lett.* **15**, 855–858 (1988).

²⁰Fahey, D. W. et al. The detection of large HNO₃-containing particles in the winter Arctic stratosphere. *Science* **291**, 1026–1031 (2001).

- ²¹Wegner, T. *et al.* Heterogeneous chlorine activation on stratospheric aerosols and clouds in the Arctic polar vortex. *Atmos. Chem. Phys.* **12**, 11095–11106 (2012).
- ²²Kawa, S. R. *et al.* Activation of chlorine in sulfate aerosol as inferred from aircraft Observations. *J. Geophys. Res. Atmos.* **102**, 3921–3933 (1997).
- ²³Drdla, K. & Müller, R. Temperature thresholds for chlorine activation and ozone loss in the polar stratosphere. *Ann. Geophys.* **30**, 1055–1073 (2012).
- ²⁴Wohltmann, I., Lehmann, R. & Rex, M. A quantitative analysis of the reactions involved in stratospheric ozone depletion in the polar vortex core. *Atmos. Chem. Phys.* **17**, 10535–10563 (2017).

This paragraph in which this text had appeared has been modified, starting on line to read:

Chemical loss of O₃ in the Arctic stratosphere occurs following the activation of chlorine on the surfaces of either cold sulphate aerosols^{18,19}, supercooled ternary (H₂SO₄-HNO₃-H₂O) solution droplets²⁰ (STS), or nitric acid trihydrate (NAT) particles²¹. When temperatures fall during Arctic winter, STS and NAT particles^{22,23} are the first types of PSCs to form. The timescale for chemical processing of chlorine reservoir gases on STS droplets transitions from weeks to days near the temperature at which NAT becomes thermodynamically stable (T_{NAT})¹⁹. T_{NAT} is governed by the vapour pressure of nitric acid (HNO₃) and water (H₂O)²¹. Here, we use a constant volume mixing ratio of stratospheric H₂O equal to 4.6 parts per million (ppmv) and a profile of HNO₃, both based on satellite observations, to find T_{NAT} (see Methods). The quantity V_{PSC} represents the volume of air for which temperature is less than or equal to T_{NAT}, evaluated between potential temperatures of 400 and 700 K. The formation of PSCs in the Arctic stratosphere also depends on factors such as cooling rate, the degree of supersaturation, the chemical composition of pre-existing nuclei, as well as the surface coating of condensed particles²⁴. The profile of HNO₃ can be altered by the physical sedimentation of nitrate-bearing PSCs, termed denitrification²⁵. Nonetheless, our approach captures the essence of the century and a half long variations in meteorological conditions on the primary factor that drives chemical loss of Arctic O₃: that is, temperatures cold enough to allow for the formation of PSCs. We arrive at remarkably similar conclusions based upon consideration of the temperature at which chlorine is activated on aerosols^{18,26}, rather than T_{NAT}, in the calculations that follow because these two temperature thresholds are so closely related (see Methods).

with citations to:

- ¹⁸Drdla, K. & Müller, R. Temperature thresholds for chlorine activation and ozone loss in the polar stratosphere. *Ann. Geophys.* **30**, 1055–1073 (2012).
- ¹⁹Wegner, T. *et al.* Heterogeneous chlorine activation on stratospheric aerosols and clouds in the Arctic polar vortex. *Atmos. Chem. Phys.* **12**, 11095–11106 (2012).
- ²⁰Carlaw, K. S. *et al.* Stratospheric aerosol growth and HNO₃ phase depletion from coupled HNO₃ and water uptake by liquid particles. *Geophys. Res. Lett.* **21**, 2479–2482 (1994).
- ²¹Hanson, D. & Mauersberger, K. Laboratory studies of the nitric acid trihydrate: Implications for the south polar stratosphere. *Geophys. Res. Lett.* **15**, 855–858 (1988).
- ²²Peter, T., Müller, R., Crutzen, P. J. & Deshler, T. The lifetime of leewave-induced ice particles in the Arctic stratosphere: II. Stabilization due to NAT-coating. *Geophys. Res. Lett.* **21**, 1331–1334 (1994).
- ²³Northway, M. J. *et al.* An analysis of large HNO₃-containing particles sampled in the Arctic stratosphere during the winter of 1999/2000. *J. Geophys. Res. Atmos.* **107**, SOL 41-1-SOL 41-22 (2002).
- ²⁴Molleker, S. *et al.* Microphysical properties of synoptic-scale polar stratospheric clouds: in situ measurements of unexpectedly large HNO₃-containing particles in the Arctic vortex. *Atmos. Chem. Phys.* **14**, 10785–10801 (2014).
- ²⁵Waibel, A. E. *et al.* Arctic ozone loss due to denitrification. *Science* **283**, 2064 – 2069 (1999).

As noted above in our reply to Major Comment #1, the statement of “remarkably similar conclusions” is supported by Supplementary Figures 8, 9, 10, 11 and 18, all of which are based on T_{ACL} rather than T_{NAT} and exhibit very similar behaviour as their counterparts in the Main paper.

- I. 72: note that the ECMWF says: “The entire ERA5 dataset from 1950 to present is expected to be available for use in 2020”.

The ERA5 dataset for years prior to 1979, at the time of this revision, is in “beta” mode. Meteorological fields from 1980 to 2020 from ERA5 are available at 137 vertical model levels and at pressure levels. We used in our analysis the model level data. Two months ago, ECMWF published *preliminary* data prior to 1979, which are only available at 37 vertical pressure levels (6 between 100 hPa and 10 hPa). Consequently, we continue to only show results from ERA5 data from 1979 onwards.

- I. 78: boundary of the vortex (see also Methods): I would state the value of 36 sec^{-1} for nPV in the main paper. But PV is strongly altitude dependent (which is why nPV is used), so how well does nPV ‘correct’ for this altitude dependence? How does nPV compare to other forms of scaled PV (e.g., Lait, 1994)? Perhaps more importantly; the paper states that temperatures are changing in the stratosphere because of global change; at the same time the assumption is that PV is not affected so that using the constant vortex boundary value of 36 sec^{-1} for nPV is correct. How well is this assumption justified? (see also Liu et al., 2020)

The figure below compares the scaling factors from Rex et al. (1999) to that of Lait (1994):

These two scaling factors agree quite well; the same conclusion would be reached if either was used.

In terms of the validity of the 36 sec^{-1} boundary as climate changes, a detailed quantitative analysis is beyond the scope of this study. We have however examined fields of nPV, contours for T_{NAT} , and T_{ICE} , for 1 Feb of every tenth year, from 1960 to 2100, for the 26 CMIP6 GCMs that archived results for SSP5-8.5. We see no issues with the use of the 36 PVU boundary out to end of century. The following text, accompanied by the new Supplementary Figure 2, has been added to Methods (line 362) to address this important point:

We have examined maps of nPV and temperature plotted for 1 February of years, 1960 to 2100, every 10 years, for all 26 CMIP6 GCMs that archived results for SSP5-8.5. These maps show that the nPV = 36 sec⁻¹ boundary for the Arctic vortex is not greatly affected by climate change until the end of century; maps for the four CMIP6 GCMs highlighted in Fig. 5 of the paper are shown in Supplementary Fig. 2.

- I. 88: why do you start in 1993?

We start in 1993 because this was the first cold winter after the inception of regular ozonesonde launches. We never calculated an ozone column loss for the first Match winter 1991/92, which in fact was a warm winter. The text starting on line 107 has been modified to read:

Figure 1a shows values of column ozone loss based on our analyses of ozonesonde measurements in the Arctic vortex, plotted as function of PFP. Data values are shown for all of the cold winters that have occurred since the inception of regular ozonesonde launches, in 1991. The estimates of ΔO_3 are based either on Match events (situations where individual air masses are usually probed twice above different measurement stations)^{2,3,6,34} or on the difference between a passive ozone tracer and the vortex mean, observed profile of ozone¹⁶.

with citations to:

²Rex, M. *et al.* Arctic ozone loss and climate change. *Geophys. Res. Lett.* **31**, (2004).

³Rex, M. *et al.* Arctic winter 2005: Implications for stratospheric ozone loss and climate change. *Geophys. Res. Lett.* **33**, (2006).

⁶Harris, N. R. P., Lehmann, R., Rex, M. & von der Gathen, P. A closer look at Arctic ozone loss and polar stratospheric clouds. *Atmos. Chem. Phys.* **10**, 8499–8510 (2010).

¹⁶Wohltmann, I. *et al.* Near-complete local reduction of Arctic stratospheric ozone by severe chemical loss in spring 2020. *Geophys. Res. Lett.* **47**, e2020GL089547 (2020).

³⁴Rex, M. *et al.* Chemical ozone loss in the Arctic winter 1994/95 as determined by the Match technique. *J. Atmos. Chem.* **32**, 35–59 (1999).

- I. 111: the meaning of ‘p-values’ is not clear here

We have changed the text starting on line 158 to read:

We have conducted Monte-Carlo simulations to assess the statistical significance of S_{PFP-LM} and the 1σ uncertainty in S_{PFP-LM} (ΔS_{PFP-LM}) found using the ISA selection procedure (see Methods): p-values resulting from the MC-simulations are all less than 0.05 for $S_{PFP-LM}/\Delta S_{PFP-LM}$ from all four meteorological data centres (Table 1), indicating statistical significance at better than the 2σ confidence level for this important metric of the trend in PFP^{LM} .

to hopefully clarify that the p-values are found from the Monte-Carlo simulations, as described in Methods.

- I. 119: quantify ‘substantial’

The original text had read:

To compensate for temperature biases of the GCMs, which can be substantial, the temperature threshold for formation of PSCs has been offset by a constant amount unique to each model so that the overall magnitude of PFP^{LM} in the GCM matches the observed magnitude of PFP^{LM} over the modern satellite era (see Methods).

We have modified this sentence, starting on line 168, to read:

To compensate for temperature biases of the GCMs, which can approach 5 K and together with the low bias of these models relative to observed stratospheric H₂O (ref ⁴³) leads to an underestimate of the accumulated exposure to PSCs in the Arctic⁴⁴, the temperature threshold for formation of PSCs has been offset by a constant amount specific to each model such that the overall magnitude of PFP^{LM} in the GCM matches the observed magnitude of PFP^{LM} over the modern satellite era.

with citations to:

⁴³Keeble, J. *et al.* Evaluating stratospheric ozone and water vapor changes in CMIP6 models from 1850-2100. *Atmos. Chem. Phys. Discuss.* **2020**, 1–68 (2020).

⁴⁴Butchart, N. *et al.* Multimodel climate and variability of the stratosphere. *J. Geophys. Res. Atmos.* **116**, (2011).

- I. 126: I think I know what you mean, but this sentence is not really clear

The original text, starting on line 124 (start of sentence), had read:

The majority of the ISA-based values of $S_{\text{PFP-LM}}$ from the 24 GCM simulations with archived results for SSP5-8.5 lie between about 1.0 and 2.0 d decade⁻¹ (Fig. 4); 17 of the runs exhibit values of $S_{\text{PFP-LM}}$ and 22 of the runs exhibit values of $S_{\text{PFP-LM}} / \Delta S_{\text{PFP-LM}}$ with statistical significance at better than the 2 σ level.

We have modified this sentence, starting on line 182, to read:

The majority of the ISA-based values of $S_{\text{PFP-LM}}$ from the 26 GCM simulations with archived results for SSP5-8.5 lie between about 1.0 and 2.0 d decade⁻¹ (Fig. 5); statistical significance at better than the 2 σ level is exhibited for $S_{\text{PFP-LM}}$ in 16 of these runs and for $S_{\text{PFP-LM}} / \Delta S_{\text{PFP-LM}}$ in 24 of these runs.

- I. 146: do you mean ‘stratosphere’ here or ‘polar stratosphere’ ?

Great catch: we’ve changed to “polar stratosphere”

- I. 158: (and the discussion in the entire paragraph). This discussion sounds a bit speculative, I suggest to shorten. The discussion focuses on ‘internal variability’, however, given the large range of the temperature (and thus PFP) mismatch in the CCM results, I could imagine other inadequacies of the models leading to problems simulating $S_{\text{PFP-LM}}$ than just ‘internal variability’.

The original text (with internal variability emphasized for convenience of the finding this phrase) had read:

The mean and standard deviation of the empirical value of $S_{\text{PFP-LM}}$ over 1980 (or 1981) to 2020 from the four reanalysis datasets is compared to GCM-based values (for the same time period) in Fig. 5a. The rationale for this comparison is the models have undergone a similar rise in the RF of climate over these four decades as the atmosphere. As discussed above for the longer time period, there is also more spread in the CMIP5 values of $S_{\text{PFP-LM}}$ than the CMIP6 values, when considering only the past four decades. The observationally-based trend lies near the upper 1 σ value of the GCMs. Over this shorter period internally generated climate variability may play a substantial role and during these four decades the one realisation of climate that developed in earth's climate system may have coincidentally followed a path that led to values of $S_{\text{PFP-LM}}$ at the upper range of the frequency function from the GCMs. We interpret the results in Fig. 5a as follows: there is strong similarity in the four observationally based estimates of $S_{\text{PFP-LM}}$, and this value is consistent with a subset of the

GCMs. It is difficult to attach any further meaning to this comparison; because of the potential role of *internal variability* we would caution against asserting that GCMs with the best match to the empirically-based value $S_{\text{PFP-LM}}$ will provide a more realistic forecast of the future.

Based on Major Comment 4 from this review, we have modified this paragraph (starting on line 229; again with the phrase internal variability emphasized) to read:

The mean and standard deviation of the empirical value of $S_{\text{PFP-LM}}$ over 1980 (or 1981) to 2020 from the four reanalysis datasets is compared to GCM-based values (for the same time period) in Fig. 6a. The rationale for this comparison is the models have undergone a similar rise in the RF of climate over these four decades as the atmosphere. As discussed above for the longer time period, there is also more spread in the CMIP5 values of $S_{\text{PFP-LM}}$ than the CMIP6 values, when considering only the past four decades. The observationally-based trend lies near the upper 1σ value of the GCMs. Over this shorter period internally generated climate variability may play a substantial role and during these four decades the one realisation of climate that developed in earth's climate system may have coincidentally followed a path that led to values of $S_{\text{PFP-LM}}$ at the upper range of the frequency function from the GCMs.

On the other hand, tropospheric climate exhibited a shift in the early 2000s that weakened the intensity of planetary wave activity propagating into the stratosphere⁴⁸, which could be responsible for a portion of the larger observed value of $S_{\text{PFP-LM}}$ compared to results from GCMs. Shifts in patterns of sea surface temperature in the North Pacific have also been implicated as a causal factor in decreased planetary wave activity and the strengthening of the Arctic vortex⁴⁹. The potential association of these drivers of Arctic, stratospheric temperature with climate change is an area of active research⁴⁸. We interpret the results in Fig. 6a as follows: there is strong similarity in the four observationally based estimates of $S_{\text{PFP-LM}}$, and this value is consistent with a subset of the GCMs. It is difficult to attach further meaning to this comparison; because of the potential role of *internal variability* in planetary wave activity, we caution against asserting that GCMs with the best match to the empirically-based value $S_{\text{PFP-LM}}$ will provide a more realistic forecast of the future

with references to:

⁴⁸Hu, D., Guo, Y. & Guan, Z. Recent Weakening in the Stratospheric Planetary Wave Intensity in Early Winter. *Geophys. Res. Lett.* **46**, 3953–3962 (2019).

⁴⁹Liu, M., Hu, D. & Zhang, F. Connections between stratospheric ozone concentrations over the Arctic and sea surface temperatures in the North Pacific. *J. Geophys. Res. Atmos.* **125**, e2019JD031690 (2020).

We have made these changes to provide greater context to the phrase “internal variability”.

- I. 183: what is shown in this paper cannot have affected the ‘reluctance’ in the past mentioned here.

Great point; this sentence has been deleted.

- I. 186: the authors of this paper clearly do not agree with the results of Rieder and Polvani (2013) and Rieder et al. (2014) - I think that for the present paper a consensus on this issue (e.g., VAS against ISA) should not be required; I would expect further discussion on this issue in the scientific literature

This sentence has likewise been deleted, due to the new direction of the paper and the need to keep to 5000 words.

- Ref. 16: see also the special issue in JGR/GRL on the ozone loss in the Arctic in 2020 (preprints available at: <https://www.essoar.org/>) and the discussion paper by Dameris et al. (2020).

Given the new focus of the paper towards EESC^{1,2}×PFP and away from GHG forcing of the ozone loss that occurred in the Arctic during 2020, the fact that Ingo Wohltmann has now joined our paper due to provision of ATLAS modeling results, plus the central role that the value of ΔO_3 for 2020 from Wohltmann et al. (2020) plays in our analysis (ref. 16 of the original paper), we have decided to retain this citation and not add any more citations to Arctic O₃ in 2020.

- I. 200: See Rigby et al. (2019) for the origin of illegal recent production of CFC-11; also note that stratospheric bromine levels might change (e.g., Fernandez et al., 2017)

Much thanks! We have changed the focus of the paper and, for space constraints, have removed any mention of illegal production of CFC-11. See our reply to your Major Comment 1 for more details.

- I. 203: can you quantify 'significant' ?

This sentence has been removed.

- I. 211: I agree, but assuming that society takes action in reducing the emissions of GHGs, what would be the timescale over which stratospheric temperatures might change (e.g. compared to the expected decline in the stratospheric halogen loading).

This sentence has also been removed, again due to the new focus of the paper.

- I. 222: you say 'we might have': what would be the expected consequences if you did so.

The original text, in Methods, had read:

It is possible to adopt a more sophisticated approach for the specification of H₂O and HNO₃, for instance, we could have considered declines in HNO₃ due to the sedimentation of PSC particles²⁰ or we might have evaluated meteorological conditions based upon the temperature for chlorine activation, which depends on temperature, H₂O, and sulphate surface area^{23,49}.

with citations to:

²⁰Fahey, D. W. *et al.* The detection of large HNO₃-containing particles in the winter Arctic stratosphere. *Science* **291**, 1026–1031 (2001).

²³Drdla, K. & Müller, R. Temperature thresholds for chlorine activation and ozone loss in the polar stratosphere. *Ann. Geophys.* **30**, 1055–1073 (2012).

⁴⁹Tilmes, S. *et al.* Evaluation of heterogeneous processes in the polar lower stratosphere in the Whole Atmosphere Community Climate Model. *J. Geophys. Res. Atmos.* **112**, (2007).

We have deleted this sentence from methods, because starting on line 67 of the main article, the text has been modified to read:

The profile of HNO₃ can be altered by the physical sedimentation of nitrate-bearing PSCs, termed denitrification²⁵. Nonetheless, our approach captures the essence of the century and a half long variations in meteorological conditions on the primary factor that drives chemical loss of Arctic O₃: that is, temperatures cold enough to allow for the formation of PSCs. We arrive at remarkably similar conclusions based upon consideration of the temperature at which chlorine is activated on aerosols^{18,26}, rather than T_{NAT}, in the calculations that follow because these two temperature thresholds are so closely related (see Methods).

with citations to:

¹⁸Drdla, K. & Müller, R. Temperature thresholds for chlorine activation and ozone loss in the polar stratosphere. *Ann. Geophys.* **30**, 1055–1073 (2012).

²⁵Waibel, A. E. *et al.* Arctic ozone loss due to denitrification. *Science* **283**, 2064 LP – 2069 (1999).

²⁶Tilmes, S. *et al.* Evaluation of heterogeneous processes in the polar lower stratosphere in the Whole Atmosphere Community Climate Model. *J. Geophys. Res. Atmos.* **112**, (2007).

We have added Supplementary Figures 10 and 18 which shows times series of $EESC^{1-2} \times ARP$, for which Aerosol Reactivity Potential (ARP) is an analogue of PFP, but based on the volume of air for which temperature lies below the chlorine activation temperature (T_{ACL}) rather than T_{NAT} . The close agreement of these figures to Figs. 5 and 9 in the main section of revised paper supports the insensitivity of our primary, high-level result to details of how exactly chlorine activation is defined.

- I. 226: as discussed above how well is the constant vortex boundary value of 36 sec^{-1} for nPV justified? Over which altitude regime is it valid?

Please see reply to line 78, for the scaling factor comparison.

We have examined PV maps for 1 February for all of the GCMs shown in Supplemental Figure 2 at potential temperature levels of 400 K, 425 K, 650 K, and 700 K (in addition to the 475 K level shown in Supplemental Figure 2) and the 36 sec^{-1} value for the vortex boundary appears valid. At the end of this review, we append figures showing maps at 400 and 650 K, for the four GCMs (EC-Earth3, MIROC6, MPI-ESM1-2-LR, and UKESM1-0-LL) highlighted in Supplemental Figure 2.

- I. 241: it would be good if the value $c(\theta)$ is available in the paper (e.g., for reproducing the results described here)

We have placed tabulated data on the web electronically in the data center Zenodo, with the identifier <http://doi.org/10.5281/zenodo.4414823>.

- I suggest to be more explicit here what is meant with ‘anomalous behaviour’.

The original text had read:

Values of $V_{PSC}(t)$ and $V_{VORTEX}(t)$ found using Eq. (2) and (3) as well as the ratio of these terms are shown in Supplementary Fig. 1; the anomalous behaviour of Arctic winter 2020, particularly after the vernal equinox, is readily apparent.

The text has been modified starting on line 387 to read:

Values of $V_{PSC}(t)$ and $V_{VORTEX}(t)$ found using Eq. (2) and (3) as well as the ratio of these terms are shown in Supplementary Fig. 1. The anomalous behaviour of Arctic winter 2020, such as record high values for V_{PSC} in March and V_{VORTEX} in March and April, is readily apparent.

- I. 248: what about the vertical resolution?

Details about the vertical resolution of the GCM output has been added, starting on line 380:

The GCM output is generally available on a daily basis, although some modelling groups have archived output every 6 hours; details are provided in Supplementary Table 3. The models

that archive output every 6 hours provide high model vertical resolution fields on the native model grid, whereas the daily output is generally provided for only a limited number of pressure levels (i.e., 100 hPa, 50 hPa, and 10 hPa). In cases where output for the SSP1-2.6, SSP2-4.5, and SSP3-7.0 scenarios are available only in low resolution (daily), we use low resolution for the SSP5-8.5 scenario from the corresponding GCM run, even if a higher resolution is available for SSP5-8.5.

- I. 273, 274: This is not clear to me, what is a “possible 1 K offset” ?

The original text had read:

These offsets have been determined based on the criterion that a trend line fit to the local maxima of PFP (PFP^{LM}) from the GCM over 1980-2020 using the ISA selection procedure (described below), should have a value in year 2000 (mid-point of the data record) that lies closest to the value of the fit to PFP^{LM} data from ERA5/ERA5.1 in year 2000, among all possible 1 K offsets to T_{NAT} (including no offset) ranging from -9 K to +9 K.

Here, we are attempting to describe that we compute 19 trend lines to PFP^{LM} from the output of each GCM: one for -9, -8, ... -1, 0, 1, ... 8, and 9 K offsets to the Y fields. We then choose the offset for which the value of the associated trend line for year 2000 lies closest to the year 2000 value of the trend line fit to ERA5/ERA5.1 data. We could have computed many more trend lines by using, for example, 0.1 K increments. We had discussed this possibility, but decided on 1 K increments for simplicity as well as use of an appropriate level of significant figures for the offset.

In the revised paper, we have added the word “incremental” (line 417), such that the last phrase reads “all possible 1 K incremental offsets”

- Fig. 1: the colour scale used here is not ideal, some of the years are difficult to distinguish

We have used a different color scale; along with three types of symbols for the data and a fourth for the model results. We hope various years can now be distinguished on Figure 1.

- Data availability: I suggest also to make the data (Ozone loss, PFP etc) use to produce the figures in the paper easily available as tables in the electronic supplement. This would considerably help making the results of this paper easily reproducible in follow up studies.

We have placed tabulated data on the web electronically in the data center Zenodo, with the identifier <http://doi.org/10.5281/zenodo.4414823>.

References

Chipperfield, M. P., Dhomse, S. S., Feng, W., McKenzie, R. L., Velders, G. J. M., and Pyle, J. A.: Quantifying the ozone and ultraviolet benefits already achieved by the Montreal Protocol, *Nat. Commun.*, 6, 7233, <https://doi.org/10.1038/ncomms8233>, 2015.

Dameris, M., Loyola, D. G., Nützel, M., Coldewey-Egbers, M., Lerot, C., Romahn, F., and van Roozendaal, M.: First description and classification of the ozone hole over the Arctic in boreal spring 2020, *Atmos. Chem. Phys. Discuss.*, 2020, 1-26, <https://doi.org/10.5194/acp-2020-746>, URL <https://acp.copernicus.org/preprints/acp-2020-746/>, 2020.

*Drdla, K. and Müller, R.: Temperature thresholds for chlorine activation and ozone loss in the polar stratosphere, *Ann. Geophys.*, 30, 1055-1073, <https://doi.org/10.5194/angeo-30-1055-2012>, 2012.

Fernandez, R. P., Kinnison, D. E., Lamarque, J.-F., Tilmes, S., and Saiz-Lopez, A.: Impact of biogenic very short-lived bromine on the Antarctic ozone hole during the 21st century, *Atmos. Chem. Phys.*, 17,

1673-1688, <https://doi.org/10.5194/acp-17-1673-2017>, URL <http://www.atmos-chem-phys.net/17/1673/2017/>, 2017.

Hoffmann, L., Spang, R., Orr, A., Alexander, M. J., Holt, L. A., and Stein, O.: A decadal satellite record of gravity wave activity in the lower stratosphere to study polar stratospheric cloud formation, *Atmos. Chem. Phys.*, 17, 2901-2920, <https://doi.org/10.5194/acp-17-2901-2017>, URL <http://www.atmos-chem-phys.net/17/2901/2017/>, 2017.

Hu, D., Guo, Y., and Guan, Z.: Recent Weakening in the Stratospheric Planetary Wave Intensity in Early Winter, *Geophys. Res. Lett.*, 46, 3953-3962, <https://doi.org/10.1029/2019GL082113>, URL <https://agupubs.onlinelibrary.wiley.com/doi/abs/10.1029/2019GL082113>, 2019.

Lait, L. R.: An alternative form for potential vorticity, *J. Atmos. Sci.*, 51, 1754- 1759, 1994.

Liu, M., Hu, D., and Zhang, F.: Connections Between Stratospheric Ozone Concentrations Over the Arctic and Sea Surface Temperatures in the North Pacific, *J. Geophys. Res.*, 125, e2019JD031 690, <https://doi.org/10.1029/2019JD031690>, URL <https://agupubs.onlinelibrary.wiley.com/doi/abs/10.1029/2019JD031690>, e2019JD031690 2019JD031690, 2020.

Molleker, S., Borrmann, S., Schlager, H., Luo, B., Frey, W., Klingebiel, M., Weigel, R., Ebert, M., Mitev, V., Matthey, R., Woiwode, W., Oelhaf, H., Dörnbrack, A., Stratmann, G., Grooß, J.-U., Günther, G., Vogel, B., Müller, R., Krämer, M., Meyer, J., and Cairo, F.: Microphysical properties of synoptic-scale polar stratospheric clouds: in situ measurements of unexpectedly large HNO₃-containing particles in the Arctic vortex, *Atmospheric Chemistry and Physics*, 14, 10 785-10 801, <https://doi.org/10.5194/acp-14-10785-2014>, URL <http://www.atmos-chem-phys.net/14/10785/2014/>, 2014.

Pitts, M. C., Poole, L. R., and Gonzalez, R.: Polar stratospheric cloud climatology based on CALIPSO spaceborne lidar measurements from 2006 to 2017, *Atmos. Chem. Phys.*, 18, 10 881-10 913, <https://doi.org/10.5194/acp-18-10881-2018>, URL <https://www.atmos-chem-phys.net/18/10881/2018/>, 2018.

*Rex, M., Salawitch, R. J., von der Gathen, P., Harris, N. R. P., Chipperfield, M. P., and Naujokat, B.: Arctic ozone loss and climate change, *Geophys. Res. Lett.*, 31, L04116, <https://doi.org/10.1029/2003GL018844>, 2004.

*Rex, M., Salawitch, R. J., Deckelmann, H., von der Gathen, P., Harris, N. R. P., Chipperfield, M. P., Naujokat, B., Reimer, E., Allaart, M., Andersen, S. B., Bevilacqua, R., Braathen, G. O., Claude, H., Davies, J., De Backer, H., Dier, H., Dorokov, V., Fast, H., Gerding, M., Godin-Beekmann, S., Hoppel, K., Johnson, B., Kyrö, E., Litynska, Z., Moore, D., Nakane, H., Parrondo, M. C., Risley Jr., A. D., Skrivanekova, P., Stübi, R., Viatte, P., Yushkov, V., and Zerefos, C.: Arctic winter 2005: Implications for stratospheric ozone loss and climate change, *Geophys. Res. Lett.*, 33, L23808, <https://doi.org/10.1029/2006GL026731>, 2006.

*Rieder, H. E. and Polvani, L. M.: Are recent Arctic ozone losses caused by increasing greenhouse gases?, *Geophys. Res. Lett.*, 40, 4437-4441, <https://doi.org/10.1002/grl.50835>, URL <https://agupubs.onlinelibrary.wiley.com/doi/abs/10.1002/grl.50835>, 2013.

*Rieder, H. E., Polvani, L. M., and Solomon, S.: Distinguishing the impacts of ozone-depleting substances and well-mixed greenhouse gases on Arctic stratospheric ozone and temperature trends, *Geophys. Res. Lett.*, 41, 2652-2660, <https://doi.org/10.1002/2014GL059367>, URL <https://agupubs.onlinelibrary.wiley.com/doi/abs/10.1002/2014GL059367>, 2014.

Rigby, M., Park, S., Saito, T., Western, L. M., Redington, A. L., Fang, X., Henne, S., Manning, A. J., Prinn, R. G., Dutton, G. S., Fraser, P. J., Ganesan, A. L., Hall, B. D., Harth, C. M., Kim, J., Kim, K.-R., Krummel, P. B., Lee, T., Li, S., Liang, Q., Lunt, M. F., Montzka, S. A., Mühle, J., ODoherty, S., Park, M.-K., Reimann, S., Salameh, P. K., Simmonds, P., Tunnicliffe, R. L., Weiss, R. F., Yokouchi, Y., and Young, D.: Increase in CFC-11 emissions from eastern China based on atmospheric observations, *Nature*, 569, 546-550, URL <https://doi.org/10.1038/s41586-019-1193-4>, 2019.

*Tilmes, S., Müller, R., Engel, A., Rex, M., and Russell III, J.: Chemical ozone loss in the Arctic and Antarctic stratosphere between 1992 and 2005, *Geophys. Res. Lett.*, 33, L20812, <https://doi.org/10.1029/2006GL026925>, 2006.

*Tilmes, S., Kinnison, D., Müller, R., Sassi, F., Marsh, D., Boville, B., and Garcia, R.: Evaluation of heterogeneous processes in the polar lower stratosphere in the Whole Atmosphere Community Climate Model, *J. Geophys. Res.*, 112, D24301, <https://doi.org/10.1029/2006JD008334>, 2007.

Waibel, A. E., Peter, T., Carslaw, K. S., Oelhaf, H., Wetzell, G., Crutzen, P. J., Pöschl, U., Tsias, A., Reimer, E., and Fischer, H.: Arctic Ozone Loss Due to Denitrification, *Science*, 283, 2064-2069, 1999.

Wegner, T., Groß, J.-U., von Hobe, M., Stroh, F., Sumińska-Ebersoldt, O., Volk, C. M., Hösen, E., Mitev, V., Shur, G., and Müller, R.: Heterogeneous chlorine activation on stratospheric aerosols and clouds in the Arctic polar vortex, *Atmos. Chem. Phys.*, 12, 11 095-11 106, <https://doi.org/10.5194/acp-12-11095-2012>, 2012.

Wegner, T., Pitts, M. C., Poole, L. R., Tritscher, I., Groß, J.-U., and Nakajima, H.: Vortex-wide chlorine activation by a mesoscale PSC event in the Arctic winter of 2009/10, *Atmos. Chem. Phys.*, 16, 4569-4577, <https://doi.org/10.5194/acp-16-4569-2016>, URL <https://www.atmos-phys.net/16/4569/2016/>, 2016.

Zhang, J., Tian, W., Xie, F., Pyle, J. A., Keeble, J., and Wang, T.: The Influence of Zonally Asymmetric Stratospheric Ozone Changes on the Arctic Polar Vortex Shift, *J. Climate*, 33, 4641-4658, <https://doi.org/10.1175/JCLI-D-19-0647.1>, URL <https://doi.org/10.1175/JCLI-D-19-0647.1>, 2020.

Much thanks for this nice list of citations. Upon revision, we have added citations to those papers marked with blue font.

References marked with an asterisk had been cited in the original submission; all have been retained.

Thanks again for such a helpful, careful review!

Please see caption of Supplemental Figure 2 for explanation of these figures.

Reviewer comments, second round

Reviewer #1 (Remarks to the Author):

As for my comments, the authors have extensively analyzed them and provided comprehensive answers to my review, modifying their work in accordance with those answers, so the resubmission, in this form, is perfectly satisfactory. The work also benefited from the comments of the other reviewers, some very acute, who improved its form and content. I believe it can therefore be published.

Reviewer #2 (Remarks to the Author):

The revised manuscript is very good and deserving of prompt publication in Nature Communications. The authors addressed all my comments very thoroughly. My main worry about the original draft was an insufficient discussion of the impacts on ozone. This has been completely remedied in the revised version that contains a very solid analysis of this aspect of the study and some important new results. I really appreciate the great effort that the authors put into addressing the reviewers' comments including mine, evident in their revision of the manuscript and in the extremely detailed responses. Congratulations on your interesting and important result! I recommend that the paper be published as is.

Kris Wargan (NASA GMAO, SSAI)

Reviewer #3 (Remarks to the Author):

[see next page]

Second review of “The influence of climate change on chemical loss ...” by P. von der Gathen et al.

General

The authors have revised the paper in response to the comments of all three reviewers; they have done a good job in responding to the comments. Moreover the paper has substantially changed for reasons explained in the reply; even a new co-author has joined the team.

But my overall conclusion has not changed: this is a strong paper that deserves to be published in *Nat. Commun.*; the message of the paper is that cold Arctic winters are getting colder has been further substantiated and the fact that models indicate (depending on the particular scenario considered) a substantial potential for ozone depletion at the end of the century (Fig. 9) is an important message.

One addition to the paper, which was not present in the previous version is the use of EESC^{1,2}·PFP; the unit of this quantity (e.g. Fig. 1) is ppb^{1,2}·d – I do not think this is a useful unit and suggest to revise this point in the final version of the paper (see below for details).

Overall, the methodology of the paper is sound and certainly meets the expected standards in the field. I might have raised more issues than usual in a re-review; however also the paper has changed a lot, so that this fact is perhaps less surprising. Thus I am still suggesting further revisions, but I also feel that this paper deserves to be published in *Nat. Commun.*.

Discussion

Existence and formation of PSCs

The concept of V_{PSC} is central to the paper; I agree that it should be based on the existence temperature for of polar stratospheric clouds. But sometime in this paper also the ‘formation’ of PSCs is mentioned; these two concepts are not the same thing. I suggest paying attention to the formulation throughout the paper; likely ‘existence’ is always what you want to say.

Nitric acid

Nitric acid (HNO_3) is an important compound for polar ozone loss. First, the vertical profile of HNO_3 in the polar vortex will certainly be altered by sedimentation of PSCs, also in the Arctic. At least when winters are cold (e.g., in 2020, Manney et al., 2020). And these are the winters with the strongest ozone loss obviously. Thus, a calculation of V_{PSC} that does not take into account denitrification (as it is done here) will during sufficiently cold periods not describe well the V_{PSC} in the real world. However, I suspect that the volume of air that supports substantial heterogeneous chemistry might be described quite well by this value, as the heterogeneous chemistry will often be driven by liquid particles, not by NAT. It is therefore not surprising to me (personally) that the conclusions you obtain for ARP are very similar to those for V_{PSC} (calculated ignoring denitrification). I am not sure how you will want to react on this comment (in terms of changes to the paper); perhaps not at all.

Further, in the future, HNO_3 in the stratosphere should increase because of the increase in N_2O (which is of course different in different scenarios). Estimating a rate of increase of 1 ppb/year and current tropospheric mixing ratios of about 333 ppb, one would expect by 2100 a increase in N_2O of about 25%. (Of course the increase depends on the particular scenario.) I suggest that this issue is also taken into account in the discussions here; the point of CH_4 is important, but this is not the only gas that changes in the next eighty years.

Good quantitative understanding

The paper states “. . . demonstrating good quantitative understanding of the factors that govern ΔO_3 for the Arctic stratosphere”. This is of course a judgement by the authors, but my recommendation would be to be more careful here. In my opinion more is required than an agreement within roughly one σ of the partial column loss in the lower stratosphere for four Arctic winters. (I say partial column assuming that the analysis is done as in previous work (Rex et al., 2004) for the vertical range 14-24 km, but this should be more clearly stated in the paper.) I think things like a comparison with ozone profiles, analysis of chlorine activation and deactivation, of the descent in the Arctic vortex, of differences between vortex core and the vortex edge etc. should be done, before such statements are made. I also note that the ATLAS model assumes a 3 K shift in temperature when calculating the Henry constant of HCl. But in the end it is a judgement.

The use of EESC

A new aspect of the revised version of this paper is the use of $\text{EESC}^{1.2} \cdot \text{PFP}$ to describe polar ozone loss. First, this quantity is relatively closely related to the potential for the activation of chlorine (PACl), which accounts (like $\text{EESC}^{1.2} \cdot \text{PFP}$) for year-to-year variations in temperature and vortex size as well as for the long-term development of the halogen content of the stratosphere (Tilmes et al., 2007, 2008). I suggest that this similarity is mentioned in the paper.

Second, currently the dimension of $\text{EESC}^{1.2} \cdot \text{PFP}$ is $\text{ppb}^{1.2} \cdot \text{d}$ – this is a really strange unit. I suggest to scale EESC with the maximum value, so that the scaling ranges between zero and one. This would be achieved when the quantity $(\text{EESC}/\text{EESC}_{\text{max}})^{1.2} \cdot \text{PFP}$ is considered. I think that also the values of this quantity make more sense (e.g., Fig. 1b). Perhaps you want to introduce a name for the quantity; then “potential for polar ozone loss” would be a possibility.

Finally, the determination of the exponent 1.2 is based on the analysis shown in suppl. Fig. 11: how robust is this analysis? After all, the driving force behind the choice of 1.2 is the simulation of 6 Arctic winters by ATLAS, where I am sure the EESC dependence is driven by the simulations for 2060 and 2100. How would a figure like suppl. Fig. 11 look if done for a CMIP6 model (with full chemistry)?

Points from the first review

Regarding the warm winters I suggested that the ISA method could be also applied to the warm winters, not only the cold winters. My thinking here was

that such an analysis could strengthen the concept of the ISA method in general, and also the link to a rise in GHGs in particular. Such a result would also be of interest to studies addressing the impact of GHGs on the polar winter stratosphere in general. But having said all this, this point in the end is a matter of judgement by the authors.

Regarding the scaling of PV (figure enclosed in the reply); I am not sure what is shown in this figure. I presume that for the Lait (1994) line the dimensionless scaling factor $(\theta/\theta_{\text{ref}})^{-9/2}$ is shown. However, Rex et al. (1999) use normalised PV, with the unit s^{-1} , so that the PV scaling factor cannot be dimensionless. Further, the sensitivity on the vortex edge value of 36 s^{-1} could be tested. No big issue but this point could be clarified.

I also mentioned p-values: I still think that a bit more explanation (or definition?) would be helpful; if under pressure for space, such things could be discussed in the supplement.

What I mean is something along these lines: “A p-value is calculated using the sampling distribution of a given test statistic; p-values range from 0 (no chance) to 1 (absolute certainty). For example a p-value of 0.5 means a 50 % chance”

Seventy References

Again, references cited in this review (or in the first review) were used to support the points I am making; I do *not* mean to suggest that a particular reference should be cited in the manuscript. And I understand there are limits for references in *Nat. Commun.*. But perhaps adjustments with references are still possible. For example, I note that you currently have two references for NCEP (29,30) and two for MERRA-2 (31,32) – I think for such a paper one citation for MERRA-2 and NCEP each is sufficient.

Details

- l 25: you define here the concept of V_{PSC} based on the ‘existence’ of PSCs; I would agree. However elsewhere (e.g., l. 45,71) you use ‘formation’. These two terms should be used in a consistent way.
- l. 29: amounts \longrightarrow amount
- l. 33 present \longrightarrow 2005 (present is 2021, yes?)
- l. 38: suggested that
- l. 58: strictly speaking, activation may also occur on ice particles ...
- l. 59: which is the second type of PSCs? Perhaps rephrase.
- l. 68: The profile of HNO_3 *will* be altered by sedimentation of PSCs, also in the Arctic, if the winters are cold (e.g. refs 23-25 in the paper).
- l. 113-114: Are all these simulations done with the same version of the ATLAS model (ERA5, JPL2015 etc). Of course this would be desirable. If yes (this is how I read the paper), how different are the ATLAS results reported here from the ozone loss estimates reported in earlier ATLAS studies (e.g., Wohltmann et al., 2013)?

- l. 120: ATLAS points do not start in 1993
- l. 132: why 1.2 as the exponent?
- l. 136: linear relation of what? ozone depletion?
- l. 145: quantify how much larger
- l. 161: not sure if it makes sense to abbreviate Monte-Carlo with MC; see also the discussion on p-value
- l. 171: amount \longrightarrow number
- l. 187: not sure if this support is ‘theoretical’ or rather ‘numerical’ (also l. 199)
- l. 193: I suggest avoiding too many acronyms; FDF is an example for an acronym I would drop
- l. 203: might be helpful to indicate clearly, which lines you refer to here in the plots in Figs. 4 and 5
- l. 234: HNO₃ will also change in the future; see discussion.
- l. 245: Indeed the station at Boulder is not a good representation of a zonal average, let alone polar averages. But this is not what you want to discuss here.
- l. 288: EESC^{1,2}. PFP at the end of the century is not higher than contemporary values for all scenarios, is it? (See the two bottom panels of Fig. 9.)
- l. 321-325: I do not quite understand the point of “prescribed ozone”. When ozone is prescribed, the ΔO_3 is known before the run. In other words, if I prescribe the ozone from one of the full chemistry GCMs, the resulting ozone will be the same? How are the ozone values ‘prescribed’? I would not put much weight on simulations with prescribed ozone.
- l. 334: Figs. 8 and 9 do not show ΔO_3
- l. 337: planetary wave activity (and change) is not independent of the BDC; also gravity waves play a role to some extent (e.g., Li et al., 2008; Bunzel and Schmidt, 2013).
- l. 347: continue \longrightarrow continues
- l. 350: but in very cold Arctic winters (e.g., 2020) there could be dehydration also in the Arctic.
- l. 355: there is also denitrification in the Arctic, see discussion
- l. 370: one could check the sensitivity of the results on the choice of 36 s⁻¹; this might also address the ‘capping’ issue mentioned in l. 379
- l. ‘rarely present’ is likely too strong, I think they were present but the PFP values were less pronounced

- l. 472: Perhaps this calculation could be better explained. Where do the values 13.6 d and 4.59 d decade⁻¹ come from? What is the advantage of this calculation compared to the statistical significance calculation below?
- l. 497-500: I agree, but Fig. 3 only shows the fitting uncertainty. Could you include this statistical significance information also in Fig. 3?
- l. 516: can you quantify ‘very few’, perhaps a value in %?
- l. 544: what is assumed here as sulphate area density? Are these numbers available in the data archive created by the authors?
- l. 555: Typo: two times PFP here?
- l. 675: Cl is also changing with time
- l. 582: in which way an outlier?
- l. 597: how similar? I assume all runs were performed based on ERA5. Perhaps a bit more information could be given here.
- l. 600: ClONO₂ is from a climatology and HCl from observations – But then total chlorine is not correct in every case. Or do you adjust the values of the climatology? Could you provide a bit more background here?
- l. 604: Descent rates are important for polar ozone loss (e.g., Strahan et al., 2014), how are they calculated in ATLAS for the runs discussed here?
- l. 606: citations for this ‘common deficiency’?
- l. 973: You could also put the code on your Zenodo archive?
- Figure 1: Here (and perhaps elsewhere in the paper) it should be explained that the column ozone loss shown here is calculated for the partial column 14-24 km. Assuming that is the case – I am guessing here on the basis of previous publications.

References

- Bunzel, F. and Schmidt, H.: The Brewer-Dobson Circulation in a Changing Climate: Impact of the Model Configuration, *J. Atmos. Sci.*, 70, 1437–1455, <https://doi.org/10.1175/JAS-D-12-0215.1>, 2013.
- Lait, L. R.: An alternative form for potential vorticity, *J. Atmos. Sci.*, 51, 1754–1759, 1994.
- Li, F., Austin, J., and Wilson, J.: The Strength of the Brewer-Dobson Circulation in a Changing Climate: Coupled Chemistry-Climate Model Simulations, *J. Climate*, 21, 40 – 57, <https://doi.org/10.1175/2007JCLI1663.1>, 2008.

- Manney, G. L., Livesey, N. J., Santee, M. L., Froidevaux, L., Lambert, A., Lawrence, Z. D., Milln, L. F., Neu, J. L., Read, W. G., Schwartz, M. J., and Fuller, R. A.: Record-Low Arctic Stratospheric Ozone in 2020: MLS Observations of Chemical Processes and Comparisons With Previous Extreme Winters, *Geophys. Res. Lett.*, 47, e2020GL089063, <https://doi.org/https://doi.org/10.1029/2020GL089063>, 2020.
- Rex, M., von der Gathen, P., Braathen, G. O., Reid, S. J., Harris, N. R. P., Chipperfield, M., Reimer, E., Beck, A., Alfier, R., Krüger-Carstensen, R., De Backer, J., Balis, D., Zerefos, Z., O' Connor, F., Dier, H., Dorokhov, V., Fast, H., Gamma, A., Gil, M., Kyrö, E., Rummukainen, M., Litynska, Z., Mikkelsen, I. S., Molyneux, M., and Murphy, G.: Chemical Ozone Loss in the Arctic Winter 1994/95 as determined by the Match Technique, *J. Atmos. Chem.*, 32, 1–34, 1999.
- Rex, M., Salawitch, R. J., von der Gathen, P., Harris, N. R. P., Chipperfield, M. P., and Naujokat, B.: Arctic ozone loss and climate change, *Geophys. Res. Lett.*, 31, L04116, <https://doi.org/https://doi.org/10.1029/2003GL018844>, 2004.
- Strahan, S. E., Douglass, A. R., Newman, P. A., and Steenrod, S. D.: Inorganic chlorine variability in the Antarctic vortex and implications for ozone recovery, *J. Geophys. Res.*, <https://doi.org/https://doi.org/10.1002/2014JD022295>, URL <http://dx.doi.org/10.1002/2014JD022295>, 2014.
- Tilmes, S., Kinnison, D., Müller, R., Sassi, F., Marsh, D., Boville, B., and Garcia, R.: Evaluation of heterogeneous processes in the polar lower stratosphere in the Whole Atmosphere Community Climate Model, *J. Geophys. Res.*, 112, D24301, <https://doi.org/https://doi.org/10.1029/2006JD008334>, 2007.
- Tilmes, S., Müller, R., and Salawitch, R. J.: The sensitivity of polar ozone depletion to proposed geoengineering schemes, *Science*, 320, 1201–1204, <https://doi.org/https://doi.org/10.1126/science.1153966>, 2008.
- Wohltmann, I., Wegner, T., Müller, R., Lehmann, R., Rex, M., Manney, G. L., Santee, M. L., Bernath, P., Sumińska-Ebersoldt, O., Stroh, F., von Hobe, M., Volk, C. M., Hösen, E., Ravegnani, F., Ulanovsky, A., and Yushkov, V.: Uncertainties in modelling heterogeneous chemistry and Arctic ozone depletion in the winter 2009/2010, *Atmos. Chem. Phys.*, 13, 3909–3929, 2013.

Reviewer #3 Comments:

General

The authors have revised the paper in response to the comments of all three reviewers; they have done a good job in responding to the comments. Moreover the paper has substantially changed for reasons explained in the reply; even a new co-author has joined the team.

But my overall conclusion has not changed: this is a strong paper that deserves to be published in *Nat. Commun.*; the message of the paper is that cold Arctic winters are getting colder has been further substantiated and the fact that models indicate (depending on the particular scenario considered) a substantial potential for ozone depletion at the end of the century (Fig. 9) is an important message.

One addition to the paper, which was not present in the previous version is the use of EESC^{1,2} PFP; the unit of this quantity (e.g. Fig. 1) is ppb^{1,2} d : I do not think this is a useful unit and suggest to revise this point in the final version of the paper (see below for details).

Overall, the methodology of the paper is sound and certainly meets the expected standards in the field. I might have raised more issues than usual in a re-review; however also the paper has changed a lot, so that this fact is perhaps less surprising. Thus I am still suggesting further revisions, but I also feel that this paper deserves to be published in *Nat. Commun.*

Thanks for the kind words and your careful review.

As detailed below, we will normalize the new term by EESC_{MAX}^{1,2}, such that units will be days.

Another very important change to our paper is that, since the time of the past review, ERA5 has now released a reanalysis of the fully resolved stratosphere at model levels (this dataset has high resolution in the stratosphere) going back to 1950. At the time of our first revision, ERA5 had only released the reanalysis back to 1950 for pressure levels with low resolution in the stratosphere. This new data set, called ERA5 back extension (BE) (preliminary version) as described at:

<https://cds.climate.copernicus.eu/cdsapp#!/dataset/reanalysis-era5-single-levels-preliminary-back-extension?tab=overview>

allows us to extend the analysis of PFP based on ERA5 further back in time as suggested in a former review.

We have updated Figure 3 to include an analysis of the local maximum in PFP from both ERA5 and JRA-55 over 1965 to 2020. Now, the corresponding introduction of this time range for both data sets, starting on line 85, reads

Meteorological fields from ERA5 have recently been extended back to 1950 and data from JRA-55 are available from 1958 to 2020, whereas the other data sets are available from 1979 (or 1980) to 2020. Stratospheric data in the Arctic mainly rely on radiosonde soundings before 1979 and on satellite data thereafter, which could introduce potential bias (see Methods). We use ERA5 and JRA-55 only back to 1965 since this year marks the start of regular radiosonde coverage of the Arctic stratosphere.

Note: here and throughout, line numbers refer to the version of the documents for which all changes have been accepted.

Discussion

Existence and formation of PSCs

The concept of V_{PSC} is central to the paper; I agree that it should be based on the existence temperature for polar stratospheric clouds. But sometime in this paper also the 'formation' of PSCs is mentioned; these two concepts are not the same thing. I suggest paying attention to the formulation throughout the paper; likely 'existence' is always what you want to say.

Great point.

In the version of the manuscript for which this review is based, the word “existence” was used twice: once in main and once in methods.

Upon revision, we now use “existence” 11 times: 5 in main and 6 in methods.

Nitric acid

Nitric acid (HNO_3) is an important compound for polar ozone loss. First, the vertical profile of HNO_3 in the polar vortex will certainly be altered by sedimentation of PSCs, also in the Arctic. At least when winters are cold (e.g., in 2020, Manney et al., 2020). And these are the winters with the strongest ozone loss obviously. Thus, a calculation of V_{PSC} that does not take into account denitrification (as it is done here) will during sufficiently cold periods not describe well the V_{PSC} in the real world. However, I suspect that the volume of air that supports substantial heterogeneous chemistry might be described quite well by this value, as the heterogeneous chemistry will often be driven by liquid particles, not by NAT. It is therefore not surprising to me (personally) that the conclusions you obtain for ARP are very similar to those for V_{PSC} (calculated ignoring denitrification). I am not sure how you will want to react on this comment (in terms of changes to the paper); perhaps not at all.

We have not changed our formulation of T_{NAT} to account for denitrification because, as stated by the reviewer, the volume of air for which chlorine is activated by heterogeneous chemistry is well described by the formulation of T_{NAT} that is used in the paper.

The comparison of three figures in the paper (Figs. 1, 3, and 5) based on T_{NAT} to comparable figures found using T_{ACL} , which is independent of gas phase HNO_3 , supports the validity of our approach.

We have added the following sentences to Methods, starting in line 571, to clarify this point:

In the actual Arctic stratosphere, denitrification (the removal of HNO_3 by the physical sedimentation of PSCs) will prolong ozone loss²⁷ and alter T_{NAT} due to suppression of gas phase HNO_3 (ref⁵⁹). However, the volume of air for which chlorine is activated by heterogeneous chemistry is governed most strongly by temperature. The close visual relation between Figs. 1, 3 and 5 and Supplementary Figs. 9, 10, and 11 supports the validity of the definition of OLP used in the main paper, which does not explicitly represent denitrification for the computation of T_{NAT} .

with citations to:

27. Waibel, A. E. et al. Arctic ozone loss due to denitrification. *Science* 283, 2064–2069 (1999).

59. Santee, M. L. et al. Interhemispheric differences in polar stratospheric HNO_3 , H_2O , ClO , and O_3 . *Science* 267, 849–852 (1995).

Further, in the future, HNO_3 in the stratosphere should increase because of the increase in N_2O (which is of course different in different scenarios). Estimating a rate of increase of 1 ppb/year and current tropospheric mixing ratios of about 333 ppb, one would expect by 2100 an increase in N_2O of about 25%. (Of course the increase depends on the particular scenario.) I suggest that this issue is also taken into account in the discussions here; the point of CH_4 is important, but this is not the only gas that changes in the next eighty years

We have addressed by adding the following text starting at line 338:

Finally, future levels of N_2O are expected to rise⁴³, leading to higher levels of HNO_3 that will lead to more favourable conditions for the formation of PSCs²².

with citations to:

22. Hanson, D. & Mauersberger, K. Laboratory studies of the nitric acid trihydrate: Implications for the south polar stratosphere. *Geophys. Res. Lett.* 15, 855–858 (1988).

43. O’Neill, B. C. et al. The Scenario Model Intercomparison Project (ScenarioMIP) for CMIP6. *Geosci. Model Dev.* 9, 3461–3482 (2016).

Good quantitative understanding

The paper states “. . . demonstrating good quantitative understanding of the factors that govern ΔO_3 for the Arctic stratosphere". This is of course a judgement by the authors, but my recommendation would be to be more careful here. In my opinion more is required than an agreement within roughly one σ of the partial column loss in the lower stratosphere for four Arctic winters. (I say partial column assuming that the analysis is done as in previous work (Rex et al., 2004) for the vertical range 14-24 km, but this should be more clearly stated in the paper.) I think things like a comparison with ozone profiles, analysis of chlorine activation and deactivation, of the descent in the Arctic vortex, of differences between vortex core and the vortex edge etc. should be done, before such statements are made. I also note that the ATLAS model assumes a 3 K shift in temperature when calculating the Henry constant of HCl. But in the end it is a judgement.

We have changed the phrase

demonstrating good quantitative understanding of the factors that govern ΔO_3 for the Arctic stratosphere.

that now appears on line 122, to read:

demonstrating that the primary control on interannual variations in ΔO_3 over the past 15 years has been the exposure of air to PSC temperatures.

This new phrase relates more directly to the data shown in Fig. 1a.

--

Throughout the column loss of ozone for both the ozonesondes and the ATLAS model has been calculated for the partial column from 380-550 K, which is approximately 14-24 km. Thanks so much for pointing out our failure to provide this detail.

On line 106 of the main text, we now state:

Figure 1a shows values of column ozone loss between 380 and 550 K potential temperature (ΔO_3) ...

and we have also added appropriate detail to the caption of Figure 1. We realized in the midst of this revision that we had been using values of column ozone loss from ATLAS for 370 to 550 K, rather than 380 to 550 K. We have changed the ATLAS results so that ΔO_3 is for 380 to 550 K; there is a slight shift downwards in the ATLAS points on Figure 1 as a result of this change. The shift is small relative to the error bars; 380 K is the best lower altitude limit for the sonde analysis.

--

Due to limitations for the length of the manuscript and the number of citations, we are not able to provide a detailed discussion of the quality of the model simulations of O_3 loss and the various ways the ATLAS model has been evaluated with data. We have a paper under review at *Journal of Geophysical Research (JGR)* that examines some of the issues raised above. Below, we summarize highlights from some of our prior papers as well as the paper currently under review at *JGR*.

Wohltmann et al., *GMD* (2010) show detailed comparisons to ozone sondes, balloon and ER2 aircraft data from the SOLVE campaign and the HALOE satellite for 1999/2000 (Fig. 4, 6-11, supplement Figs. 1-52, 54). Wohltmann et al. *ACP* (2013) shows detailed comparisons to ozone sondes, Geophysica aircraft measurements, and ACE-FTS and MLS satellite measurements for 2009/2010 (Figure 2, Figure 3, supplement Figure 1-17). Wohltmann et al. *ACP* (2017) showed detailed comparisons between measurements from MLS and ACE-FTS and the model for four different years (Figures 145-172 in the supplement and Figure 21, 22 in the main paper). Wohltmann et al. *JGR* (under review, 2021) shows comparisons to ozone sondes and MLS for the 2019/2020 run used also in this manuscript. These comparisons cover most of the species modelled by ATLAS, e.g. CFCs, CH_4 , N_2O , H_2O , CO_2 , O_3 , ClO, Cl_2O_2 , HCl, $ClONO_2$, HNO_3 , NO, NO_2 , N_2O_5 , OH, HO_2 .

All of the studies mentioned above contain analysis of chlorine activation and deactivation. In particular, a detailed account can be found in Wohltmann et al. *ACP* (2017) and Wohltmann et al. *JGR* (2021). In general, the comparisons show a reasonable agreement to measurements and don't suggest any lack of understanding of the stratospheric processes nor any severe model deficiencies, other than for HCl

(described in detail below, in reply to comment regarding line 606). Ozone mixing ratios typically compare to within 20% with the measurements, except for times when ozone mixing ratios are exceptionally low.

The figure shown below compares our modeled N_2O (black) to MLS measurements of N_2O (red, with 1σ accuracy uncertainty of the retrieval), averaged within the Arctic vortex at 54 hPa for the four prior winters analyzed in our paper.

Overall there is quite good agreement between modeled and measured N_2O , supporting the realistic representation of vertical descent within the model. As the reviewer may know, the MLS retrievals of N_2O suffered from the loss of a channel in 2015; all of the retrievals shown above are for the new Version 4 algorithm, which has larger uncertainty than the previously used Version 3 algorithm. The most notable discrepancy shown above, the possible tendency for ATLAS to underestimate descent within the Arctic vortex during 2019/2020, is an area of active research. Nonetheless, the modeled and measured values of N_2O agree to within 1σ of the measurement uncertainty for nearly every day of all four years.

The use of EESC

a) A new aspect of the revised version of this paper is the use of $EESC^{1.2} \cdot PFP$ to describe polar ozone loss. First, this quantity is relatively closely related to the potential for the activation of chlorine (PACl), which accounts (like $EESC^{1.2} \cdot PFP$) for year-to-year variations in temperature and vortex size as well as for the long-term development of the halogen content of the stratosphere (Tilmes et al., 2007, 2008). I suggest that this similarity is mentioned in the paper.

b) Second, currently the dimension of $EESC^{1.2} \cdot PFP$ is $ppb^{1.2} d$ – this is a really strange unit. I suggest to scale EESC with the maximum value, so that the scaling ranges between zero and one. This would be achieved when the quantity $(EESC/EESC_{max})^{1.2} \cdot PFP$ is considered. I think that also the values of this quantity make more sense (e.g., Fig. 1b). Perhaps you want to introduce a name for the quantity; then “potential for polar ozone loss” would be a possibility.

We have addressed both a) & b) by: 1) normalizing the key quantity as suggested; 2) naming the quantity Ozone Loss Potential (OLP), and 3) noting the similarity to PACl from Tilmes, S. et al., *JGR*, 2007.

The key new text starting on line 131 reads:

Figure 1b shows measured and modelled values of ΔO_3 as a function of a term we shall refer to as Ozone Loss Potential (OLP), defined as:

$$OLP (yr) = \frac{EESC(yr)^{1.2}}{EESC_{MAX}^{1.2}} \times PFP (yr) \quad (2)$$

where $EESC_{MAX}$ (4.45 ppbv) is the maximum yearly value of EESC in the polar stratosphere. ... Our OLP is defined in a manner nearly identical to the potential for activation of chlorine term of Tilmes et al.²⁸, except for our use of 1.2 rather than 1 as the exponent of EESC in Eq. (2).

with citation to:

28. Tilmes, S. et al. Evaluation of heterogeneous processes in the polar lower stratosphere in the Whole Atmosphere Community Climate Model. *J. Geophys. Res. Atmos.* 112, D24301 (2007).

c) Finally, the determination of the exponent 1.2 is based on the analysis shown in suppl. Fig. 11: how robust is this analysis? After all, the driving force behind the choice of 1.2 is the simulation of 6 Arctic winters by ATLAS, where I am sure the EESC dependence is driven by the simulations for 2060 and 2100. How would a figure like suppl. Fig. 11 look if done for a CMIP6 model (with full chemistry)?

We have added a sentence to the paper on line 146 that reads:

We assess the uncertainty in ΔO_3^{REG} using lower and upper limits of 1 and 1.4 for the exponent in the expression for OLP (see Methods).

and have added the following text to the Methods section starting on line 654, supporting the use of 1 and 1.4 as lower and upper limits for the exponent, respectively:

Exponent for EESC. In the main article we assess the uncertainty in ΔO_3^{REG} using lower and upper limits of 1 and 1.4 as the exponent for EESC in the expression for OLP. The lower limit of 1 corresponds to a linear dependence of chemical loss of Arctic O_3 on EESC, based upon the work of Douglass et al.⁶⁹ who showed that ΔO_3 for the Arctic vortex varies linearly with EESC for fixed values of V_{PSC} , for values of EESC spanning 1990 to 2016. The upper limit of 1.4 was chosen because r^2 has the same value for $\eta=1$ and $\eta=1.4$ in Supplementary Fig. 11, and also because Jiang et al.⁷⁰ showed that the variation of the chemical loss of Antarctic ozone was shown to vary as a function of chlorine loading to the power of 1.4 for 1980 to 1990, a period of rapid rise in the chlorine component of EESC.

with citation to:

69. Douglass, A. R., Stolarski, R. S., Strahan, S. E. & Polansky, B. C. Sensitivity of Arctic ozone loss to polar stratospheric cloud volume and chlorine and bromine loading in a chemistry and transport model. *Geophys. Res. Lett.* 33, L17809 (2006).

70. Jiang, Y., Yung, Y. L. & Zurek, R. W. Decadal evolution of the Antarctic ozone hole. *J. Geophys. Res. Atmos.* 101, 8985–8999 (1996).

In terms of the question “How would a figure like Suppl. Fig. 11 look if done for a CMIP6 model (with full chemistry)?”, the answer to this question is beyond the scope of this paper, as such a figure cannot be formed from the GCM output currently available on the CMIP6 archive. One would need to perform full ozone loss analyses. No daily ozone values are available from the data base. Additionally, one would need tracers such as N_2O , and perhaps CH_4 , as well as inorganic chlorine and bromine levels in the stratosphere to investigate this question. Perhaps our paper will motivate further study of the exponent among the Coupled Chemistry-Climate Model initiative community.

Points from the first review

Regarding the warm winters I suggested that the ISA method could be also applied to the warm winters, not only the cold winters. My thinking here was that such an analysis could strengthen the concept of the ISA method in general, and also the link to a rise in GHGs in particularly. Such a result would also be of interest to studies addressing the impact of GHGs on the polar winter stratosphere in general. But having said all this, this point in the end is a matter of judgement by the authors.

We appreciate the effort of the reviewer to strengthen our analysis. However, we have decided to not address this any further, as explained below. The key sentence starting on line 207 reads:

The GCM simulations in Figs. 4 and 5 also show a tendency for PFP associated with the warmer Arctic winters (open circles at bottom of the data envelope) to rise slightly over time, a projected trend not yet apparent in observations⁴⁸ due perhaps to the generally small values of PFP for the warmest winters over the observational period as well as the lower limit of zero for PFP.

with reference to:

48. Rieder, H. E., Polvani, L. M. & Solomon, S. Distinguishing the impacts of ozone-depleting substances and well-mixed greenhouse gases on Arctic stratospheric ozone and temperature trends. *Geophys. Res. Lett.* 41, 2652–2660 (2014).

To provide illustrative support for the importance of the phrase “lower limit of zero for PFP”, and to support our decision to not further probe trends in the warmest winters using PFP, we include the figure shown below. The bottom row shows the trend in the local minimum of PFP found by applying ISA to the output of four GCMs constrained by SSP5-8.5, for the same PSC thresholds (i.e., 3 K, 3K, 4K, and 1K) used in Figure 5. The data points selected by ISA are shown as solid red circles, and the linear fits (and uncertainty) to these selected points are shown by the solid (and dashed) lines. These shifted thresholds are designed to assure statistical similarity of PFP between GCM output and ERA5. All trend lines are slightly positive, which would suggest slight cooling of the warmest winters.

The top row of the figure shows the resulting trend in the local minimum in PFP upon application of an additional 3 K threshold offset. As shown, each GCM exhibits a clear, dramatic rise in the local minima of PFP, which means the warmest winters are also cooling. The problem with this interpretation is the statistical distribution of PFP from these four GCMs no longer resembles the ERA5 distribution; the reason for the large difference in apparent trend between the top and bottom row is that PFP only

responds to a change in temperature the times when temperature falls below the PSC existence threshold.

While we understand the interest in the warmest winters, which is more pressing than ever given the sudden stratospheric warming of 2021, we choose to steer clear of any discussion of the warmest winters, for the reasons noted above. Such an analysis is best conducted using temperature, rather than PFP, or perhaps some combination of temperature and PFP.

Regarding the scaling of PV (figure enclosed in the reply); I am not sure what is shown in this figure. I presume that for the Lait (1994) line the dimensionless scaling factor $(\theta = \theta_{ref})^{-9/2}$ is shown. However, Rex et al. (1999) use normalised PV, with the unit s^{-1} , so that the PV scaling factor cannot be dimensionless.

Sorry the figure used in the prior reply was not properly labelled. The figure below compares the scaling factors from Rex et al. (1999) to that used by Lait (1994), with proper units:

As noted in our prior review, these two scaling factors agree quite well; the same scientific conclusion would be reached if either was used in the analysis.

Further, the sensitivity on the vortex edge value of $36 s^{-1}$ could be tested. No big issue but this point could be clarified.

We have added a new section to Methods entitled “**Vortex Boundary**” as well as a new figure to the supplement, included just below for convenience, to address this comment.

The new section that starts on line 537 reads as follows:

Vortex boundary. The vortex boundary is based on the value of $36 sec^{-1}$ for nPV. This definition of the vortex boundary is commonly used in other studies of Arctic ozone, because $nPV = 36 sec^{-1}$ tends to be closely associated with the maximum, horizontal gradient of potential vorticity^{13,36,67}. To check whether other definitions of the vortex boundary would yield insignificant results Supplementary Figure 7 shows trends in S_{PFP-LM} found by the ISA algorithm applied to data from ERA5/ERA5.1 combined with ERA5 BE (preliminary version) from 1965 to 2020 for four alternate definitions of the vortex boundary, along with the resulting trends and p-values for the quantity $S_{PFP-LM}/\Delta S_{PFP-LM}$. For each alternate vortex boundary definition, the resulting trends in S_{PFP-LM} are positive and highly statistically significant. The numerical values for PFP do vary based on how

the boundary is specified, and vary from those shown in Fig. 3b of the main paper, due largely to the use of the volume of the Arctic vortex in the denominator of the definition of PFP (Eq. 1).

with citations to:

13. Wohltmann, I. *et al.* Near-complete local reduction of Arctic stratospheric ozone by severe chemical loss in spring 2020. *Geophys. Res. Lett.* 47, e2020GL089547 (2020).

36. Rex, M. *et al.* Chemical ozone loss in the Arctic winter 1994/95 as determined by the Match technique. *J. Atmos. Chem.* 32, 35–59 (1999).

67. Wohltmann, I., Lehmann, R. & Rex, M. A quantitative analysis of the reactions involved in stratospheric ozone depletion in the polar vortex core. *Atmos. Chem. Phys.* 17, 10535–10563 (2017).

Here is the new supplementary Fig. 7, designed to be compared to Fig. 3b of the main paper.

Streibel *et al.*, *ACP*, 6, 2783-2792 (2006) and Grooß *et al.*, *ACP*, 8, 565-578 (2008) are among the many other papers that also use $nPV = 36$ sec⁻¹ as the vortex boundary.

Finally, as had been stated in second paragraph of Methods, and is still stated starting on line 357:

Our analysis requires definition of the area and volume of the Arctic polar vortex, denoted A_{VORTEX} and V_{VORTEX} . The value of 36 sec⁻¹ for normalized PV (nPV) is used to define the edge of the polar vortex, as described in section 3.3 of Rex *et al.*³⁶. Other studies utilize the maximum gradient in PV to define the boundary of the polar vortex⁶¹. We use $nPV = 36$ sec⁻¹ to define the vortex boundary because on some days the gradient method introduces a level of complexity, due to the existence of multiple maximum gradients of nearly equal magnitude separated by considerable distance, that requires human judgement. We have examined maps of nPV and temperature plotted for 1 February of the years 1960 to 2100, in increments of every 10 years, for all 26 CMIP6 GCMs that archived results for SSP5-8.5. These maps show that the $nPV = 36$ sec⁻¹ boundary for the Arctic vortex is not greatly affected by climate change until the end of century; maps for the four CMIP6 GCMs highlighted in Fig. 5 of the paper are

shown in Supplementary Fig. 2. Since PV from four reanalyses that span many decades and model output from 53 GCM simulations that span more than a century and a half are examined, it is preferable to implement a method that requires no human intervention.

I also mentioned p-values: I still think that a bit more explanation (or definition?) would be helpful; if under pressure for space, such things could be discussed in the supplement.

What I mean is something along these lines: “p-value is calculated using the sampling distribution of a given test statistic; p-values range from 0 (no chance) to 1 (absolute certainty). For example, a p-value of 0.5 means a 50 % chance”

We feel we had addressed this point in the prior revision, which may have been overlooked by the reviewer. The relevant text starting on line 511 of the latest version of the paper reads as follows:

The p-values given in Table 1 for $S_{\text{PFP-LM}}$ are equal to the probability that the slope of these random fits exceeds the slope determined from the data. In other words, 18% of the randomly generated combinations of PFP for the ERA5/ERA5.1 basis set (over the 1980 to 2020 time period) yield a value for $S_{\text{PFP-LM}}$ larger than $4.50 \text{ d decade}^{-1}$.

Later on, starting on line 521 the p-value for the quantity $S_{\text{PFP-LM}}/\Delta S_{\text{PFP-LM}}$ is addressed:

We therefore examine the quantity $S_{\text{PFP-LM}}/\Delta S_{\text{PFP-LM}}$ as a measure of the statistical significance of both the temporal rise in PFP^{LM} as well as the uncertainty in this rise. Of the randomly generated time series, 99.992% yield a value of $S_{\text{PFP-LM}}/\Delta S_{\text{PFP-LM}}$ that is smaller than the actual value of 23.6 ($4.50 \text{ d decade}^{-1}$ divided by $0.19 \text{ d decade}^{-1}$). Consequently, a p-value of 8×10^{-5} is associated with the entry for $S_{\text{PFP-LM}}/\Delta S_{\text{PFP-LM}}$ based upon ERA5/ERS5.1 data in Table 1 ...

Seventy References

Again, references cited in this review (or in the first review) were used to support the points I am making; I do not mean to suggest that a particular reference should be cited in the manuscript. And I understand there are limits for references in Nat. Commun. But perhaps adjustments with references are still possible. For example, I note that you currently have two references for NCEP (29,30) and two for MERRA-2 (31,32) – I think for such a paper one citation for MERRA-2 and NCEP each is sufficient.

For NCEP, we are obligated to use two citations because reference 29 (now 31) is for the Climate Forecast System Reanalysis (CFSR) that covers 1 Jan 1979 to 31 March 2011, whereas reference 30 (now 32) is for the NCEP Climate Forecast System (CFSv2) that covers 1 April 2011 to present, as explained at:

<https://www.ncdc.noaa.gov/data-access/model-data/model-datasets/climate-forecast-system-version2-cfsv2>

We have consistently referred to the NCEP-based values of PFP as “CFSR/CFSv2” to reflect the use of both datasets.

For MERRA-2, reference 31 (now 33) is for the peer-reviewed paper and reference 32 (now 34) refers to the dataset. One might think these two references can be combined. However, our examination of references 37 & 38 for GRACE data used in this 2019 *Nature Communications* paper:

<https://www.nature.com/articles/s41467-019-11097-w>

by Paul Palmer et al. shows the journal convention is to not combine into a single reference. Here is a screen capture of references 37 & 38 from Palmer et al. (2019), to the same (GRACE) data source:

37. Swenson, S. & Wahr, J. Post-processing removal of correlated errors in GRACE data. *Geophys. Res. Lett.* **33**, L08402 (2006).
38. Swenson, S. C. GRACE monthly land water mass grids NETCDF RELEASE 5.0. Ver. 5.0. PO.DAAC, CA, USA. <https://doi.org/10.5067/TELND-NC005> (2012).
-

Furthermore, it is stated at this website:

<https://reanalyses.org/atmosphere/merra2-references>

that both types of citations are needed.

Details

- l. 25: you define here the concept of V_{PSC} based on the 'existence' of PSCs; I would agree. However elsewhere (e.g., l. 45,71) you use 'formation'. These two terms should be used in a consistent way.

As noted above, in the version of the manuscript for which this review is based, the word "existence" was used twice: once in main and once in methods.

Upon revision, we now use "existence" 11 times: 5 in main and 6 in methods.

- l. 29: amounts → amount

Text had read:

due in part to larger amounts of chemical loss

Text changed to:

due in part to a larger amount chemical ozone loss

- l. 33 present → 2005 (present is 2021, yes?)

This phrase has been removed

- l. 38: suggested that

Change made

- l. 58: strictly speaking, activation may also occur on ice particles . . .

We have changed this sentence starting on line 56 to read:

Chemical loss of O_3 in the Arctic stratosphere occurs following the activation of chlorine on the surfaces of either cold sulphate aerosols^{19,20}, supercooled ternary (H_2SO_4 - HNO_3 - H_2O) solution droplets²¹ (STS), nitric acid trihydrate (NAT) particles²², or water ice when air is exceptionally cold.

with citations to:

19. Drdla, K. & Müller, R. Temperature thresholds for chlorine activation and ozone loss in the polar stratosphere. *Ann. Geophys.* **30, 1055–1073 (2012).**

20. Wegner, T. *et al.* Heterogeneous chlorine activation on stratospheric aerosols and clouds in the Arctic polar vortex. *Atmos. Chem. Phys.* **12, 11095–11106 (2012).**

21. Carslaw, K. S. *et al.* Stratospheric aerosol growth and HNO_3 phase depletion from coupled HNO_3 and water uptake by liquid particles. *Geophys. Res. Lett.* **21, 2479–2482 (1994).**

22. Hanson, D. & Mauersberger, K. Laboratory studies of the nitric acid trihydrate: Implications for the south polar stratosphere. *Geophys. Res. Lett.* 15, 855–858 (1988).

- I. 59: which is the second type of PSCs? Perhaps rephrase.

We have left the text “as is” because the community has for many years adopted “Type 2” PSCs to refer to water ice PSCs. In our opinion, the text is clear, especially with the new phrase added in response to the prior comment for line 58.

- I. 68: The profile of HNO₃ will be altered by sedimentation of PSCs, also in the Arctic, if the winters are cold (e.g. refs 23-25 in the paper).

The text starting on line 68 has been changed to read:

During cold Arctic winters, the profile of HNO₃ will be altered by the sedimentation of nitrate-bearing PSCs, termed denitrification²⁴⁻²⁷.

with citations to:

24. Fahey, D. W. *et al.* The detection of large HNO₃-containing particles in the winter Arctic stratosphere. *Science* 291, 1026–1031 (2001).

25. Northway, M. J. *et al.* An analysis of large HNO₃-containing particles sampled in the Arctic stratosphere during the winter of 1999/2000. *J. Geophys. Res. Atmos.* 107, 8298 (2002).

26. Molleker, S. *et al.* Microphysical properties of synoptic-scale polar stratospheric clouds: in situ measurements of unexpectedly large HNO₃-containing particles in the Arctic vortex. *Atmos. Chem. Phys.* 14, 10785–10801 (2014).

27. Waibel, A. E. *et al.* Arctic ozone loss due to denitrification. *Science* 283, 2064–2069 (1999).

- I. 113-114: Are all these simulations done with the same version of the ATLAS model (ERA5, JPL2015 etc). Of course this would be desirable. If yes (this is how I read the paper), how different are the ATLAS results reported here from the ozone loss estimates reported in earlier ATLAS studies (e.g., Wohltmann et al., 2013)?

Yes, all simulations have been done with the same version of the ATLAS model and with the same model setup. All simulations were specifically performed for this study and are not identical to runs in earlier studies (e.g. Wohltmann et al., *ACP* 2013; *ACP* 2017). The only exception is the 2019/2020 run, which is also used in Wohltmann et al. *GRL* (2021).

Wohltmann et al. *ACP* (2013) reports a value of 43 DU for the loss in the partial column 380-550 K (page 3920). This compares well with the estimate from this study (47 DU).

We agree that comparisons like these can give a general feel for the uncertainties in ΔO_3 expected resulting from different model versions, various meteorological data, and so on. At the same time, we would have to be careful with comparisons such as these, because we would expect newer model versions to perform better. All model runs reflect the state-of-the-art at the time when they were conducted, and these differences will often depend on the details of the model setup. Addressing this point would require a careful, detailed discussion that seems out of scope with the current paper, because:

- a) all results in the paper are found using the latest version of ATLAS with the same setup
- b) differences in ΔO_3 between various versions of ATLAS tend to be small, as noted above

- I. 120: ATLAS points do not start in 1993

Thanks for catching this! The sentence starting on line 116 has been changed to read:

Measured and modelled values of ΔO_3 display a compact, near linear relation with PFP for 1993 to 2020 (data) and 2005 to 2020 (ATLAS) (Fig. 1a).

- l. 132: why 1.2 as the exponent?

No change because the reason for this exponent is explained later in the paragraph, starting at line 142:

Here we use an exponent of 1.2 for EESC because this choice leads to the largest value of r^2 for the six ATLAS runs shown in Fig. 1b (see Methods).

- l. 136: linear relation of what? ozone depletion?

Text starting on line 138 has been changed to read:

Hassler et al.⁴² conducted an analysis of ozone depletion and recovery at the South Pole assuming a linear relation between ozone loss rate and EESC, even though they state the actual relation may be more complicated.

with citation to:

42. Hassler, B., Daniel, J. S., Johnson, B. J., Solomon, S. & Oltmans, S. J. An assessment of changing ozone loss rates at South Pole: Twenty-five years of ozonesonde measurements. *J. Geophys. Res. Atmos.* 116, D22301 (2011).

- l. 145: quantify how much larger

The sentence in question has been removed; please note the effect is now actually quantified, based on our new use of lower and upper limits of 1 and 1.4 for the exponent of EESC in the expression for OLP.

- l. 161: not sure if it makes sense to abbreviate Monte-Carlo with MC; see also the discussion on p-value

This abbreviation has been dropped

- l. 171: amount → number

Change made

- l. 187: not sure if this support is 'theoretical' or rather 'numerical' (also l. 199)

On line 193, "theoretical" was changed to "further"

On line 204, "theoretical" was changed to "numerical"

- l. 193: I suggest avoiding too many acronyms; FDF is an example for an acronym I would drop

This abbreviation has been dropped; we agree, there had been too many acronyms

- l. 203: might be helpful to indicate clearly, which lines you refer to here in the plots in Figs. 4 and 5

Text had read "bottom of the data envelope"; have changed line 209 to read:

open circles at bottom of the data envelope

- l. 234: HNO_3 will also change in the future; see discussion.

As noted above in our reply to the discussion, we have addressed by adding the following new sentence starting on line 338:

Finally, future levels of N₂O are also expected to rise⁴³, leading to higher levels of HNO₃ that will lead to more favourable conditions for the formation of PSCs²².

with citation to:

22. Hanson, D. & Mauersberger, K. Laboratory studies of the nitric acid trihydrate: Implications for the south polar stratosphere. *Geophys. Res. Lett.* 15, 855–858 (1988).

43. O’Neill, B. C. *et al.* The Scenario Model Intercomparison Project (ScenarioMIP) for CMIP6. *Geosci. Model Dev.* 9, 3461–3482 (2016).

This new sentence was placed at a latter part of the paper; we do not have room to mention rising N₂O in both places without removing some other sentence, and feel the latter placement is a better choice.

▪ I. 245: Indeed the station at Boulder is not a good representation of a zonal average, let alone polar averages. But this is not what you want to discuss here.

We have removed the sentence:

A balloon-borne record of stratospheric H₂O from NH mid-latitudes reveals a rise of 0.05 to 0.07 ppmv yr⁻¹ from 1965 to 2000, considerably larger than the rate of increase shown in Fig. 2d (ref. 54).

and not replaced because the inference of trends from the stratospheric H₂O record is so complicated, as detailed by:

Scherer *et al.*, *ACP*, 2008 <https://acp.copernicus.org/articles/8/1391/2008/acp-8-1391-2008.pdf>

Hegglin *et al.*, *JGR*, 2013 <https://agupubs.onlinelibrary.wiley.com/doi/pdf/10.1002/jgrd.50752>

Davis *et al.*, *ESSD*, 2016 <https://essd.copernicus.org/articles/8/461/2016/essd-8-461-2016.pdf>

Khosrawi *et al.*, *AMT*, 2018 <https://amt.copernicus.org/articles/11/4435/2018>

We simply could not provide a simple, one sentence summary to replace the sentence in question, and we are already at the word limit, so we have decided to remove this and not replace.

Furthermore, due to space limitations, the sentence:

The satellite-based data record for H₂O that affords coverage of the polar regions starts in 1984, but trends are difficult to discern due to offsets between retrievals from various instruments that are commonly larger than the expected increase in polar H₂O since 1984 (ref ⁶⁸).

with reference to:

68. Davis, S. M. *et al.* The Stratospheric Water and Ozone Satellite Homogenized (SWOOSH) database: a long-term database for climate studies. *Earth Syst. Sci. Data* 8, 461–490 (2016).

was moved to **Stratospheric H₂O** section of Methods (line 599), because this seemed to be a more appropriate location.

▪ I. 288: EESC^{1,2}·PPF at the end of the century is not higher than contemporary values for all scenarios, is it? (See the two bottom panels of Fig. 9.)

Great catch. One line 283 we have changed

are higher than contemporary values.

to

are higher than contemporary values of these quantities for the SSP5-8.5 and SSP3-7.0 simulations.

- l. 321-325: I do not quite understand the point of “prescribed ozone”. When ozone is prescribed, the ΔO_3 is known before the run. In other words, if I prescribe the ozone from one of the full chemistry GCMs, the resulting ozone will be the same? How are the ozone values ‘prescribed’? I would not put much weight on simulations with prescribed ozone.

The text had read:

The temporal evolution of EESC^{1,2}×PFP and ΔO_3 found using results from the four GCMs with interactive chemistry is somewhat lower (i.e., about 20 to 25% at end of century) than that found for the other 16 CMIP6 GCMs

which we now see would have been confusing. Here, the quantity ΔO_3 had been intended to represent the ozone column loss found by the regression of ΔO_3 versus EESC^{1,2}×PFP shown in Fig. 1b, combined with GCM-based values of EESC^{1,2}×PFP.

PFP was (and still is) the only quantity we directly based on GCM output.

The specific text in question, starting on line 317, has been changed to read:

The temporal evolution of OLP found using results from the four GCMs with interactive chemistry is about 20 to 25% lower at end of century than that found for the other 16 GCMs; nonetheless, ΔO_3^{REG} remains close to the contemporary value until the end of century for the SSP3-7.0 and SSP5-8.5 simulations conducted using these interactive GCMs (see Methods).

- l. 334: Figs. 8 and 9 do not show ΔO_3

Actually, these figures did show ΔO_3 based on the right-hand-side y-axis.

Upon revision, we have placed ΔO_3 on the left-hand-side y-axis of Figures 8 and 9, to provide greater emphasis for this key quantity.

- l. 337: planetary wave activity (and change) is not independent of the BDC; also gravity waves play a role to some extent (e.g., Li et al., 2008; Bunzel and Schmidt, 2013).

The text in question had read:

Future levels of Arctic column ozone during late winter and early spring are expected to increase due to factors such as intensification of the BDC, upper stratospheric cooling, and possible changes in planetary wave activity that will exert a strong influence on the abundance of column ozone within the Arctic vortex during its formation in early winter^{13,59}.

with citations to:

13. Bednarz, E. M. *et al.* Future Arctic ozone recovery: the importance of chemistry and dynamics. *Atmos. Chem. Phys.* 16, 12159–12176 (2016).
59. Dhomse, S. S. *et al.* Estimates of ozone return dates from Chemistry-Climate Model Initiative simulations. *Atmos. Chem. Phys.* 18, 8409–8438 (2018).

The text starting on line 331 has been changed to read:

Future levels of Arctic column ozone during late winter and early spring are expected to increase due to factors such as intensification of the BDC, upper stratospheric cooling, as well as possible changes in planetary and gravity wave activity that exert a strong influence on the abundance of column ozone within the Arctic vortex during its formation in early winter and dynamically induced increases during winter^{15,16,58}. Langematz et al.¹⁵ project maximum V_{PSC} to occur around 2060 with a subsequent decline due to enhanced dynamical warming of the Arctic vortex in February and March, based on simulations conducted with their CCM.

with citations to:

15. Langematz, U. *et al.* Future Arctic temperature and ozone: The role of stratospheric composition changes. *J. Geophys. Res. Atmos.* **119**, 2092–2112 (2014).
16. Bednarz, E. M. *et al.* Future Arctic ozone recovery: the importance of chemistry and dynamics. *Atmos. Chem. Phys.* **16**, 12159–12176 (2016).
58. Dhomse, S. S. *et al.* Estimates of ozone return dates from Chemistry-Climate Model Initiative simulations. *Atmos. Chem. Phys.* **18**, 8409–8438 (2018).

We studied the two suggested citations, as well as this classic paper:

Butchart et al., *J. Clim.*, 2010: <https://doi.org/10.1175/2010JCLI3404.1>

and decided the best course of action, to be responsive to the comment about gravity waves, was to add this term to the sentence with a citation to Langematz et al. (2014), since this paper is already cited in several other places. We unfortunately are at the 70 citation limit, so to cite one of the two suggested papers, we would have to remove some other citation.

Based upon how the sentence is written, wherein we provide a list of dynamical factors, we did not feel it was necessary to re-write to point out that “planetary wave activity (and change) is not independent of the BDC”, as had been noted by the reviewer. We did not make this change because we feel this detail is known by the specialists, and furthermore all of the dynamical factors are connected in some fashion. Finally, the sentence starting “Langematz et al.¹⁵ project ...” was added to emphasize the uncertainty in how the real Arctic vortex will evolve, due to planetary wave activity.

- l. 347: continue → continues

This sentence has been removed and replaced with a stronger final sentence starting on line 346:

Consequently, anthropogenic climate change has the potential to partially counteract the positive effects of the Montreal Protocol in protecting the Arctic ozone layer.

to provide greater emphasis on the most important aspect of our work, *the future state of the Arctic ozone layer*.

- l. 350: but in very cold Arctic winters (e.g., 2020) there could be dehydration also in the Arctic.

Dehydration is unusual for the Arctic. MLS data indicate the presence of a single dehydration event at the start of Feb 2020 that lowered the abundance of H₂O, at 475 K, from about 5.7 to 5.2 ppm. This decline in H₂O would have lowered T_{NAT} by less than 0.5 K: e.g., from 196.14 K to 195.76 K at 50 hPa for HNO₃ = 9 ppb (our climatological value of HNO₃ at 50 hPa) or from 193.09 to 192.72 for HNO₃ = 1 ppb. Khosrawi et al. *ACP* (2017) reported that dehydration in the Arctic vortex during the winter of 2015/2016 led to a loss of H₂O between 0.6 and 1 ppmv.

Since the effect of this amount of dehydration on T_{NAT} is so small, and dehydration in the Arctic has been modest to date, we have left the text unchanged.

In the future, if the Arctic does indeed continue to cool, perhaps dehydration will become more important. If so, as detailed below, the effect would like provide a modest reduction in Arctic ozone loss compared to a model simulation otherwise identical, but for which no dehydration were to occur. We quote below a paragraph related to dehydration and Arctic ozone from Section 3.3.4 from the 2002 WMO/UNEP Scientific Assessment of Ozone Depletion Report chapter on Polar Ozone, which three of us (Markus, Ross, and Peter) helped write:

Dehydration, unlike denitrification, can moderate ozone loss for two reasons (Portmann et al., 1996; Chipperfield and Pyle, 1998). First, in a drier atmosphere it is harder for PSCs to form. Second, heterogeneous reaction rates that lead to active chlorine production drop exponentially with decrease in relative humidity. Sensitivity studies show that dehydration (to the level of ice saturation) in the Antarctic can decrease column ozone loss by about 20% (Portmann et al., 1996; Brasseur et al., 1997). No large-scale model calculations have yet been performed to evaluate the role that dehydration may play in Arctic ozone loss and recovery in the future. However, it is unlikely that climate change in the near future could cause extensive dehydration in the Arctic region. Some air mass trajectory statistical analyses indicate that even a substantial cooling of lower stratospheric temperatures (by 3 to 4 K) is still insufficient to trigger the occurrence of severe dehydration in the Arctic vortex (Tabazadeh et al., 2000).

We have chosen to not address this in our paper, because there has been no rigorous assessment of the role of dehydration of the future of Arctic ozone, and most importantly because if any Arctic winter in the near future were cold enough to experience dehydration on the scale of that which regularly happens in the Antarctic stratosphere, the resulting ozone depletion would be enormous whether or not some of the resulting loss were moderated by the loss of H_2O .

- I. 355: there is also denitrification in the Arctic, see discussion

We have done our best to address this comment in the “Nitric Acid” section, near the top of this review.

- I. 370: one could check the sensitivity of the results on the choice of 36 s^{-1} ; this might also address the ‘capping’ issue mentioned in I. 379

We have added a new section to Methods entitled “**Vortex Boundary**” as well as a new figure to the supplement. Further detail is provided in our reply, above, to the first point under label **Points from the first review**.

- I. 408 ‘rarely present’ is likely too strong, I think they were present but the PFP values were less pronounced

We agree. The text starting on line 409 has been changed to read:

Supplementary Fig. 3 supports our conclusion, shown in Fig 3d of the main article, that conditions conducive for the existence of PSCs tended to be less common between 1965 and 1979, compared to the past few decades.

- I. 472: Perhaps this calculation could be better explained. Where do the values 13.6 d and $4.59 \text{ d decade}^{-1}$ come from? What is the advantage of this calculation compared to the statistical significance calculation below?

These numerical values were chosen such that the lower and upper limits of PFP match the values of PFP from ERA5 in a statistical fashion, as now explained starting on line 471 of Methods:

For this set of MC simulations, one million time series of PFP were generated for a 41 year long record (matching the time period 1980-2020), each with PFP distributed between a lower bound of 0 and an upper bound that starts at 13.6 d (first winter) and rises with a slope of

4.59 d decade⁻¹. Each PFP data point is uniformly, randomly distributed between the time varying upper bound and the lower bounds, chosen to match the lower and upper limits of PFP from ERA5 in a statistical fashion.

- I. 497-500: I agree, but Fig. 3 only shows the fitting uncertainty. Could you include this statistical significance information also in Fig. 3?

p-values have been added to Figure 3, as well as to Supplementary Figure 10.

- I. 516: can you quantify 'very few', perhaps a value in %?

The meaning of "very few" had been explained in the paragraph starting on line 521 of the reviewed version of the paper. To make clear to the reader that an explanation is to follow, we have modified the sentence in question, starting on line 518, to read:

As explained below, very few of the randomly generated time series yield a high value of $S_{\text{PFP-LM}}$ in combination with a low value of $\Delta S_{\text{PFP-LM}}$.

- I. 544: what is assumed here as sulphate area density? Are these numbers available in the data archive created by the authors?

We used the formulation for sulphate area density given in the caption of figure 5 of Drdla and Müller (2012). The sentence starting on line 556 has been changed to read:

The term aerosol reactivity potential (ARP) is similar to PFP, except in Eq. (2) the quantity T_{NAT} is replaced by T_{ACL} , which represents the temperature at which chlorine is activated. Values of T_{ACL} are computed as a function of H_2O and sulphate surface area density at 210 K using equation (2) and information in the caption of figure 5 of Drdla and Müller¹⁹.

Of course, with a citation to:

19. Drdla, K. & Müller, R. Temperature thresholds for chlorine activation and ozone loss in the polar stratosphere. *Ann. Geophys.* 30, 1055–1073 (2012).

- I. 555: Typo: two times PFP here?

Thanks, typo has been fixed.

- I. 575: Cl is also changing with time

Yes, but since the effect of stratospheric Cl on CH_4 in the Arctic vortex is "second order" (i.e., stratospheric loss of CH_4 occurs primarily via reaction with OH), the effect of changing stratospheric Cl is small. It should be clear to anyone reading this deep that stratospheric Cl varies over time. We prefer therefore to keep text unchanged in this location.

- I. 582: in which way an outlier?

Great point. Text starting on line 603 has been modified to include the phrase "large future rise in stratospheric H_2O " and reads as follows:

neglecting results from the UKESM1-0-LL GCM, because the results from this GCM seem to be an outlier (large future rise in stratospheric H_2O) compared to results from the other nine GCMs.

- I. 597: how similar? I assume all runs were performed based on ERA5. Perhaps a bit more information could be given here.

The model setup is identical except for the dates of model initialization and the initialization of the chemical species. The model description was kept short because the Methods section is already much longer than the suggested word count and the ATLAS model is well described in the literature.

Nonetheless, we have added these two sentence starting on line 619:

Descent rates are calculated directly from the heating rates provided by the ERA5 reanalysis. From the two different options provided by ECMWF, we use the total (“all sky”) heating rates and not the “clear sky” heating rates.

and the phrase “as described in section 6.1 of Wohltmann et al. 2017 (ref 67)” to the sentence starting on line 630, which now reads:

A common deficiency of CTMs is a pronounced discrepancy between measured and modelled HCl mixing ratios in the Antarctic polar vortex, as described in section 6.1 of Wohltmann et al. 2017 (ref 67).

with a citation to:

67. Wohltmann, I., Lehmann, R. & Rex, M. A quantitative analysis of the reactions involved in stratospheric ozone depletion in the polar vortex core. *Atmos. Chem. Phys.* 17, 10535–10563 (2017).

- I. 600: ClONO₂ is from a climatology and HCl from observations – But then total chlorine is not correct in every case. Or do you adjust the values of the climatology? Could you provide a bit more background here?

The initialization of ClONO₂ and Cl_y is indeed one of the more challenging aspects of the model initialization. Unfortunately, we were not able to go into more detail in the paper because of length restrictions.

There are three factors that play a role: a) The need for a global initialization of all species without any gaps, b) the sparsity of ClONO₂ measurement data, c) perhaps a little bit surprisingly, rather large uncertainties in determining Cl_y.

While there is usually sufficient data for a reasonable initialization of HCl (e.g. daily global data from the MLS satellite instrument), global measurement coverage for ClONO₂ is sparse. There are basically two satellite instruments that provide data for ClONO₂: MIPAS and ACE-FTS. MIPAS data were not available for use in this study. ACE-FTS data coverage is sparse, since ACE-FTS is a solar occultation instrument. It is therefore only possible to initialize from ACE-FTS globally when using long-term averages. Consequently, we used a climatology for the years 2004-2013 (Koo et al., 2017). Note that we did not correct for the trend in Cl_y between 2004-2013 and the respective years of the model runs. This correction would have been smaller than 0.1 ppb in all cases and was deemed to be negligible.

The figure above shows the average HCl and ClONO₂ profiles for the 1 October 2019 and latitudes greater 85 degrees north obtained in this manner, and the sum of both species. For comparison, the total chlorine in chlorine source gases (CFCs etc.) at the tropopause in 2016 according to the WMO assessment is also shown (as an estimate of “maximum Cl_y”).

The Cl_y obtained in this manner likely has some inaccuracy. Somewhat surprisingly, it is however not that easy to determine the level of accuracy, because measurements of Cl_y in the polar stratosphere are surprisingly uncertain.

There are basically two methods to obtain Cl_y from measurements. The first method is to calculate the sum of all chlorine-containing species. This is most easy in autumn or early winter, when most of the chlorine is either present as HCl or ClONO₂, and no additional species with sparse measurement coverage as ClO and Cl₂O₂ (which can be inferred from ClO, as done for instance by Canty et al., JGR, 2016) having to be considered. This method is more or less what we have done, with the drawback that we could not use actual global measurements for the given date for ClONO₂.

The second method is to infer Cl_y from the tropospheric mixing ratios of CFCs and other source gases, which are very well known. The problem here is that the amount of Cl_y that is set free from the source gases and the total amount of chlorine in the source gases depend on the age of air and the fractional release factors of the source gases. The latitude and altitude dependence of these quantities are quite uncertain.

Another method would be to initialize Cl_y from a tracer-tracer relationship with CH₄ or N₂O, but this makes the results dependent on the quality of the tracer-tracer relationships (which are often derived from a few local campaign measurements), the trend of Cl_y between the date of measurement and the time of the model run, as well as the quality of the N₂O or CH₄ data (and their trends).

The figure above shows that the maximum Cl_y from the sum of HCl and ClONO₂ is in reasonable agreement with the maximum Cl_y from the source gases. We did however explore initializing Cl_y from a tracer-tracer relationship (and then to calculate ClONO₂ as Cl_y-HCl). Results from this approach gave unrealistic results (e.g. oscillations and negative ClONO₂ mixing ratios) for 2019. In part, this was caused by very unusual N₂O measurements from MLS for this winter, shown above in reply the comment entitled **Good quantitative understanding**. For this reason, the plot just above only shows maximum Cl_y from tropospheric source gases, and not Cl_y taking into account age of air and fractional release factors.

• I. 604: Descent rates are important for polar ozone loss (e.g., Strahan et al., 2014), how are they calculated in ATLAS for the runs discussed here?

Good point. We had not provided this information in the prior version of the paper due to length restrictions. Descent rate is an important detail. and we have added the following two sentences to Methods, starting on line 619 (as noted above in reply to the comment about 597):

Descent rates are calculated directly from the heating rates provided by the ERA5 reanalysis. From the two different options provided by ECMWF, we use the total (“all sky”) heating rates and not the “clear sky” heating rates.

- I. 606: citations for this ‘common deficiency’?

The overestimation of HCl in the polar vortices in winter is a well-known deficiency of most CTMs (including e.g. CLaMS, ATLAS or SD-WACCM), which points to a lack of understanding of the chemical and physical processes changing HCl in the polar vortices.

As noted above in reply to the comment about 597, we have on line 632 the phrase:

as described in section 6.1 of Wohltmann et al. 2017 (ref ⁶⁷).

with a citation to our 2017 ACP paper.

There are other studies that discuss the HCl discrepancy in detail, such as Brakebusch et al., JGR, 2013, Solomon et al., JGR, 2015, Grooß et al., ACP, 2018. Given the non-essential role of the HCl deficiency for the primary thesis of our paper, and the limitation on the overall number of citations, we have decided to not add citations to any of these studies. We do acknowledge the central importance of all three papers with respect to this issue.

- I. 973: You could also put the code on your Zenodo archive?

We include the statement “Code relating to this study is available from the corresponding author on request” because our code is extensive, not well documented and makes use of different commercial software packages, which cannot be provided due to license regulations. It will simply be best for us to know who needs what, rather than post all of the code in such an undocumented fashion. If our paper proceeds and if this statement poses a problem with the editor, we’ll seek to provide the code as best we can.

- Figure 1: Here (and perhaps elsewhere in the paper) it should be explained that the column ozone loss shown here is calculated for the partial column 14-24 km. Assuming that is the case – I am guessing here on the basis of previous publications.

You are correct and this detail has been added to the caption of Figure 1.

References

- Bunzel, F. and Schmidt, H.: The Brewer-Dobson Circulation in a Changing Climate: Impact of the Model Configuration, J. Atmos. Sci., 70, 1437-1455, <https://doi.org/10.1175/JAS-D-12-0215.1>, 2013.
- Lait, L. R.: An alternative form for potential vorticity, J. Atmos. Sci., 51, 1754-1759, 1994.
- Li, F., Austin, J., and Wilson, J.: The Strength of the Brewer-Dobson Circulation in a Changing Climate: Coupled Chemistry-Climate Model Simulations, J. Climate, 21, 40 - 57, <https://doi.org/10.1175/2007JCLI1663.1>, 2008.

Thanks for an amazingly thorough review, which has led to a significantly improved manuscript!

We hope you will choose to receive credit for this review by revealing your identity under the new Nature system, that should allow for any reviewer to be named, as all of the reviews (and our responses) will be published along with the final paper.

Reviewer comments, third round

Reviewer #3 (Remarks to the Author):

Please find my final comments attached as a pdf-file.

I recommend accepting this paper.

Rolf Müller

Final comments on “The influence of climate change on chemical loss ...” by P. von der Gathen et al.

General

The authors have revised the paper another time, largely in response to the comments of this reviewer; I think they have done a very good job in responding to the comments. In addition they have improved their work; I think that the inclusion of the ECMWF ERA5 BE data set is a substantial improvement.

As for the previous versions, my overall conclusion is that the paper should be published in *Nat. Commun.*. The few remaining comments (see below) regard details and wording. The authors might want to take these comments into account when writing the final version of this paper. I suggest accepting the paper for publication in *Nat. Commun.*.

Also, as suggested by the authors, I am happy to sign my reviews and agree that they can be published along with the replies by the authors.

Rolf Müller

Forschungszentrum Jülich, Germany

Writing Reviews

Reviews are still written these days by a single author; consultation of colleagues is possible but not common. This is in contrast to almost all other scientific writing, including single author papers (let alone PhD. theses) that are always subject to some review and commenting. Insofar reviews (including those discussed here) may contain some “rough edges” that are frequently ironed out in other publications.

Some Details

- l 11: you could give a number here for the rise; not sure if there is enough space in the abstract for that.
- l 56: You say “on” the surfaces here; the recent PSC review (Tritscher et al., Rev. Geophys., 2021, Table 6) uses the formulation: “on/in liquid SSA/STS droplets, much less on NAT or ice particles”.
- l 70: cold \longrightarrow low
- l 73: I am not sure if the thresholds are “closely related”. I think they are different. But they lead to very similar temperature thresholds indeed.
- l 120/121: I think “minimum in early 1993” is not quite correct. If I inspect Fig. 2a, I see no minimum in 1993.
- l. 163: “form” or “sustain”?
- l 166: “p-values”: you correctly pointed out that this term is explained in some detail in the methods section (and also in the caption of Table 1). Nonetheless, if somebody reads the paper first he/she is exposed here

first time to the term “p-values” and it is not so clear that at this point the Methods should be consulted. Perhaps you could reformulate the part of the sentence after the colon by stating the most important message first. \approx This indicates/shows a statistical significance at better ..., i.e., p-values are less than ... (see Methods).

- l 215-216: I do not think the difference between CMIP5 and CMIP6 for 1980-2020 (Fig. 6a) is very pronounced. I think one could drop the sentence starting “ Again, there is ...”.
- l 227: Is it clear here which subset of GCMs?
- l 256: say briefly why this is expected?
- l 339: I would formulate “formation and existence” here; actually ref. 22 is not on the formation of PSCs.
- l 1021 (caption of table 1): yields \longrightarrow yield

Reviewer #3 Comments:

General

The authors have revised the paper another time, largely in response to the comments of this reviewer; I think they have done a very good job in responding to the comments. In addition they have improved their work; I think that the inclusion of the ECMWF ERA5 BE data set is a substantial improvement.

As for the previous versions, my overall conclusion is that the paper should be published in *Nat. Commun.* The few remaining comments (see below) regard details and wording. The authors might want to take these comments into account when writing the final version of this paper. I suggest accepting the paper for publication in *Nat. Commun.*

Also, as suggested by the authors, I am happy to sign my reviews and agree that they can be published along with the replies by the authors.

Rolf Müller

Forschungszentrum Jülich, Germany

Rolf, thanks for the kind words about this manuscript, your careful reviews of numerous versions of the paper, and for signing the review.

We learned in the author guidelines that *Nature Communications* does not allow us to formally thank reviewers in the acknowledgement section of the paper. We express our very sincere thanks for the enormous amount of time you spent reviewing this paper, and your tremendously helpful comments that resulted in a much improved final manuscript.

Writing Reviews

Reviews are still written these days by a single author; consultation of colleagues is possible but not common. This is in contrast to almost all other scientific writing, including single author papers (let alone PhD. theses) that are always subject to some review and commenting. Insofar reviews (including those discussed here) may contain some “rough edges” that are frequently ironed out in other publications.

Your reviews have all been fantastic: very detailed and filled with so many helpful suggestions.

Some Details

- I 11: you could give a number here for the rise; not sure if there is enough space in the abstract for that.

This is the only comment upon which we decided not to change the paper. The abstract, which we have left unchanged, is 148 words. Since the numerical values given in the paper are for the local maxima of PSC formation potential, and since PSC formation potential is itself a quantity that requires some description, we felt we would have had to re-write much of the abstract to explain the meaning of any numerical value that would have been inserted.

- I 56: You say “on” the surfaces here; the recent PSC review (Tritscher et al., Rev. Geophys., 2021, Table 6) uses the formulation: “on/in liquid SSA/STS droplets, much less on NAT or ice particles”.

The text has been changed to read:

Chemical loss of O₃ in the Arctic stratosphere occurs following the activation of chlorine on or within the surfaces of either cold sulphate aerosols^{61,62} and supercooled ternary (H₂SO₄-HNO₃-H₂O) solution droplets⁶³ (STS), and on the surfaces of nitric acid trihydrate (NAT) particles⁶⁴; or water ice when air is exceptionally cold.

- I 70: cold → low

Thanks for catching this error; change has been made.

- I 73: I am not sure if the thresholds are “closely related”. I think they are different. But they lead to very similar temperature thresholds indeed.

We arrive at remarkably similar conclusions based upon consideration of the temperature at which chlorine is activated on aerosols^{61,70}, rather than T_{NAT}, because these two temperature thresholds are so closely related similar (see Methods).

- I 120/121: I think “minimum in early 1993” is not quite correct. If I inspect Fig. 2a, I see no minimum in 1993.
Here, we meant that EESC had the lowest value over the time period in question, in year 1993.
We see how the text could have been confusing.
The text has been changed to read:
This behaviour occurs because over this time period, the abundance of stratospheric halogens, commonly represented by Equivalent Effective Stratospheric Chlorine (EESC)⁸² (Fig. 2a), varies by only ~11% between ~~the maximum in mid-2001 and the minimum value in early 1993 and the maximum in mid-2001.~~
- I. 163: “form” or “sustain”?
“form” has been changed to “sustain” on line 17
- I 166: “p-values”: you correctly pointed out that this term is explained in some detail in the methods section (and also in the caption of Table 1). Nonetheless, if somebody reads the paper first he/she is exposed here first time to the term “p-values” and it is not so clear that at this point the Methods should be consulted. Perhaps you could reformulate the part of the sentence after the colon by stating the most important message first ≈ This indicates/shows a statistical significance at better . . . , i.e., p-values are less than . . . (see Methods).
The text has been changed to read:
We have conducted Monte-Carlo simulations to assess the statistical significance of $S_{\text{PFP-LM}}$ and the 1σ uncertainty in $S_{\text{PFP-LM}}$ ($\Delta S_{\text{PFP-LM}}$) found using the ISA selection procedure (see Methods). ~~These simulations indicate statistical significance at better than the 2σ confidence level for this important metric of the trend in PFP^{LM}, based upon p-values are all less than 0.001 for $S_{\text{PFP-LM}}/\Delta S_{\text{PFP-LM}}$ from all four meteorological data centres that are less than 0.001 (see Methods, Table 1), indicating statistical significance at better than the 2σ confidence level for this important metric of the trend in PFP^{LM}.~~
- I 215-216: I do not think the difference between CMIP5 and CMIP6 for 1980-2020 (Fig. 6a) is very pronounced. I think one could drop the sentence starting “Again, there is . . . ”.
The sentence in question has been deleted, as suggested.
- I 227: Is it clear here which subset of GCMs?
The text has been changed to read:
We interpret the results in Fig. 6a as follows: there is strong similarity in the four observationally based estimates of $S_{\text{PFP-LM}}$, and this value is consistent with a subset of the GCMs (i.e., those with the largest values of $S_{\text{PFP-LM}}$).
- I 256: say briefly why this is expected?
The text has been changed to read:
The trend in $S_{\text{PFP-LM}}$ found using archived output from the EC-Earth3 GCM for SSP5-8.5 increases from 2.27 ± 0.13 d decade⁻¹ for time-invariant H₂O (Fig. 5a) to 3.93 ± 0.13 d decade⁻¹ when both of the factors driving the potential future rise in stratospheric H₂O are considered, because a more humid future stratosphere is more conducive to the chlorine activation and the formation of PSCs.
- I 339: I would formulate “formation and existence” here; actually ref. 22 is not on the formation of PSCs.
The text has been changed to read:
Finally, future levels of N₂O are expected to rise⁸⁵, leading to higher levels of HNO₃ that will lead to more favourable conditions for the formation and existence of PSCs⁶⁴.
- I 1021 (caption of table 1): yields → yield

“yields” has been changed to “yield”.

Thanks for the incredibly detailed review to find this very subtle typo.

Other changes to the paper:

a) the title has been changed to “Climate change favours large seasonal loss of Arctic ozone” as suggested to us in the Author Guidelines

b) the 42 citations to the GCM models have been incorporated into the main text; thanks for allowing these papers to be cited

c) the Acknowledgements section has been shortened and the reviewers are no longer thanked. This section is probably still longer than would be preferred by the editorial team; many of the sentences are boiler plate material we are asked to include, in Acknowledgements, when using meteorological data from various centres or climate model output from various archives

d) a new sentence reading:

The mean and 1σ standard deviation of $S_{\text{PFP-LM}}$ over 1980 to 2020 from these four centres is $4.26 \pm 0.45 \text{ d decade}^{-1}$.

has been added to line 167, because while contemplating changing the abstract to reply to the first comment (for line 11), we realized the numerical value of this important red point, shown on the upper panel of Figure 3, had not previously been provided in the paper.

e) the section header details and sub-section headers have been modified to conform with the instructions in the Author Guidelines.

f) additional changes to address the Authors Guidelines and minor bugs.